**Crossing hydrological and geochemical modeling to understand the spatiotemporal**
**variability of water chemistry in a headwater catchment (Strengbach, France)**
Julien Ackerer, Benjamin Jeannot, Frederick Delay, Sylvain Weill, Yann Lucas, Bertrand Fritz,
Daniel Viville, François Chabaux
Laboratoire d'Hydrologie et de Géochimie de Strasbourg, Université de Strasbourg, CNRS,
ENGEES, 1 rue Blessig, 67084 Strasbourg Cedex, France
*Corresponding authors:
Julien Ackerer (julien.ackerer@orange.fr), Benjamin Jeannot (bjeannot.pro@gmail.com),
Frederick Delay (fdelay@unistra.fr), François Chabaux (fchabaux@unistra.fr)

**Abstract**

Understanding the variability of the chemical composition of surface waters is a major issue for the scientific community. To date, the study of concentration-discharge relations has been intensively used to assess the spatiotemporal variability of the water chemistry at watershed scales. However, the lack of independent estimations of the water transit times within catchments limits the ability to model and predict the water chemistry with only geochemical approaches. In this study, a dimensionally reduced hydrological model coupling surface flow with subsurface flow (i.e., the Normally Integrated Hydrological Model, NIHM) has been used to constrain the distribution of the flow lines in a headwater catchment (Strengbach watershed, France). Then, hydrogeochemical simulations with the code KIRMAT (i.e., KInectic Reaction and MAss Transport) are performed to calculate the evolution of the water chemistry along the flow lines. Concentrations of dissolved silica ($H_4SiO_4$) and in basic cations ($Na^+$, $K^+$, $Mg^{2+}$, and $Ca^{2+}$) in the spring and piezometer waters are correctly reproduced with a simple integration along the flow lines. The seasonal variability of hydraulic conductivities along the slopes is a key process to understand the dynamic of flow lines and the changes of water transit times in the watershed. The covariation between flow velocities and active lengths of flow lines over changing hydrological conditions reduces the variability of water transit times and explains why transit times span much narrower variation ranges than the water discharges in the Strengbach catchment. These findings demonstrate that the general chemostatic behavior of the water chemistry is a direct consequence of the strong hydrological control of the water transit times within the catchment. Our results also show that a better knowledge of the concentration-mean transit time relations (C-MTT relations) is an interesting new step to understand the diversity of C-Q shapes for chemical elements. The good matching between the measured and modeled concentrations while respecting the

water-rock interaction times provided by the hydrological simulations also shows that it is
possible to capture the chemical composition of waters using simply determined reactive
surfaces and experimental kinetic constants. The results of our simulations also strengthen
the idea that the low surfaces calculated from the geometrical shapes of primary minerals are
a good estimate of the reactive surfaces within the environment.
**1- Introduction**
Understanding the effects of ongoing climatic changes on the environment is a major issue for
the coming years. The global increase of temperature is expected to affect the hydrological
cycle at a large scale, and providing a precise estimation of its repercussion on the evolution
of soils and on the chemistry of waters remains difficult. This challenge results from the wide
diversity of hydrological, geochemical, and biological processes, and of their coupling, that
operate at the Earth's surface (e.g., Gislason et al., 2009; Goddéris et al., 2013; Beaulieu et al.,
2012; 2016). Up today, the study of concentration-discharge relations (C-Q relations) has been
intensively used to assess the coupling between hydrological and geochemical processes at
the hillslope or watershed scales (Godsey et al., 2009; Kim et al., 2017; Ameli et al., 2017;
Diamond and Cohen, 2018).
C-Q relations are acknowledged to integrate critical zone structure, the hydrological dynamics
and the geochemical processes of watersheds (Chorover et al., 2017). Recent studies debated
to which extend the chemical variability of waters is explained by a mixing of different water
sources (Zhi et al., 2019), the chemical contrasts between deep and shallow waters (Kim et
al., 2017), the variability of transit times (Ackerer et al., 2018) and/or seasonally variable flow
paths (Herndon et al., 2018). It is clear that a good knowledge of the water flow paths and of
their seasonal variability is an important new step to better constrain the water transit times
within catchments, and then, to correctly understand the temporal fluctuations of the
composition of waters. Modeling such variability of water flow paths and water geochemical
composition would require further development of modeling approaches able to combine
hydrological and geochemical processes (e.g. Steefel et al., 2005; Kirchner, 2006).
Recent efforts in hydrological modeling were conducted to develop spatially distributed
approaches that better consider the interplay between surface and subsurface processes (e.g.,
Gunduz and Aral, 2005; Kampf and Burges, 2007; Camporese et al., 2010). Due to the
complexity of flows in the hydrological processes, many modeling approaches are based on
the full resolution of Richard's and Saint Venant equations to correctly describe the
interactions between stream, overland and subsurface waters (Kampf and Burges, 2007).
These approaches have shown their ability to capture the hydrological functioning of various
watersheds, knowing that the full resolution of Richard's and Saint Venant equations requires
long computational times and faces calibration and parameterization difficulties (Ebel and
Loague, 2006; Mirus et al., 2011). Questions have been raised regarding the optimal
complexity of the equations that are needed to correctly treat the hydrology of catchments in
their surface and subsurface compartments with reasonable computation times (Gunduz and
Aral, 2005).
Low-dimensional models have attracted growing interest because they represent an
interesting compromise between equation complexity, computational time, and result
accuracy (Pan et al., 2015; Hazenberg et al., 2016; Weill et al., 2013; 2017; Jeannot et al.,
2018). The reduction of dimensionality is mainly associated with a subsurface compartment
(including both the vadose and the saturated zones) modeled as a two-dimensional layer.
Some low-dimensional models, as the one employed in this study, can solve subsurface flow
via an integrated Richard's equation, meaning that flow and transport processes are
integrated over a vertical direction or a direction normal to bedrock, and manipulate averaged
(integrated) hydrodynamic properties. This type of low-dimensional approach recently
demonstrated its ability to reproduce the results from fully dimensioned approaches in small
catchments while reducing computational costs (Pan et al., 2015; Jeannot et al., 2018).
Nonetheless, the water transit times calculated from these depth-integrated models are
rarely confronted with the water-rock interaction times inferred from hydrogeochemical
modeling of water chemistry in watersheds.
For its part, the understanding of the hydrogeochemical functioning of the critical zone has
been significantly advanced by the implementation of reactive-transport laws in geochemical
modeling codes (Steefel et al., 2005; Lucas et al., 2010; 2017; Goddéris et al., 2013; Li et al.,
2017). These developments allow for considering a variety of processes, such as flow and
transport processes, ion exchanges, biogeochemical reactions, and the interplay between
primary mineral dissolution and secondary mineral precipitation (Moore et al., 2012;
Lebedeva and Brantley, 2013; Ackerer et al., 2018). Reactive transport models have been used
to explore a wide variety of scientific issues, including the study of global atmospheric $CO_2$
consumption by weathering reactions (Goddéris et al., 2013; Li et al., 2014), the formation
and evolution of soil and regolith profiles (Maher et al., 2009; Navarre-Sitchler et al., 2009;
Lebedeva and Brantley, 2013), and the variability of water quality and chemistry in the
environment (Lucas et al., 2010; 2017; Ackerer et al., 2018). However, these approaches
usually rely on a simple 1D flow path through a regolith column or along a hill slope to model
flow in the system (e.g. Maher, 2011; Moore et al., 2012; Lucas et al., 2017; Ackerer et al.,
2018). 1D reactive-transport models are useful to discuss the key processes involved in the
regolith formation and in the acquisition of the water chemical composition, but these models
cannot consider the complexity of the flow trajectories in watersheds, and hence, its effects
on the water chemistry.
A new step is therefore necessary for the development of hydrogeochemical modeling
approaches that are applicable at the watershed scale and are able to integrate the complexity
of the water flows and the diversity of the water-rock interaction processes. Recent efforts
have been undertaken in the direction of merging hydrological and geochemical codes, with
for example, the parallel reactive transport code ParCrunchFlow (Beisman et al., 2015), or the
coupled hydrogeochemical code RT-Flux-PIHM (Bao et al., 2017; Li et al., 2017).  As an
alternative to fully dimensioned codes, this work proposes an original low-dimensional
approach, with relatively short computation times and applicable at the watershed scale. This
study is combining for the first time in this manner the results from a hydrological low-
dimensional (depth-integrated for the subsurface) but spatially distributed model (NIHM) with
a reactive-transport model (KIRMAT). The combination allows for simulating over time and
space the flow trajectories, the flow rates, the weathering reactions, and the evolution of the
water chemistry within a headwater system, the Strengbach catchment.
This catchment is one of the reference observatories of the French critical zone network
(OZCAR), where multidisciplinary studies, including hydrological, geochemical and geological
investigations, have been performed since 1986 ("Observatoire Hydrogéochimique de
l'Environnement", OHGE; http://ohge.unistra.fr; El Gh'Mari, 1995; Fichter et al., 1998; Viville
et al., 2012; Gangloff et al., 2014; 2016; Prunier et al., 2015; Pan et al., 2015; Ackerer et al.,
2016; 2018; Beaulieu et al., 2016; Chabaux et al., 2017; 2019; Schmitt et al., 2017; 2018; Daval
et al., 2018; see also Pierret et al., 2018 for an updated overview of the Strengbach
watershed).

## 2- Site presentation and data acquisition

The Strengbach catchment is a small watershed (0.8 km²) located in the Vosges Mountains of northeastern France at altitudes between 883 and 1147 m. Its hydroclimatic characteristics can be found in Viville et al. (2012) or in Pierret et al. (2018). It is marked by a mountainous oceanic climate, with an annual mean temperature of 6 °C and an annual mean rainfall of approximately 1400 mm, with 15 to 20% falling as snow during two to four months per year. The snow cover period is quite variable from year to year, and may not be continuous over the entire winter. The annual mean evapotranspiration is of approximately 600 mm, and the annual mean infiltration (no significant surface runoff observed) of approximately 800 mm (Viville et al., 2012). The watershed is currently covered by a beech and spruce forest. The bedrock is a base-poor Hercynian granite covered by a 50 to 100 cm-thick acidic and coarse-in-texture soil. The granitic bedrock was fractured and hydrothermally altered, with a stronger degree of hydrothermal overprinting in the northern than the southern part of the catchment (Fichter et al., 1998). The granite was also affected by surface weathering processes during the Quaternary (Ackerer et al., 2016). The porous and uppermost part of the granitic basement constitutes an aquifer from 2 to approximately 8 m thickness. In the Strengbach watershed, the major floods and high-flow events usually occur during snowmelt periods at the end of the winter season or in the early spring. By contrast, the low-flow periods commonly happen at the end of the summer or during the autumn. Several springs are captured for drinkable water supply directly in the subsurface by small collectors (figure 1). The watershed has been equipped with several piezometers and boreholes since 2012, those being located along the slopes on both sides of the watershed (figure 1 in Chabaux et al., 2017).

Spring waters have been regularly collected and analyzed since 2005, with monthly sampling
supplemented by a few specific campaigns to cover the complete range of water discharges
in the watershed. Piezometer waters have been collected only during specific sampling
campaigns over the period 2012-2015, and, as for the spring waters, these sampling
campaigns cover different hydrological conditions from wet to dry periods. The soil solutions
were collected with a monthly frequency on the southern slope at a beech site (named HP)
and to the north at a spruce site (named VP; figure 1; more details in Prunier et al., 2015). For
all the collected waters, the concentrations of the major dissolved species and the pH were
determined by following the analytical techniques used at LHyGeS (Strasbourg, France) and
detailed in Gangloff et al. (2014) and Prunier et al. (2015). Discharges of water from the springs
were measured during the sampling campaigns, as were the water levels within the
piezometers.
The mineralogy and the porosity of the bedrock have been studied in detail in previous studies
(El Gh'Mari, 1995; Fichter et al., 1998). On the southern part of the catchment, the weakly
hydrothermally altered granite (named HPT, figure 1) is mainly composed of quartz (35%),
albite (31%), K-feldspar (22%) and biotite (6%). It also contains small amounts of muscovite
(3%), anorthite (2%), apatite (0.5%) and clay minerals (0.5%). On the northern part of the
catchment, the lithology is more variable, with the presence of gneiss close to the crest lines
and the occurrence of hydrothermally altered granite on the rest of the slopes (El Gh'Mari,
1995, figure 1).
The hydrological, geochemical and petrological data obtained from these field investigations
are the basis of the modeling exercise presented in this study. More precisely, this study is
based on hydrogeochemical data from 2005 to 2015 for waters from four springs of the
southern part (CS1, CS2, CS3 and CS4) and one spring of the northern part (RH3) of the
watershed (figure 1). Hydrogeochemical data obtained over the period 2012-2015 for two
piezometers (PZ3, PZ5) of the southern part of the watershed are also studied (figure 1). The
overall hydrogeochemical database is available as supplementary tables (tables EA1 to EA9).
The specific chemical data from spring and piezometer waters modeled in this study are
reported in table 1.
**3- Modeling methods**
The modeling developments presented in this study represent a new step in the efforts
undertaken to constrain the mechanisms controlling the geochemical composition of surface
waters and to understand their spatial and temporal variations at the scale of headwater
mountainous catchments (Schaffhauser et al., 2014; Lucas et al., 2017; Ackerer et al., 2018).
The main innovation of this present work is to couple a spatially distributed and low-
dimensional hydrological model with a reactive transport code to constrain the
spatiotemporal variability of chemical composition of waters. To the best of our knowledge,
this is the first time that such a coupling between low-dimensional hydrological and
hydrogeochemical modeling approaches has been attempted in this way at the watershed
scale.
**3-1 Hydrological modeling**
To assess the water flows in the watershed, several simulations were performed with the
hydrological code NIHM (Normally Integrated Hydrological Model; Pan et al., 2015; Weill et
al., 2017; Jeannot et al., 2018). This code is a coupled stream, overland, and low-dimensional
(depth-integrated) subsurface flow model developed at LHyGeS and already tested in the
Strengbach watershed (Pan et al., 2015). The stream and overland flows are described by a
diffusive-wave equation, and the subsurface flow is handled through an integration (in a
direction normal to bedrock) of the unsaturated-saturated flow equation from the bedrock to
the soil surface (Weill et al., 2017). The exchanges of water between the surface and
subsurface flows are addressed via a first-order exchange coefficient involving the thickness
and the hydraulic conductivity of an interface layer (e.g., the riverbed, for interactions
between surface routing and subsurface compartments), and the hydraulic head differences
between the compartments (Jeannot et al., 2018).
Regarding the hydrological simulations, NIHM was used with only its stream flow and
subsurface flow compartments activated, the Strengbach catchment having never evidenced
diffuse two-dimensional surface runoff or subsurface exfiltration over large areas. In addition,
and because of the steep slopes, the stream flow process revealed almost insensitive to the
roughness and Manning's parameters of the riverbed, which were set to usual values for very
small streams of mountainous landscapes. By contrast, the parameters of the subsurface were
adjusted in NIHM through a calibration-validation process. Several zones of heterogeneity
(figure 2) were defined based on field observations (Ackerer et al., 2016; Chabaux et al., 2017).
In each of these zones, the saturated hydraulic conductivity, the depth of substratum, and the
porosity, were set to uniform values. Other parameters (the residual water content, the
specific storage, the Van Genuchten coefficients n and α, and the saturated hydraulic
conductivity of the interface layer between the groundwater compartment and the surface
compartment) were set to uniform values over the whole catchment (table 2). The thickness
of the aquifer that was used for the simulations varied from 2 m near the main crests to up to
8 m in the middle of the watershed (figure 2), in agreement with the data obtained during the
recent geological investigations and drilling campaigns undertaken at the catchment (Ackerer
et al., 2016; Chabaux et al., 2017). The uniform precipitations over space applied at the surface

of the catchment are drawn from data of the pluviometric station located at the highest

elevation of the watershed (site PA, figure 1). The hydrological model NIHM was then run over

a first time period (years 1996-1997). By a Monte-Carlo approach, the parameters were

"randomly" sought to improve the fitting between the observed and simulated flow rates at

the outlet of the catchment (table 2). The fit was quantified by the root mean square error

(RMSE) and the Kling-Gupta efficiency coefficient (KGE; Gupta et al., 2009), applied to the

outlet flow rate of the stream, which is the only reliable and always available hydrological

variable monitored in the system.

Once the best fit was obtained, the model was then run over another time period (2010-2015),

but without changing the parameters anymore, and the quality of the fit was re-assessed for

this new time-period with the KGE and RMSE. Figure 2 shows the result for the 2010-2015

time period. After the water discharges were correctly reproduced at the outlet, a

backtracking approach was used to identify which subsurface flow lines reach the sampled

sites. To back track the water particles, the velocity fields calculated by the NIHM model were

inverted in their direction, and the locations of the backtracked particles were saved at each

time-step. A daily time-step was used for the backtracking, as a compromise between

computational efforts and a refined description of the transient velocity fields. A schematic

representation of the backtracking approach is given in figure 3. This methodology allows for

constraining the flow lines that bring waters for a given time and at a given position on the

catchment. This information is of major interest to determine the origin of the spring and

piezometer waters. It is shown at the catchment scale, that flows are mainly driven by gravity

in association with the steep slopes of the watershed, the latter being almost evenly drained

over its whole surface area (figure 4). For each water sampling area, ten flow lines that bring

water to the location of interest were determined (figure 4), together with a few features of
the flow lines, including: local velocities, mean velocities, and length of the flow paths.
It is worth noting that NIHM is a depth-integrated model for its subsurface compartment
where flow is simulated over a 2D-mesh and under the assumption of an instantaneous
hydrostatic equilibrium in the direction perpendicular to the substratum. Therefore, times
calculated along the backtracked streamlines correspond to a date, x days before arrival, at
which a water particle entered the subsurface or passed at a given location along the
streamline. Streamlines calculated via backtracking and reaching sampling sites only consider
flow in the subsurface compartment and are conditional to an arrival date at a prescribed
location. As backtracked streamlines are not associated with mean water flux values, the
transit time distributions drawn from streamline calculations are only an approximation of the
actual transit time distributions.
It should also be noted that, knowing the water head at a given location, the assumption of
an instantaneous hydrostatic equilibrium over the direction perpendicular to the substratum
directly renders the associated water pressure over the whole aquifer along that direction.
Then, since the water pressure, saturated hydraulic conductivity, porosity, residual water
content, and Van Genuchten coefficients are known, the Van Genuchten equation can be
integrated numerically, which gives to NIHM the possibility to calculate local depth-integrated
hydraulic conductivities over the direction perpendicular to the substratum.
With a conditioning of NIHM limited to the reproduction of the stream flow rates at its outlet,
it can be questioned on the reliability of the solution, equifinalities in model outputs being
usually all the more present that few data are available to condition the model. The point is
that there is no other reliable information on flow patterns, and for example, the few
boreholes available (mainly drilled for rock core sampling) are deep enough to intercept a few
fractures in the bedrock (under the bottom of the aquifer simulated by NIHM). This renders
the water levels monitored in these open boreholes unable to reflect hydraulic pressure heads
in the active shallow porous aquifer of the catchment. Nevertheless, the steep slopes of the
catchment are the main feature conditioning water velocities, thus rendering transit times
(the variable of interest for a geochemical study) very stable over time, irrespective of hydro-
meteorological conditions and current head pressure in the system. After the present study
was completed, NIHM was employed at the Strengbach to simulate water content
distributions with the aim to mimic data from magnetic resonance sounding (Weill et al.,
2019). The model was slightly improved in terms of storage and its variability over space, but
the modeled distribution of flow paths, their variability, and the associated transit time
distributions remained unchanged.
**3-2 Hydrogeochemical modeling**
The simulations of the water chemical composition along the flow lines were performed with
the hydrogeochemical KIRMAT code (KInectic of Reaction and MAss Transport; Gérard et al.,
1998; Lucas et al., 2010; Ngo et al., 2014; Lucas et al., 2017). KIRMAT is a thermokinetic model
derived from the Transition State Theory (TST, Eyring, 1935; Murphy and Helgeson, 1987) that
simultaneously solves the equations describing geochemical reactions and transport mass
balance in a 1D-porous medium. The mass transport includes the effects of one-dimensional
convection, diffusion and kinematic dispersion. Chemical reactions account for the dissolution
of primary minerals and oxido-reduction reactions, in addition to the formation of secondary
minerals and clay minerals. KIRMAT includes the oxido-reduction processes of iron (Fe), sulfur
(S) and other important species for the corrosion of iron (Ngo et al., 2014). Oxido-reduction
reactions are handled through Nerst equations (Gerard et al., 1998; Ngo et al., 2014). The
calculation of the dissolution rates of primary minerals is based on the TST and on a kinetic
law (equation 1 in Ackerer et al., 2018, equation 1 in Ngo et al.,2014). Thermodynamic and
kinetic data for the primary minerals are available in supplementary materials (supplementary
tables EA10, EA11 and EA12).
The clay fraction is defined as a solid solution made up of a combination of pure clay end-
members. The clay end-members are defined on the basis of X-ray diffraction analyses of clay
minerals present in bedrock samples collected in the field (Fichter et al., 1998; Ackerer et al.,
2016; 2018). They consist of K-Illites, Mg-Illites, Ca-Illites, Montmorillonites, Na-
Montmorillonites, K-Montmorillonites, Ca-Montmorillonites and Mg-Montmorillonites
(supplementary material table EA13). During the hydrogeochemical simulations, the clay solid
solution is precipitated at thermodynamic equilibrium and precipitation is not described by a
kinetic law. The amount of a given clay mineral precipitated at any step of the simulated
reaction is calculated to maintain the chemical equilibrium from the moment it is reached in
the geochemical reaction. The amount of clay precipitated depends on the solubility product
(K) of the clay end members (Tardy and Fritz, 1981). This multicomponent solid solution
reproduces the impurity of the clay minerals formed during low-temperature water-rock
interactions (Tardy and Fritz, 1981), and its composition varies over time, depending on the
evolution of the water chemistry and the bedrock mineralogy (Ackerer et al., 2018). For the
secondary minerals other than clay minerals, the precipitation rates are derived from TST and
described by a kinetic law (equation 2 in Ngo et al., 2014). Precipitation of typical secondary
minerals such as carbonates, hematite or amorphous silica was tested, but these minerals
were not formed given the saturation states calculated in the geochemical modeling
(supplementary table EA14). Secondary mineral precipitation is therefore controlled by clay
mineral formation.
The KIRMAT code also includes feedback effects between mineral mass budgets, reactive
surfaces, and the evolution of bedrock porosity (Ngo et al., 2014). The reactive surfaces of the
primary minerals were calculated by assuming a simple spherical geometry for all the
minerals, and the mean size of the minerals was estimated from the observation of thin
sections from bedrock samples. During simulations, clay mineral precipitation and the
evolution of the reactive surfaces of primary minerals are tracked together with chemical
processes and water chemical composition. Given the short time scales reported by the
hydrological simulations (monthly timescale), changes in the reactive surfaces of primary
minerals over the simulation time were negligible. The KIRMAT code has already been applied
in geochemical modeling of alluvial subsurface waters (Lucas et al., 2010) and surface waters
(Lucas et al., 2017; Ackerer et al., 2018).
For this study, the modeling strategy is adapted from Ackerer et al. (2018) to consider the new
transit time constrains provided by the hydrological code NIHM. To capture the chemical
composition of the spring and the piezometer waters, numerical simulations were performed
along the subsurface streamlines that were determined through the backtracking approach.
A sketch of the hydrogeochemical modeling strategy is provided in figure 5. For each
streamline, several KIRMAT simulations were performed with different starting positions
along the active part of the line. The starting positions represent the locations at which the
soil solutions percolate through the subsurface shallow aquifer. These starting positions are
spaced with a constant lag distance of 1 m along the subsurface streamlines, which results in
a sub-continuous percolation of solutions along the whole length of the lines. The deepest soil
solutions collected to the south at the beech site (HP) and to the north at the spruce site (VP)
were considered representative of the soil solutions for the southern and northern slopes of
the catchment, respectively. The data of soil solution chemistry used in this study are available
in Prunier et al. (2015) and in supplementary tables (tables EA6 and EA7). These soil solutions
integrate the surface processes occurring before water percolation into the weathered
bedrock (regolith). Because the soil solutions can be injected into the aquifer at various times,
the temporal variability of the soil solution chemistry and its impact on the water-rock
interactions along the flow paths are accounted for in the modeling approach.
Data related to the regolith properties, such as the mineralogical compositions, the mineral
reactive surfaces and the thermodynamic and kinetic constants are given in Ackerer et al.
(2018) and in supplementary tables (tables EA10 to EA14). Mineral phases are assumed
homogeneously distributed over the regolith layer. By following this strategy, the simulations
that consider soil solutions percolating at the upper part of the catchment reflect the chemical
evolution of waters with long path lengths and long transit times within the aquifer. By
contrast, shorter path lengths and shorter transit times are associated with the percolation of
soil solutions that occurs in the vicinity of the sampling locations (figure 5). Because the springs
or the piezometers collect waters from different origins and with various transit times,
integration along each water flow line was performed. The aim of the integration is to
determine the mean chemical composition resulting from the mixing of the waters
characterized by variable transit times (figure 5). The integrated chemical composition of the
waters provided by a given flow line is calculated by taking the arithmetic mean of the solute
concentrations calculated by the succession of the KIRMAT simulations along the flow line
(figure 5). This arithmetic mean reflects a simple full mixing of uniform water fluxes along a
stream line irrespective of the short or long transit times. In other words, the geochemical

simulations are based on the hypothesis of spatially homogenous water-rock interactions along the flow lines. The soil solutions are assumed to percolate uniformly within the aquifer and are then conveyed along the slopes by uniformly distributed masses of water until reaching the sampling locations. When needed, the eventual calculation of water chemistry exiting several stream lines reaching a sampling location accounts for the spreading associated with various flow paths, spatial variability of water velocities and related travel times.

**4- Hydrological modeling results**

**4-1 Spatial variability of the flow lines**

The results provided by the hydrological code NIHM show that to the first order, the Strengbach catchment is well drained and that the topography exerts an important control on the flow line distribution (figure 4). Along the hillsides presenting linear or slightly convex slopes, the water flow lines show simple characteristics. The flow paths are nearly parallel, and the water velocities are similar along the different flow lines on this type of hillside. The water velocities tend to increase when moving downstream, with slower velocities near the main crests and higher velocities on the steepest parts of the hillsides. The waters collected along this type of hillside are therefore characterized by small variability of transit times. This is the case for the CS1, CS3 and RH3 spring waters located on the southern and northern parts of the catchment (figure 4). This is also the case for the piezometers PZ3 and PZ5 in the southern part of the watershed (figure 4). For the sites located on linear or slightly convex slopes (CS1, CS3, RH3, PZ3 and PZ5), all the characteristics of the different flow lines that feed each site are therefore comparable for a given site and for a given date.

By contrast, in the vicinity of the valley and in the topographic depressions, the hydrological modeling indicates that the flow line characteristics are more variable. Because flow lines

coming from different hill-sides can feed a topographic depression, mixing of different flow
lines with variable flow paths and contrasted water velocities can occur at these locations.
The waters collected in valleys or in topographic depressions are therefore characterized by a
higher variability of transit times. This is the case for the CS2 and CS4 springs, which are
located in a depression, in the axe of the small valley, and surrounded by slopes with various
orientations, and a complex flow line distribution (figure 4). For these two springs, the
characteristics of the different flow lines can be different for a given date.
**4-2 Temporal variability of the flow lines**
Hydrological modeling under general transient conditions can render the evolution over time
of water flows in the watershed but also of other hydraulic variables. As an example, after an
important rainfall event (30/03/2010 in figure 6), snapshots of the integrated hydraulic
conductivity (modeled via the Van Genuchten formulation) in the subsurface and simulated
by NIHM at the scale of the mesh size show increasing values with decreasing elevation in the
watershed. The same observation holds for conductivities during drought periods (see
29/11/2011, in figure 6). Provided that the hydraulic head gradient is largely dominated by
the topography and therefore almost constant over time (figure 6), the water velocities are
increasing along the flow lines from crests to valleys, irrespective of the wet versus dry
hydrological periods. However, it is noticeable that wet periods are favorable to a large
extension in the valleys of high values of depth averaged hydraulic conductivity indicating that
the aquifer is locally almost completely saturated from bottom to top (e.g., values of $6.5 \times 10^{-5}$
$ms^{-1}$ in figure 6 for a saturated bound at $8 \ 10^{-5} \ ms^{-1}$).
For the CS1 spring, the mean flow velocities along the flow lines vary from approximately 1
m/day to 7 m/day between the severe drought of 29/11/2011 and the strong flood of
30/03/2010 (figures 7A and 7B). These events correspond to the annual minimum and
maximum flow rates at the outlet of the Strengbach watershed. For the same dates, the mean
velocities vary from 2 – 12 m/day, 1 – 4 m/day and 1 – 9 m/day for the springs CS2, CS3 and
CS4, respectively. The variations from drought to flood are very similar for the piezometer
waters, with velocities in the ranges 2 – 10 m/day and 2 – 12 m/day for the PZ3 and PZ5
piezometers, respectively. The RH3 spring located on a steeper part of the northern slopes
exhibits flow velocity variations from 5 to 20 m/day from dry to flood conditions.
In addition to the flow velocity variations, the hydrological simulations also reveal variability
in the lengths of the active parts of the flow lines. For illustration, the active parts of the flow
lines are reduced from 160 m to 110 m from the flood to the drought events for the CS1 spring
(figures 7A and 7B). Such variability is triggered by the particular seasonal variations of the
hydraulic conductivities within the catchment. After important precipitations, high water
content and large integrated hydraulic conductivities (sometimes up to the saturated bound)
are simulated in the vicinity of the crests and all along the small valley of the catchment (figure
6). During periods of drought, the simulations indicate a strong decrease of hydraulic
conductivities close to the main crests and much smaller variations at mid-slopes (figure 6).
The crests rapidly dry out, whereas the areas at mid-slopes still supply some water to the
stream network. These contrasting hydrological behaviors result from the differences in
aquifer thickness and water storage between the crests and the other parts of the catchment
(figure 2). Thin aquifer, flow divergence and absence of feeding areas prevent large water
storage on the crests, in opposition to mid-slope parts with much thicker aquifers and the
presence of feeding areas upstream. This particular pattern simulated for the hydraulic
conductivities implies that the active parts of the flow lines extend up the main crests during
important floods, whereas they are limited to mid-slopes after a long dry period.
The consequence of this hydrological functioning is to moderate the seasonal variations of the
transit times of waters, as the active lengths of flow lines vary simultaneously with water flow
rates. Calculations indicate that for the spring and piezometer waters collected in this study,
the mean transit times of waters only vary from approximately 1.75 to 4 months between the
strongest flood and the driest conditions. Notably, these short subsurface water transit times
are explained by the small size of the catchment and the steep slopes.
**5- Hydrogeochemical modeling results**
**5-1 CS1 and CS3 springs (southern slope)**
The CS1 and CS3 springs emerge on the same slope and drain the same rocks. Their
hydrological behavior is also very similar in terms of flow lines and water transit times. The
interesting consequence of the simple flow line distribution for these springs is that a single
flow line can be considered as representative of all the flow lines that are feeding the spring,
irrespective of the hydrological conditions. Hydrogeochemical simulations were performed
along a single flow line for different hydrological periods using the methodology illustrated in
figure 5. The case of CS1 spring is used below to highlight the main results obtained from this
approach. For the strong flood of 30/03/2010, the KIRMAT simulations modeling the waters
coming from the vicinity of the spring and characterized by short transit times produced too
much diluted solutions, whereas the waters coming from the main crests were too much
concentrated to reproduce the spring water chemical composition. However, after an
integration of all the waters arriving at CS1 with the different transit times employed for the
simulation, the resulting geochemical composition correctly reproduces the chemical
composition of CS1 spring water at this date ($H_4SiO_4$, $Na^+$, $K^+$, $Mg^{2+}$, and $Ca^{2+}$ concentrations,
figure 7D). A similar conclusion is obtained for the important drought of 29/11/2011. Again,
geochemical integration of all the waters arriving at CS1 along a water line but with different
transit times correctly reproduces the chemical composition of the CS1 spring waters collected
on this date (figure 7C). This comment applies regardless of the time period considered.
The coupled hydrological and hydrogeochemical approach has been applied for the CS1 spring
for 6 dates covering the whole range of the water discharges of the spring (table 1). The
modeling results capture the seasonal variations of the water chemical composition of the CS1
spring over the whole range of observed flow rates at CS1 (figure 8). Simulations especially
reproduce the 20-30% variation in $H_4SiO_4$ concentrations (figure 8A), the 10-20% variation in
$Na^+$ concentrations (figure 8C), and the relatively stability of the $K^+$, $Mg^{2+}$ and pH of the CS1
waters (figure 8E, 8F and 8D). The response of each chemical element to a change in water
discharge is related to the initial soil solution concentration, the nature of primary minerals
controlling its budget and the degree of its incorporation into clay minerals. Specific
concentration-mean transit time relations (C-MTT relations) explain why the response of
solute concentrations to hydrological changes (C-Q relations) is different for each element
(figure 9). Similar results are obtained for the CS3 spring (figure EA1), showing, as for the CS1
spring, that the model correctly simulates the water chemical composition of the CS3 spring.
Because the lengths of the flow lines vary over time, the patterns of dissolution rates for
primary minerals and precipitated amount of clay minerals are mainly controlled by the spatial
and temporal variability of the flow lines. During wet conditions, the upper parts of the
catchment are the areas of maximal dissolution rates of primary minerals and of maximal
formation of clay minerals in the regolith. During dry conditions, the dissolution and
precipitation are maximal at mid-slopes, as the upper parts of the catchment are simply dry.
**5-2 PZ3 and PZ5 piezometers (southern slope)**
The two piezometers PZ3 and PZ5 are located on the southern part of the catchment, and
their waters drain a granitic bedrock similar to that drained by the CS sources. As for the CS1
and CS3 springs, the NIHM modeling results show that the flow lines arriving at the PZ3
piezometer are characterized by a relatively simple distribution (figure 4). For the PZ5
piezometer located downstream, the flow lines cover a larger area on the slope, especially
during droughts (figure 4). However, for a given date, all the flow lines show similar velocities,
with particularly fast flow in the lower portion of the hillslope. These results imply that, as for
the CS1 and CS3 springs, the hydrogeochemical simulations of PZ3 and PZ5 piezometer waters
can be performed by relying upon a single flow line representative of all the waters collected
by the piezometers on a given date. The geochemical integration is able to reproduce the
chemical composition of the waters of the two piezometers, as illustrated in figure 10 for the
flood of the 05/05/2015 and in figure EA2 for the dry conditions of 10/11/2015. Together,
these modeling results show that the flow along linear or slightly convex slopes on the
southern part of the catchment allows to correctly capture the water chemistry of each
sampling site with a straightforward integration along a single and representative flow line.
**5-3 The CS2 and CS4 springs (in the valley axe)**
CS2 and CS4 spring waters drain the same granitic bedrock as the CS1 and CS3 waters, but are
located in the direction of the small valley of the Strengbach stream and surrounded by slopes
of various orientations and inclinations (figure 4). Consequently, the distribution of the flow
lines is much more scattered than for the CS1 and CS3 springs. For the CS2 spring, and for all
the hydrological conditions, two different groups of flow lines have been determined by the
backtracking approach: a northern group characterized by relatively slow velocities and a
southern group with higher velocities (figure 4 and figures 11A, 11B). This scattered
distribution of the flow lines implies that a single specific flow line cannot be representative
of all the waters collected by the spring. The flow lines calculated using the NIHM model allow
for constraining the trajectories of the waters within the watershed; however, the simulations
performed in this study cannot provide the mass fluxes of water carried by each flow line.
Consequently, a straightforward calculation of the chemistry of the CS2 spring, such as
depicted above for CS1, is not applicable because the mixing proportions between the
different flow lines are unknown.
Alternatively, it is possible to determine the concentrations in the waters carried by the
slowest and the fastest flow lines that are feeding the spring and to compare the results with
the observed chemistry of the spring water. The results indicate that for all the hydrological
conditions, the concentrations calculated from the geochemical integration along the slowest
and the fastest flow lines are able to correctly frame the chemical composition in terms of
$H_4SiO_4$, $Na^+$, $K^+$, $Mg^{2+}$, and $Ca^{2+}$ of the CS2 spring waters (results are reported for $H_4SiO_4$ and
$Na^+$ in figures 11C and 11D). The modeling results for CS2 also suggest that the contributions
of the slow and fast flow lines are comparable over most of the hydrological conditions, as the
observed concentrations are in general at the midpoint between the min (i.e., fast) and max
(i.e., slow) boundaries (figures 11C and 11D). It is only for the important droughts that the
spring chemistry seems to be mainly controlled by the southern and faster group of flow lines.
Further works to precisely estimate the mass fluxes of water carried by each flow line are
necessary to model the chemistry of the CS2 spring water with a weighted mixing calculation.
The same conclusions apply to the CS4 spring located close to CS2.
**5-4 The RH3 spring (northern slope)**
The RH3 spring is located on the northern part of the catchment (figure 4), where steep slopes
imply fast water velocities and subparallel flow lines. However, if the distribution of the flow
lines on the RH3 hillside is simple (as for the CS1 and CS3 springs) the precise lithological
nature of the bedrock drained by the RH3 waters is more difficult to constrain (Ackerer et al.,
2018). Unlike the southern slope, the bedrock of the northern part of the catchment reveals
a complex lithology, with gneiss outcropping in the upper part of the slope and granite of
variable degree of hydrothermal overprinting in the intermediate and lower parts. These
lithological variations can explain the differences in chemical composition between the RH3
spring waters and the waters of the southern part of the catchment: the RH3 spring waters
are characterized by systematically higher concentrations of $K^+$ and $Mg^{2+}$ cations but show
similar concentrations for the other major elements (Ackerer et al., 2018; Pierret et al., 2018).
The vertical extension of the gneiss and the spatial variability of the hydrothermal overprinting
along the northern slopes are not well known, with the consequence that a straightforward
modeling of water chemistry as done for CS1 is not possible for RH3.
Alternatively, simulations of two extreme cases can be performed by assuming that the flow
lines only run, either on gneiss or on hydrothermally altered granite. When only considering
the hydrothermally altered granite (VS facies), the simulated concentrations of $H_4SiO_4$ and $Na^+$
are close to the measured ones. Nevertheless, the concentrations of $K^+$ and especially $Mg^{2+}$
are clearly underestimated (figure 12B). In the case of the flow lines only running on gneiss
(GN facies), the simulated concentrations of $H_4SiO_4$ and $Na^+$ also match the data. However,
due to the higher abundance of biotite in the gneiss, the simulated concentrations of $K^+$ and
$Mg^{2+}$ are higher than the measured ones (figure 12A). At this stage, it is therefore reasonable
to propose that the chemical composition of the RH3 spring waters reflects mixing of the two
lithological influences. By assuming a geochemical conservative mixing, which is likely a too
simplistic scenario, the results would indicate that the flow lines portions running on gneiss
and on hydrothermally altered granite count for approximately 40-50% and 50-60% of the
total water path length, respectively.
Further works to estimate the location of the contact between gneiss and granite are required
for more realistic modeling and hence a deeper interpretation of the chemical composition of
the RH3 spring waters. In any case, the important point to stress here based on the above
simulations is that the complex lithology and bedrock heterogeneity mainly impact the $K^+$ and
the $Mg^{2+}$ budget of the RH3 waters, but not or only slightly the $H_4SiO_4$ and $Na^+$ concentrations,
which control the main part of global weathering fluxes carried by the Strengbach spring
waters. These results readily explain why although the RH3 spring waters exhibits higher $Mg^{2+}$
and $K^+$ concentrations than the other CS springs, they carry relatively similar global weathering
fluxes (Viville et al., 2012; Ackerer et al., 2018).
**6- Discussion**
The coupling of the NIHM and KIRMAT codes allows for building a better modeling scheme to
those commonly used in previous studies regarding the hydrogeochemical modeling of
surface waters at the watershed scale. In such previous works, the geochemical simulations
were performed mainly along a single 1D flow line, only characterized by homogeneous mean
hydrological properties (Goddéris et al., 2006; Maher, 2011; Moore et al., 2012; Lucas et al.,
2017; Ackerer et al., 2018). In a previous study on the Strengbach watershed (Ackerer et al.,
2018), the soil solutions were also assumed to percolate in the bedrock only at a single starting
point of the flow lines. Although these previous approaches were useful for determining the
long-term evolution of regolith profiles and/or the mean chemistry of waters at the pluri-
annual scale, they cannot be used to discuss the seasonal variations of the water chemical
composition. The NIHM-KIRMAT coupling approach makes this possible, as it provides the
spatial distribution of the flow lines at the watershed scale and their variations over time.
Furthermore, the proposed modeling approach also integrates a soil solution percolation
scheme with inlets uniformly distributed along the slope, which is more realistic than a
scheme assuming that each sampled site is fed by a single flow line carrying waters with a
unique transit time. The good agreement between modeling results and observations over a
large panel of hydrological conditions gives strength to the conclusions and implications that
can be drawn regarding the hydrogeochemical functioning of this headwater catchment.
**6-1 Choices of the reactive surfaces and the kinetic constants**
For the geochemical simulations performed in this study, the kinetic constants that were used
to describe the dissolution reactions of the primary minerals are standard constants
determined through laboratory experiments (supplementary table EA12). The reactive
surfaces of the primary minerals were calculated by assuming a simple spherical geometry for
all the minerals (supplementary table EA10). Over the last years, several studies have
suggested that the kinetic constants determined through laboratory experiments
overestimated the rates of the dissolution reactions in natural environments (White and
Brantley, 2003; Zhu, 2005; Moore et al., 2012; Fischer et al., 2014). The origin of this
laboratory-field discrepancy is still a matter of debate (Fischer et al., 2014). Different
processes have been proposed to explain the gap between laboratory and field estimates,
such as the crystallographic anisotropy (Pollet-Villard et al., 2016), progressive occlusion of
the primary minerals by clays (White and Brantley, 2003), or the formation of passivation
layers at the surfaces of the minerals (Wild et al., 2016, Daval et al., 2018). The difficulty to
reconcile field and laboratory estimates can also be related to the challenge of defining
relevant reactive surfaces at different space scales (Li et al., 2006; Navarre-Sitchler and
Brantley, 2007).
The present modeling work regarding the Strengbach catchment shows that the chemical
composition variability of the spring and piezometer waters is fully captured via geometric
reactive surfaces and standard kinetic constants, while respecting the water-rock interaction
times within the catchment. This result suggests that the mean rates of the weathering
reactions employed in this modeling work are realistic, which in turn implies that the modeling
approach developed in this study does not underline significant mismatches between field
and laboratory reaction rates. The calculated rates of the dissolution reactions depend on the
product between the kinetic constants of the reactions and the mineral reactive surfaces. In
the experimental studies performed for determining the kinetic constants of dissolution
reactions, the constants are usually determined by normalizing the experimental weathering
rates with the Brunauer-Emmett-Teller surfaces determined from experiments of gas
absorption (BET surfaces; Chou and Wollast, 1986; Lundstrom and Ohman, 1990; Acker and
Bricker, 1992; Amrhein and Suarez, 1992; Berger et al., 1994; Guidry and Mackenzie, 2003).
In table 3, the BET surfaces are compared with the geometric surfaces of the minerals involved
in the dissolution experiments, recalculated from the size ranges of the minerals. For most of
the minerals (apatite, quartz, albite, K-feldspar, and anorthite), the geometric surfaces are
within the same order of magnitude as the BET surfaces, even if often slightly lower (table 3).
However, as the BET surfaces are determined with fairly large uncertainties, especially for low
BET surfaces (up to ± 70%), and as they can be very different depending on the gas used (up
to 50% of difference between N2 or Kr absorption; Brantley and Mellott, 2000), the above
differences between the geometrical and the BET surfaces cannot be considered significant

for the majority of minerals used in the Strengbach simulations. A significant difference only appears for biotite, with the geometric surfaces one order of magnitude less than the BET surfaces (table 3). However, for biotite, due to its layered structure, it has been shown that approximately 80 – 90% of the surface area accessible by the gases used to estimate BET surfaces is not accessible for weathering reactions (Nagy, 1995).

The above considerations explain why for a granitic bedrock as found in the Strengbach catchment, the geometric surfaces are relevant to describe the surfaces of water-rock interactions at the space and time scales of this study. An immediate corollary is that the values of the standard kinetic constants (table EA12) are also appropriate to calculate reaction rates with mineral geometric surfaces in our modeling approach. This ability may be related to the fact that all the minerals that have been used in the dissolution experiments and in the kinetic studies were collected in the field (e.g., Acker and Bricker, 1992; Amrhein and Suarez, 1992). These minerals were likely affected by anisotropy, passivation layers, and any types of aging effects related to long-term water-rock interactions. Our results might therefore mean that the standard kinetic constants obtained in such experiments integrate the aging effects that have affected the reactivity of the primary minerals in natural environments. This would explain why it is possible to capture the full variability of the water chemistry in a headwater catchment with simple geometric reactive surfaces and standard kinetic constants.

At this stage, the results of our simulations strengthen the idea that the low surfaces calculated from the geometrical shapes of minerals provide good estimates of the reactive surfaces within this type of environment (Brantley and Mellott, 2000; Gautier et al., 2001; White and Brantley, 2003; Zhu, 2005; Li et al., 2017). They are certainly the values to be used for hydrogeochemical modeling such as that performed in this work, in addition to the use of

the experimental kinetic constants for mineral dissolution. These conclusions are certainly not
specific to the Strengbach catchment and could be applicable to many other headwater
granitic catchments.
**6-2 Implications for the acquisition of the water chemistry**
The results of the NIHM-KIRMAT hydrogeochemical modeling have strong implications
regarding the hydrogeochemical dynamic of the Strengbach watershed. This work reinforces
several hypotheses formulated by previous studies conducted in the Strengbach watershed
(Viville et al., 2012; Pierret et al., 2014; Pan et al., 2015; Chabaux et al., 2017; Weill et al., 2017;
Ackerer et al., 2018), but also brings new insights on the hydrogeochemical functioning of the
catchment. Firstly, the modeling results emphasize the importance of water transit times
within the watershed as a main feature controlling the chemical composition of subsurface
waters. Along all the slopes, the waters coming from the vicinity of the crests and
characterized by long transit times systematically render higher concentrations than the
waters with shorter pathways and transit times. When the hydrological conditions change
from wet to dry periods, the solute concentrations also tend to increase with the increase in
the mean transit time of waters. Our results show that for the spring and piezometer waters,
the spatial and temporal variations of their geochemical composition are fully explained by
the differences in water transit times (figure 13). Transit time variations between high and low
discharge periods explain the temporal variations of geochemical signatures within each site.
Various mean transit times of waters supplying the different sites explain the various chemical
compositions between the sites (figure 13). This key role of the water-rock interaction time is
in agreement with previous reactive-transport studies conducted in the Strengbach
watershed (Ackerer et al., 2018) and in other sites (e.g. Maher, 2010; Moore et al., 2012;
Lebedeva and Brantley; 2013).
This study also brings new constrains on the spatial distribution of the weathering processes.
For the modeling strategy employed, the chemical composition of the spring and piezometer
waters are calculated by integrating the chemical composition of waters introduced at
different starting locations along the active part of the flow lines (figure 5). The modeling
results show that through the geochemical integration, the concentrated waters coming from
the main crests are naturally counterbalanced by the diluted waters infiltrating close to the
sampling sites. The solute chemistry is acquired through reactions and weathering processes
that are spatially relatively homogenous along the flow lines of the watersheds. This spatial
homogeneity of the weathering processes helps us to understand why the chemical fluxes
carried by the Strengbach stream (Viville et al., 2012), the chemical fluxes from the Strengbach
spring waters (Ackerer et al., 2018) and the weathering fluxes locally determined along a
regolith profile sampled in the catchment (Ackerer et al., 2016), are all very similar.
The modeling also shows that the hydrogeochemical functioning of the watershed is properly
simulated by water circulations in the shallow subsurface, i.e., in a saprolitic aquifer. No
contribution of waters circulating in the deep fracture network of the granitic bedrock and
observed during the drilling campaigns (see Chabaux et al., 2017) is necessary. The deep-water
circulations are probably disconnected from the shallow subsurface network, as recently
suggested by geochemical studies conducted in the Strengbach watershed (Chabaux et al.,
2017; Pierret et al., 2018). This is also in agreement with recent hydrological modeling studies
arguing that the catchment behaves like a vertically thin but horizontally wide reservoir (Pan
et al., 2015; Weill et al., 2017). The modeling results also show that water in the shallow
aquifer flows along streamlines with fairly simple geometries. At the scale of the catchment
(figure 4), the geometry of the flow lines validates the hypothesis based on the geochemical
and Sr-U isotopic data that the spring waters of these mid-mountain basins are supplied by
waters from distinct flow paths without real interconnections (i.e., the Strengbach and
Ringelbach watersheds; Schaffhauser et al., 2014; Pierret et al., 2014). Flow paths are
therefore distinct along the slopes and occur within the shallow saprolitic aquifer but are not
controlled by deep fractures in the bedrock.
**6-3 Origins of general chemostatic behavior and of specific C-Q relations**
The hydrogeochemical monitoring of the spring, piezometer, and stream waters performed in
the Strengbach catchment clearly shows that this catchment has a general chemostatic
behavior (e.g., Viville et al., 2012; Ackerer et al., 2018). All the spring and the piezometer
waters have chemical concentrations impacted by changes in the hydrological conditions, but
the concentration variation ranges are by far narrower than variation ranges of water
discharges, which define the chemostatic behavior of a hydrological system. For waters
showing the largest concentration variations (spring CS1), there is a modest increase of
approximately 10-30% in the concentrations of $H_4SiO_4$ and Na+ from floods to drought events,
while the water discharges may vary by a factor of 15 (figure 8). This modest variability of the
solute concentrations over a wide range of water discharges is not specific to the Strengbach
catchment; it has been observed in several watersheds spanning different climates and
hydrological contexts (Godsey et al., 2009; Clow and Mast, 2010; Kim et al., 2017).
Different origins for the chemostatic behavior have been proposed, such as a modification of
the mineral reactive surfaces during changing hydrological conditions (Clow and Mast, 2010),
a small concentration difference between slow and fast moving waters (Kim et al., 2017), or
the fact of reaching an equilibrium concentration along the water pathway (Maher, 2010). The
coupled approach NIHM-KIRMAT renews the opportunity to discuss on the origin of the
chemostatic behavior in catchments. It is worth noting that the acquisition and the evolution
of the water chemistry can be simulated along flow lines that have been determined via timely
and spatially distributed hydrological modeling. The strength of this approach is to constrain
water transit times independently and before any geochemical simulation.
The results from the hydrological model show that the characteristics of the flow lines are
affected by the changes in the hydrological conditions (section 4.2). This hydrological
functioning implies a covariation between flow velocity and flow length over changing
hydrological conditions, with faster flows along longer paths during wet conditions and slower
flows along shorter paths during dry periods. This hydrological behavior attenuates the
variations of the water transit times over changing hydrological conditions. It also explains
why the mean transit times span much narrower variation ranges than the water discharges
at the collected springs. For example, the calculated mean transit times of waters for the CS1
spring vary from 1.75 to 3.13 months between the strongest flood and the driest period that
have been studied, whereas the water discharges vary from 1.523 L/s to 0.098 L/s (figure 8B).
Because the time of the water-rock interactions exerts a first-order control on the chemical
composition of waters, the weak variability of the mean transit times is directly responsible
for the relative stability of the chemical composition of waters within the catchment. A
seasonal expansion and contraction of the hydrological network was also recently highlighted
in Alpine headwater catchments (Van Meerveld et al., 2019).

In addition to this general chemostatic behavior, each chemical element has a specific response to a change in water transit time as exemplified in figure 9 where are given the concentration-mean transit time relations (C-MTT relations) for $H_4SiO_4$ and the major cations. In the relevant transit time window for the spring and piezometer waters (figure 9b), the C-MTT relations are linear and C-MTT slopes are significant for $H_4SiO_4$, modest for $Na^+$ and weak for $Mg^{2+}$ and $K^+$ concentrations. The modeling results indicate that the C-MTT slopes are controlled by the competition between primary mineral dissolution and element incorporation into clay minerals. When elemental fluxes from primary mineral dissolution to solution are much higher than fluxes from solution to clay minerals (e.g., $H_4SiO_4$), the element can accumulate in solution, resulting in a significant C-MTT slope. By contrast, when elemental fluxes from primary mineral dissolution to solution are only slightly higher than fluxes from solution to clay minerals (e.g., $K^+$), the element accumulates only slowly in solution, resulting in a weak C-MTT slope. Interestingly, when fitting power-laws along C-Q relations ($C=aQ^b$, in caption of figure 8), both 'a' coefficient controlling the height of the C-Q laws and 'b' coefficient controlling the curvature of the C-Q laws are sensitive to the C-MTT slopes (figure 9c and 9d). 'a' coefficient is positively corelated with C-MTT slopes while 'b' coefficient is negatively corelated. Solute species with significant C-MTT slopes are more chemodynamic and display higher mean annual concentrations ($H_4SiO_4$ , $b(H_4SiO_4)=-0.1$, $a(H_4SiO_4)=10^{-4}$), whereas species with weak C-MTT slopes show low mean annual concentrations and are nearly perfectly chemostatic ($a(Mg^{2+})=10^{-5}$, $b(Mg^{2+})=-0.016$, $a(K^+)=10^{-5}$, $b(K^+)=0$, figures 8, 9c and 9d). Our results show that a better knowledge of C-MTT relations is important to explain the contrasted C-Q shapes of chemical elements.

It is important to underline that the hydrological modeling with the NIHM code is performed independently and before any geochemical simulations with the KIRMAT code. The fact that

the flow rates are well reproduced for all the hydrological contexts between 2010 and 2015
supports that the water transit times inferred from the NIHM code are realistic. The fact that
the chemical composition of waters is well captured indicates that the combination of the
geochemical parameters used in KIRMAT code is able to generate realistic reaction rates, as
chemistry is well reproduced while respecting realistic water transit times. No modifications
of the reactive surfaces and of the dissolution kinetic constants were necessary to reproduce
the seasonal variability of the water chemistry. It is also important to emphasize that the
simulated chemical compositions of waters remain far from a state of chemical equilibrium
with respect to primary minerals. The calculated Gibbs free energy for the primary minerals
ranges from -120 to -100 kJ/mol for apatite, -90 to -80 kJ/mol for biotite and anorthite and -
30 to -20 kJ/mol for albite and K-feldspar. These far-from-equilibrium values for the Gibbs free
energy imply that the reaction rates calculated using hydrogeochemical codes such as
KIRMAT, which are based on the transient state theory (TST, Eyring, 1935; Murphy and
Helgeson, 1987), are realistic for most of the primary minerals in this type of hydrological
context. Regarding the simulations performed in this study, the relatively short residence
times of waters and the precipitation of clay minerals prevent reaching a state of chemical
equilibrium between waters and primary minerals at the watershed scale. A water transit time
around 8-12 years and a distance as long as 15-20 km would be necessary to reach a chemical
equilibrium between water and primary minerals (see Ackerer et al., 2018). This long
equilibrium length is explained by the precipitation and the dynamic behavior of clay minerals
removing ions from solution and retarding chemical equilibrium with respect to primary
minerals. Relying upon a clay solid solution is also appropriate to mimic the clay mineral
dynamic in this type of watershed, and a clay mineral assemblage precipitating at
thermodynamic equilibrium is able to generate reliable water chemistry (this study) and
realistic amount of clay minerals (mass fraction of clay minerals of 2-3 % in the regolith after
20 kyr of weathering, more detail in Ackerer et al., 2018).
Our results indicate that it is not necessary to mix in different proportions soil and deep waters
to generate chemostatic behavior, as proposed by Zhi et al. (2019). Chemostatic behavior can
be generated within a single regolith layer with a homogeneous mineralogy, if as
demonstrated, the transit time variability of shallow subsurface waters is dampened by
seasonal fluctuations of flow line properties. A large storage of primary minerals and
weathering product in the subsurface, as proposed in Musolff et al. (2015), is required but not
sufficient to generate chemostatic behavior. Chemostatic behavior also depends on the
covariation between flow velocities and flow lengths over changing hydrological conditions.
Chemostatic behavior is not explained by a modification of the reactive-surface of minerals in
the subsurface (i.e., Clow and Mast, 2010), or by an absence of chemical contrast between
slow and rapid flows (i.e., Kim et al., 2017). The precipitation of clay minerals is essential to
correctly capture the water chemistry in our study, but the dissolution or redissolution of clays
is not a key process to explain chemostatic behavior (i.e., Li et al., 2017). Our study clearly
supports the idea defended by Herndon et al. (2018) that a spatial and temporal variability in
flow paths is a key process to explain C-Q relations in this type of headwater catchment. Our
conclusions can most likely be extended to the other mountainous and relatively steep
watersheds of this type, in which water pathways and short transit times are mainly controlled
by gravity driven flow along slopes (Weill et al., 2019).
**7- Conclusion**
This study exemplifies the potential of coupling of low-dimensional and depth-integrated
hydrological modeling with hydrogeochemical modeling as a way to better understand
variability over time and space of the composition of surface and subsurface waters. The
independent estimation of the water transit times provided by hydrological simulations is a
clear added value to constrain the geochemical modeling approaches. Our study
demonstrates that the seasonal variability of hydraulic conductivities along the slopes is a key
process to understand the dynamic of flow lines and the changes of water transit times in the
watershed. The variations in flow lines distributions from drought to flood events result in a
modest seasonal variability of mean water transit times, which in turn explains the relative
stability of the solute concentrations in waters. Our results also show that a better knowledge
of the concentration-mean transit time relations (C-MTT relations) is an interesting new step
to understand the diversity of C-Q shapes for different chemical elements. The consistency
between measured and modeled concentrations while respecting the water-rock interaction
times provided by the hydrological simulations shows that it is possible to capture the
chemical composition of waters with simply determined reactive surfaces and standard
kinetic constants. The results of our simulations strengthen the idea that the low surfaces
calculated from the geometrical shapes of minerals are a good estimate of the reactive
surfaces in this type of granitic catchment, and certainly the values to be used for
hydrogeochemical modeling such as that performed in this work, in addition to the use of the
experimental kinetic constants for mineral dissolution.

**Acknowledgements**: This work and the Julien Ackerer's salary were financially supported by
the French ANR Program (Project CANTARE- Alsace) under grant agreement ANR-15-CE06-
0014. This work also benefited from fruitful discussions with D. Daval. The authors thank all
the reviewers for their constructive comments that improved the quality of the manuscript.

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

 **Figure and table captions**

 Figure 1: sampling locations within the Strengbach catchment. Blue stars represent springs,

 blue diamonds represent piezometers, and the blue circle represents the stream at the outlet

 of the watershed. Green circles represent soil solution locations, and black diamonds

 represent bedrock facies locations.

 Figure 2: on the left: calibrated field of thicknesses of the weathered material constituting the

 shallow unconfined aquifer at the Strengbach catchment used for the simulations by NIHM.

 The 1D surface draining network used in NIHM is represented by the black lines. The mesh for

 the groundwater compartment is represented by grey lines. On the right: fitting observed flow

 rates from the Strengbach stream at the outlet of the catchment with simulations of flow

 within the watershed (illustrated from 2010 to 2015). The subsurface compartment inherits

 from the aquifer thicknesses reported in the left panel, and the topography lets the natural

 outlet of the subsurface compartment being the surface draining network.

 Figure 3: principle of the method of backtracking used to determine flow lines that generate

 flow at the outlet of the Strengbach catchment. Particles are dispatched along the wet fraction

 of the 1D river network (only one is represented here at a position $a$ on 01/01/2010 at 23:59).

 NIHM generates an output heterogeneous velocity field at that date for the whole watershed,

 denoted $V_{01/01/2010}$. By using a velocity field of the same magnitude but opposite direction to

 the particle, the position of the particle is backtracked until 31/12/2009 23:59. Then, to further

 backtrack the trajectory of the particle, the velocity field is updated accordingly. The

 frequency of velocity field updates is set to one day.

Figure 4: at the top, flow lines of the subsurface that feed with water the surface draining
network on March 1$^{st}$, 2010 (on the left, high-flow period) and July 1$^{st}$, 2010 (on the right, low-
flow period). The color scale indicates that a water particle reaching the river at a given date
started its travel along the streamline or passed at a given location on the streamline x days
prior. The density of streamlines is associated with the flowing versus dry fraction of the river
network at a prescribed date. Below, flow lines of the subsurface that feed with water the
geochemical sampling sites on March 30$^{th}$, 2010 (on the left, flood event) and November 29$^{th}$,
2011 (right, drought event) according to NIHM simulations. For each sampling site, 10 particles
were dispatched in the direct neighborhood of the site and then backtracked to render 10
stream lines. The color scale for times is similar to that of the top plot.
Figure 5: conceptual scheme used in the modeling of the water chemistry. The soil solutions
are used as input solution. Cells represent the grid of the reactive-transport code KIRMAT. The
regolith is discretized into a 1D succession of cells along the active parts of the flow lines
determined by the NIHM hydrological model. The hydrogeochemical model KIRMAT evaluates
transport and geochemical processes within each cell. The integrated chemistry of sampled
waters is the arithmetic mean of solute concentrations with regularly distributed inlet points
along a stream line.
Figure 6: maps of piezometric gradient and depth-integrated hydraulic conductivity for the
Strengbach catchment, as simulated by NIHM, on 29/11/2011 (dry period) and 30/03/2010
(high flows period). The mean hydraulic conductivity is integrated normal to bedrock of the
aquifer and thus depends on the water saturation of the vadose zone and the location of the
water table.

Figure 7: simulation results for the CS1 spring for an important drought (29/11/2011) and a strong flood event (30/03/2010). At the top, active parts of the flow lines bringing the waters to the CS1 spring for the two sampling dates (7A and7B). Below, simulated chemical compositions of CS1 spring waters after integration along the flow lines and comparison with the initial soil solution and the spring chemistry data (7C and7D). Error bars show analytical uncertainties on measured concentrations and induced uncertainties in model results (the propagation in the KIRMAT simulations of analytical uncertainties from pH and chemical concentrations measured in the soil solutions).

Figure 8: simulation results for the CS1 spring over the whole range of the water discharges from the spring. Results are presented for $H_4SiO_4$, $Na^+$, $K^+$ and $Mg^{2+}$ concentrations (8A, 8C, 8E and 8F), pH (8D) and mean water transit time (8B). Red lines indicate simulated parameters after integration along the flow lines, and blue points show measured values collected between 2005 and 2015. Corresponding dates and data for the modeled samples are given in table 1. The overall geochemical database is available in supplementary table EA1. Error bars show analytical uncertainties on measured concentrations and induced uncertainties in model results (the propagation in the KIRMAT simulations of analytical uncertainties from pH and chemical concentrations measured in the soil solutions). Fitting a power law of type $C=a*Q^b$ along the C-Q relations gives the following parameters: $a(H_4SiO_4)=10^{-4}$, $b(H_4SiO_4)=-0.1$; $a(Na^+)=7\times10^{-5}$, $b(Na^+)=-0.053$; $a(Mg^{2+})=10^{-5}$, $b(Mg^{2+})=-0.016$; $a(K^+)=10^{-5}$, $b(K^+)=0$.

Figure 9: (9A) evolution of solute concentrations for $H_4SiO_4$, $Na^+$, $K^+$, $Mg^{2+}$ and $Ca^{2+}$ as a function of mean water transit time in the Strengbach watershed. Water transit times are between 1.75 and 4 months for all the springs and piezometers in this study. (9B) Focus on the transit time window (1.75-4 months) for the studied waters and equations linking mean water transit

times and concentrations for $H_4SiO_4$, $Na^+$, $K^+$, $Mg^{2+}$ and $Ca^{2+}$. Relations between transit times
and concentrations are linear within this window (9C) relations between 'b' coefficients
($C=a*Q^b$) and the concentration-transit time slopes for the chemical elements. (9D) relations
between 'a' coefficients ($C=a*Q^b$) and the concentration-transit time slopes for the chemical
elements. Elements with significant concentration-mean transit time slopes are slightly
chemodynamic (e.g. $H_4SiO_4$ and $Na^+$), while elements with low concentration-mean transit
time slopes are almost chemostatic in the watershed (e.g. $K^+$ and $Mg^{2+}$). $Ca^{2+}$ is not shown on
9C and 9D figures as this element is affected by a strong multi-annual concentration decrease
that prevents a meaningful C-Q power law analysis (Ackerer et al., 2018).
Figure 10: simulation results for the PZ3 and PZ5 piezometers for a flood event (05/05/2015).
At the top, active parts of the flow lines that bring waters to the two sampling sites (10A and
10B). Below, simulated chemical compositions of the piezometer waters after integration
along the flow lines and comparison with the initial soil solution and the water chemistry data
(10C and 10D). Error bars show analytical uncertainties on measured concentrations and
induced uncertainties in model results (the propagation in the KIRMAT simulations of
analytical uncertainties from pH and chemical concentrations measured in the soil solutions).
Figure 11: simulation results for the CS2 spring. At the top, active parts of the flow lines that
bring water to the CS2 spring for drought (29/11/2011) and flood (30/03/2010) events (11A
and 11B). The CS2 location results in more scattered flow lines than for CS1 spring. Below,
simulation results for the CS2 spring over the whole range of experienced discharges (11C and
11D). Blue lines indicate simulated parameters after integration along the slowest flow line,
yellow lines indicate simulated parameters after integration along the fastest flow line, and
blue points show measured values collected between 2005 and 2015 (data in table 1 and in
supplementary table EA2). Error bars show analytical uncertainties on measured
concentrations and induced uncertainties in model results (the propagation in the KIRMAT
simulations of analytical uncertainties from pH and chemical concentrations measured in the
soil solutions).
Figure 12: simulation results for the RH3 spring chemistry and for a flood event (30/03/2010).
Left, simulated concentrations by assuming flow lines running through gneiss (GN) only (12A).
Right, simulated concentrations by assuming flow lines running through hydrothermally
altered granite (VS) only (12B). Error bars show analytical uncertainties on measured
concentrations and induced uncertainties in model results (the propagation in the KIRMAT
simulations of analytical uncertainties from pH and chemical concentrations measured in the
soil solutions).
Figure 13: overview of the simulated flow lines in the subsurface that feed with water the
geochemical sampling sites CS1, PZ3, and PZ5 on May $5^{th}$, 2015. The simulated chemical
compositions after geochemical integration along the flow lines are compared with the initial
soil solution and the spring chemistry data.
Table 1: measured pH, water discharges and chemical concentrations of $H_4SiO_4$, $Na^+$, $K^+$, $Mg^{2+}$,
and $Ca^{2+}$ in water samples collected at the Strengbach catchment and used for the
hydrogeochemical modeling. The sampling sites include springs (CS1, CS2, RH3) and
piezometers (PZ3, PZ5).
Table 2: Initial and calibrated values of the hydrodynamic parameters of the aquifer in the
hydrological simulation of the Strengbach catchment by NIHM.
Table 3: Comparison between BET surfaces and geometric surfaces for the major primary
minerals present in a granitic context. BET surfaces were measured via gas absorption
experiments by [1] Berger et al., 1994; [2] Chou and Wollast, 1985; [3] Lundstrom and Ohman, 1990;
[4] Amrhein and Suarez, 1992; [5] Acker and Bricker, 1992; and [6] Guidry and Mackenzie, 2003.
Geometric surfaces were recalculated from the granulometric ranges of the minerals and by
assuming a spherical geometry.

| | Na$^+$ (mmol/L) | K$^+$ (mmol/L) | Mg$^{2+}$ (mmol/L) | Ca$^{2+}$ (mmol/L) | H$_4$SiO$_4$ (mmol/L) | pH | Water Discharge (L/s) |
|---|---|---|---|---|---|---|---|
| | | | | | | | |
| **Spring CS1** | | | | | | | |
| 16/09/2008 | 0.071 | 0.013 | 0.017 | 0.044 | 0.129 | 6.28 | 0.954 |
| 30/03/2010 | 0.074 | 0.014 | 0.015 | 0.043 | 0.120 | 5.61 | 1.523 |
| 29/03/2011 | 0.074 | 0.013 | 0.015 | 0.038 | 0.145 | 6.23 | 0.345 |
| 04/10/2011 | 0.080 | 0.012 | 0.016 | 0.042 | 0.176 | 6.57 | 0.122 |
| 29/11/2011 | 0.088 | 0.015 | 0.019 | 0.034 | 0.177 | 6.30 | 0.098 |
| 05/05/2015 | 0.065 | 0.012 | 0.012 | 0.054 | 0.121 | 5.33 | 1.410 |
| | | | | | | | |
| **Spring CS2** | | | | | | | |
| 30/03/2010 | 0.090 | 0.020 | 0.020 | 0.080 | 0.122 | 6.15 | 6.274 |
| 29/03/2011 | 0.090 | 0.020 | 0.020 | 0.070 | 0.144 | 6.18 | 0.956 |
| 02/08/2011 | 0.090 | 0.020 | 0.020 | 0.060 | 0.170 | 6.50 | 2.171 |
| 04/10/2011 | 0.100 | 0.020 | 0.020 | 0.070 | 0.177 | 6.76 | 0.413 |
| 29/11/2011 | 0.100 | 0.020 | 0.020 | 0.060 | 0.180 | 6.22 | 0.285 |
| 05/05/2015 | 0.077 | 0.016 | 0.018 | 0.074 | 0.123 | 6.14 | 7.500 |
| | | | | | | | |
| **Spring RH3** | | | | | | | |
| 30/03/2010 | 0.083 | 0.028 | 0.032 | 0.081 | 0.127 | 6.28 | - |
| | | | | | | | |
| **Piezometer PZ3** | | | | | | | |
| 05/05/2015 | 0.074 | 0.013 | 0.011 | 0.053 | 0.153 | 6.29 | - |
| | | | | | | | |
| **Piezometer PZ5** | | | | | | | |
| 05/05/2015 | 0.072 | 0.013 | 0.017 | 0.058 | 0.132 | 6.16 | - |

*Table 1*

| Parameter | Unit | Initial Value | Calibrated value |
|---|---|---|---|
| Depth of substratum | m | 4 | See figure 2 |
| Saturated hydraulic conductivity (all zones except the low depth zone at the catchment peak (see figure 2)) | m/s | 1E-04 | 8E-05 |
| Saturated hydraulic conductivity (catchment peak) | m/s | 1E-04 | 1E-04 |
| porosity (all zones except the low depth zone at the catchment peak) | - | 0.1 | 0.08 |
| Porosity (catchment peak) | - | 0.1 | 0.2 |
| Residual water content (all zones) | - | 0.01 | 0.01 |
| Specific storage (all zones) | $m^{-1}$ | 1E-08 | 1E-08 |
| n (Van Genuchten coefficient, all zones) | - | 2 | 2 |
| α (Van Genuchten coefficient, all zones) | $m^{-1}$ | 1 | 1.5 |
| Thickness of the interface layer between the groundwater compartment and the surface compartment | m | 0.1 | 0.05 |
| Saturated hydraulic conductivity of the interface layer between the groundwater compartment and the surface compartment | m/s | 1E-04 | 1.2E-05 |

*Table 2*

| Mineral | Mineral density (g/cm³) | Granulometric range (µm) | Particle radius (µm) | Spherical geometric surface (m²/g) | BET surface (m²/g) |
|---|---|---|---|---|---|
| | | | | | |
| Quartz[1] | 2.62 | < 50 | 1 - 25 | 1.150 - 0.046 | 0.310 |
| Albite[2] | 2.60 | 50 - 100 | 25 - 50 | 0.046 - 0.023 | 0.075 |
| K-feldspar[3] | 2.56 | < 50 | 1 - 25 | 1.170 - 0.047 | 1.420 |
| Anorthite[4] | 2.73 | 20 - 50 | 10 - 25 | 0.044 - 0.111 | 0.500 |
| Biotite[5] | 3.09 | 150 - 400 | 75 - 200 | 0.013 - 0.005 | 0.240 |
| Apatite[6] | 3.19 | 100 - 200 | 50 - 100 | 0.018 – 0.009 | 0.026 |

*Table 3*

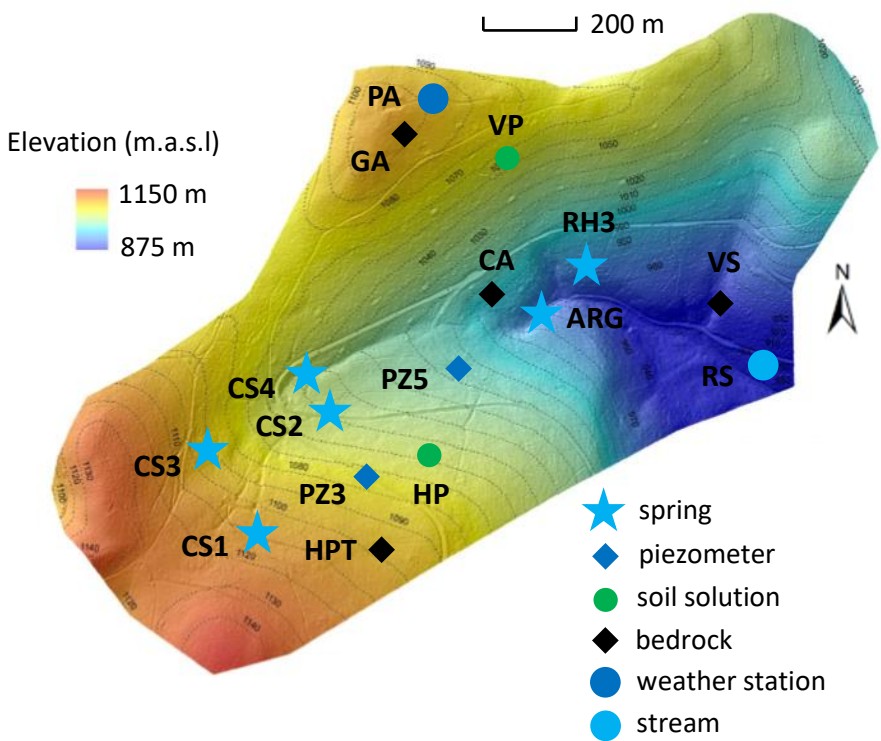

*Figure 1*

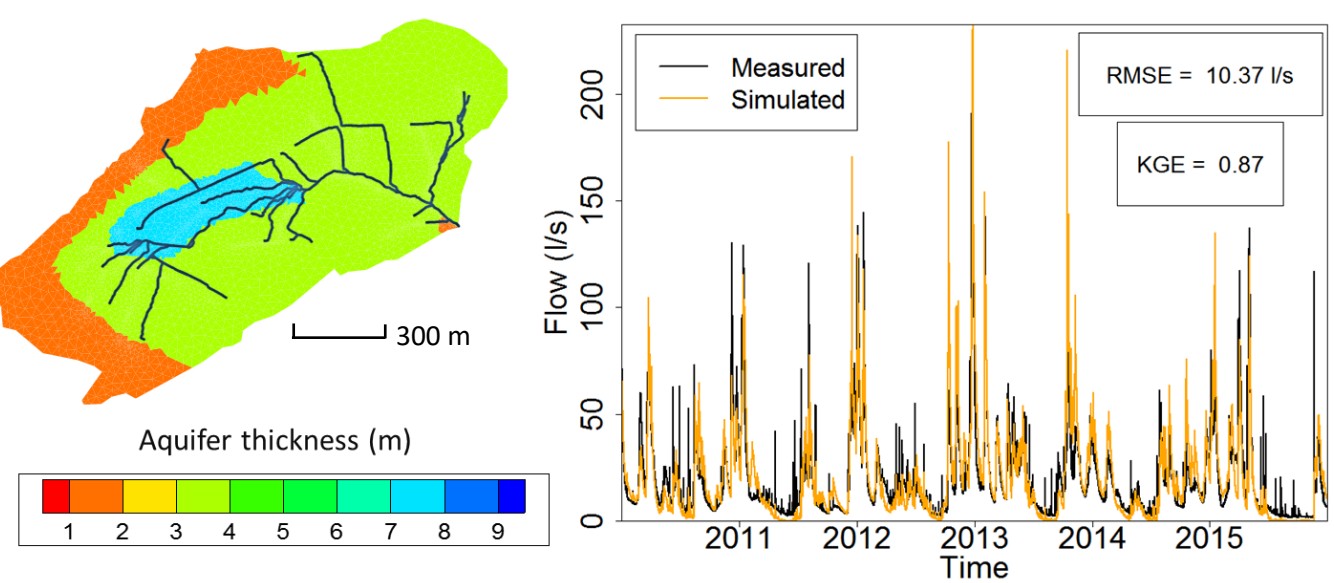

Figure 2

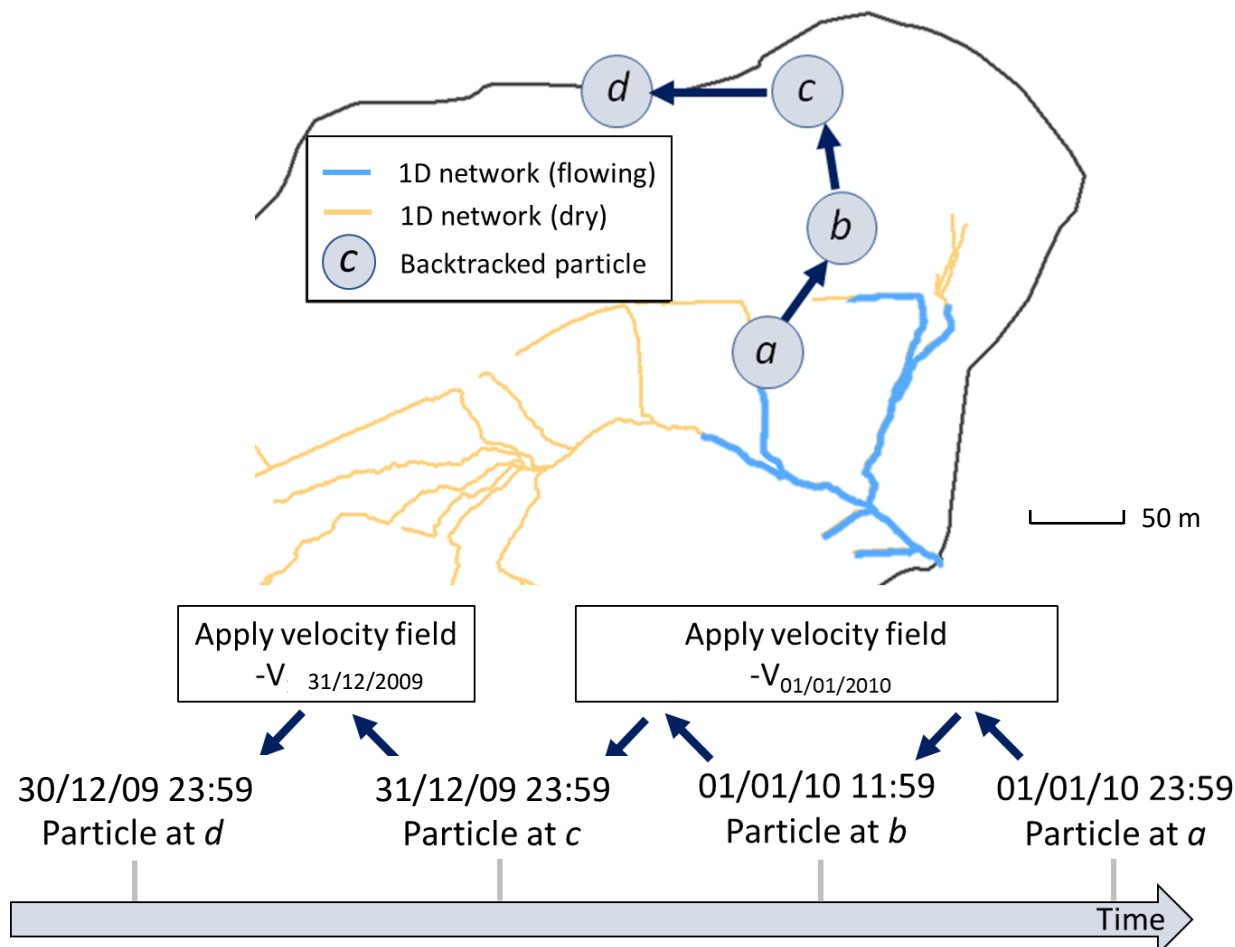

*Figure 3*

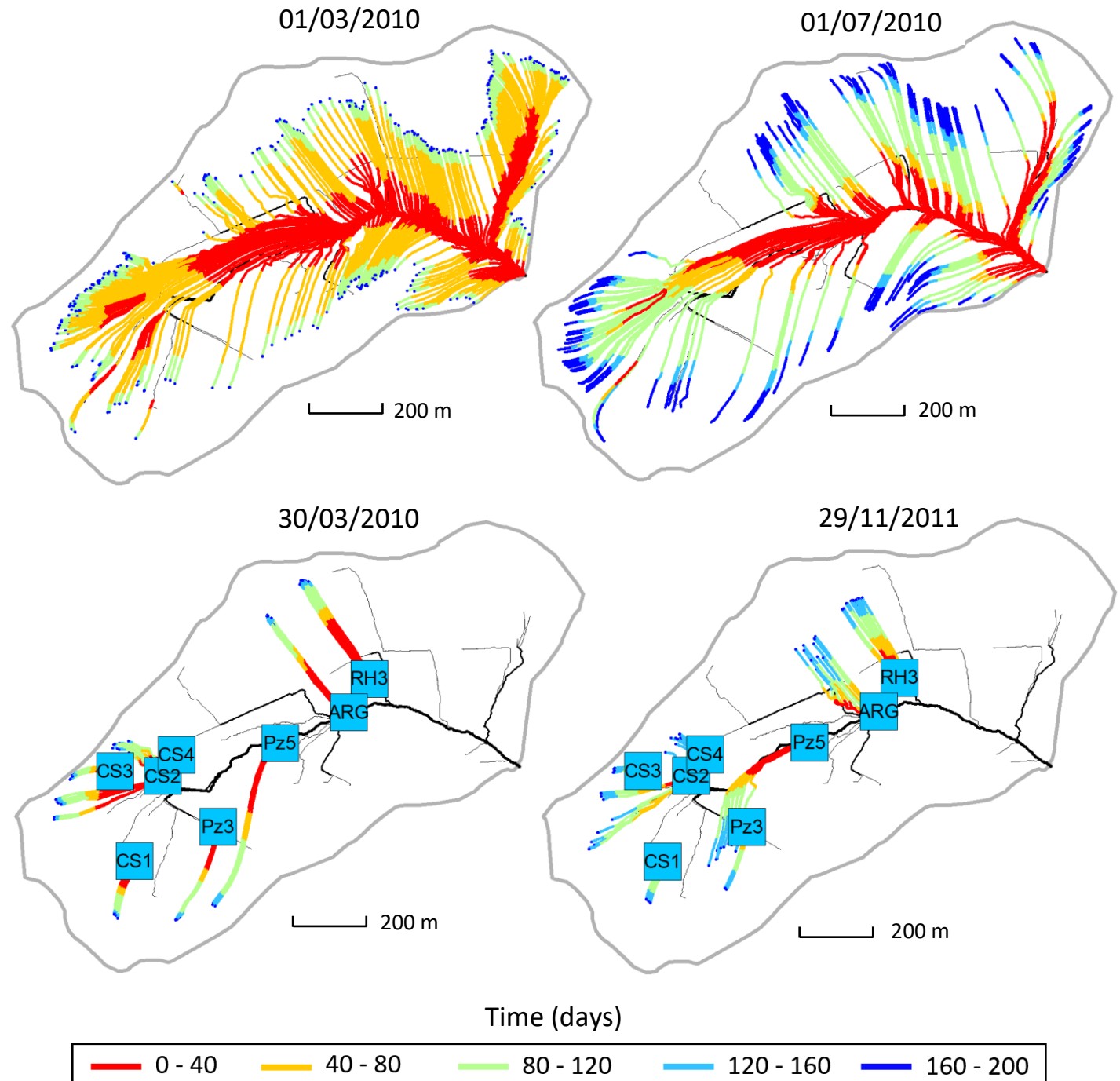

Figure 4

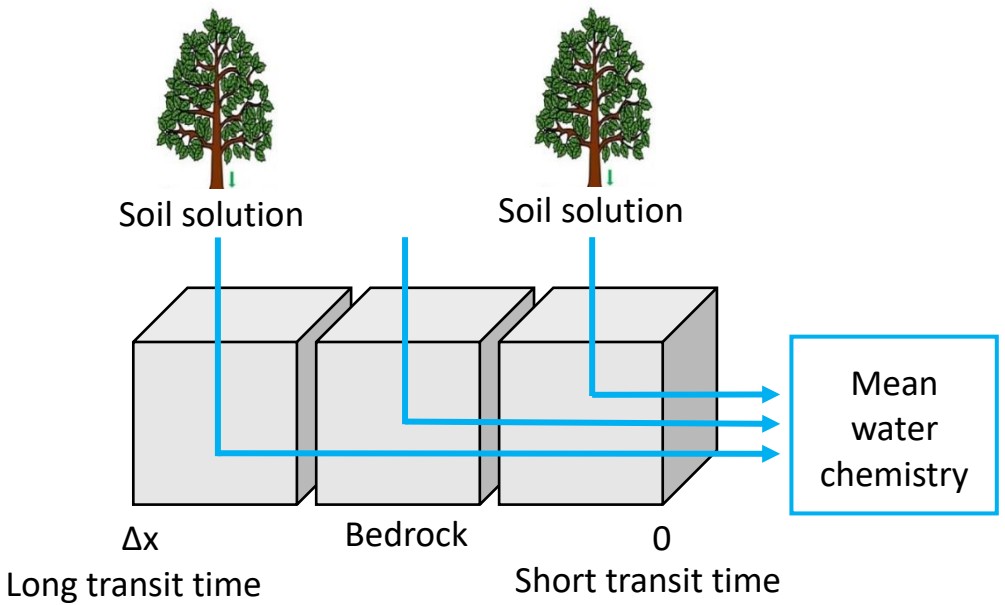

*Figure 5*

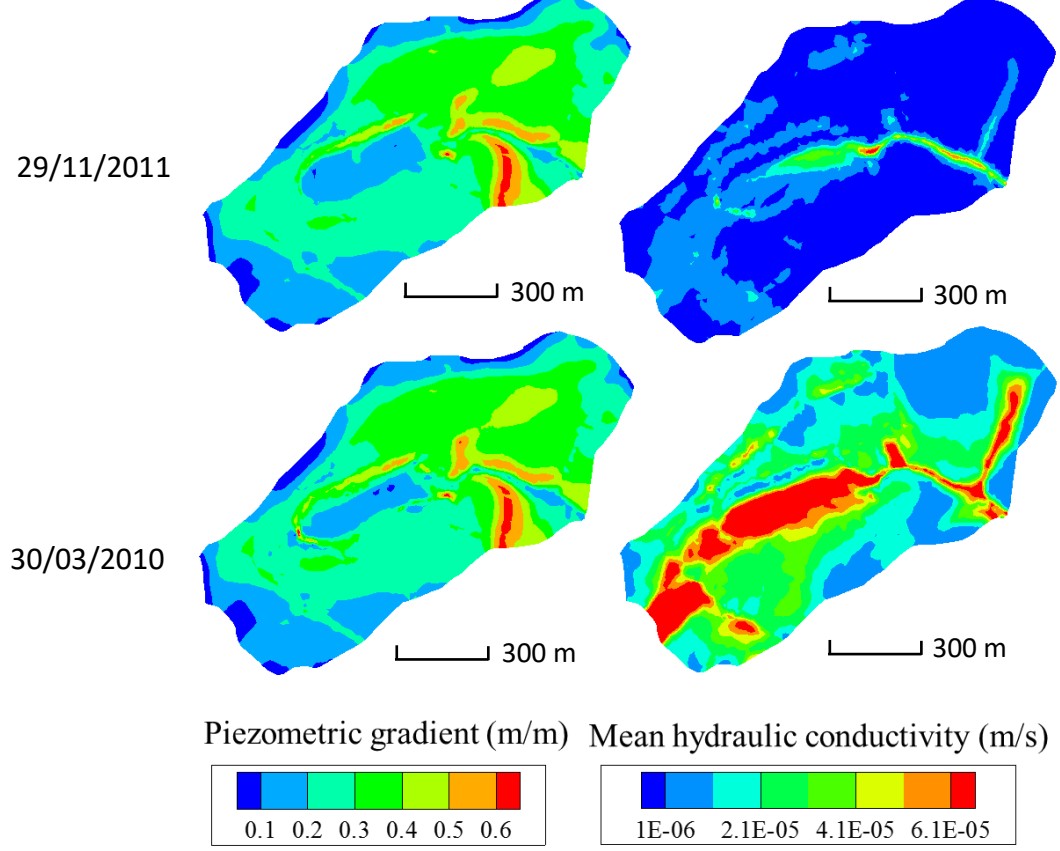

29/11/2011

30/03/2010

Piezometric gradient (m/m)    Mean hydraulic conductivity (m/s)

0.1  0.2  0.3  0.4  0.5  0.6          1E-06   2.1E-05  4.1E-05  6.1E-05

*Figure 6*

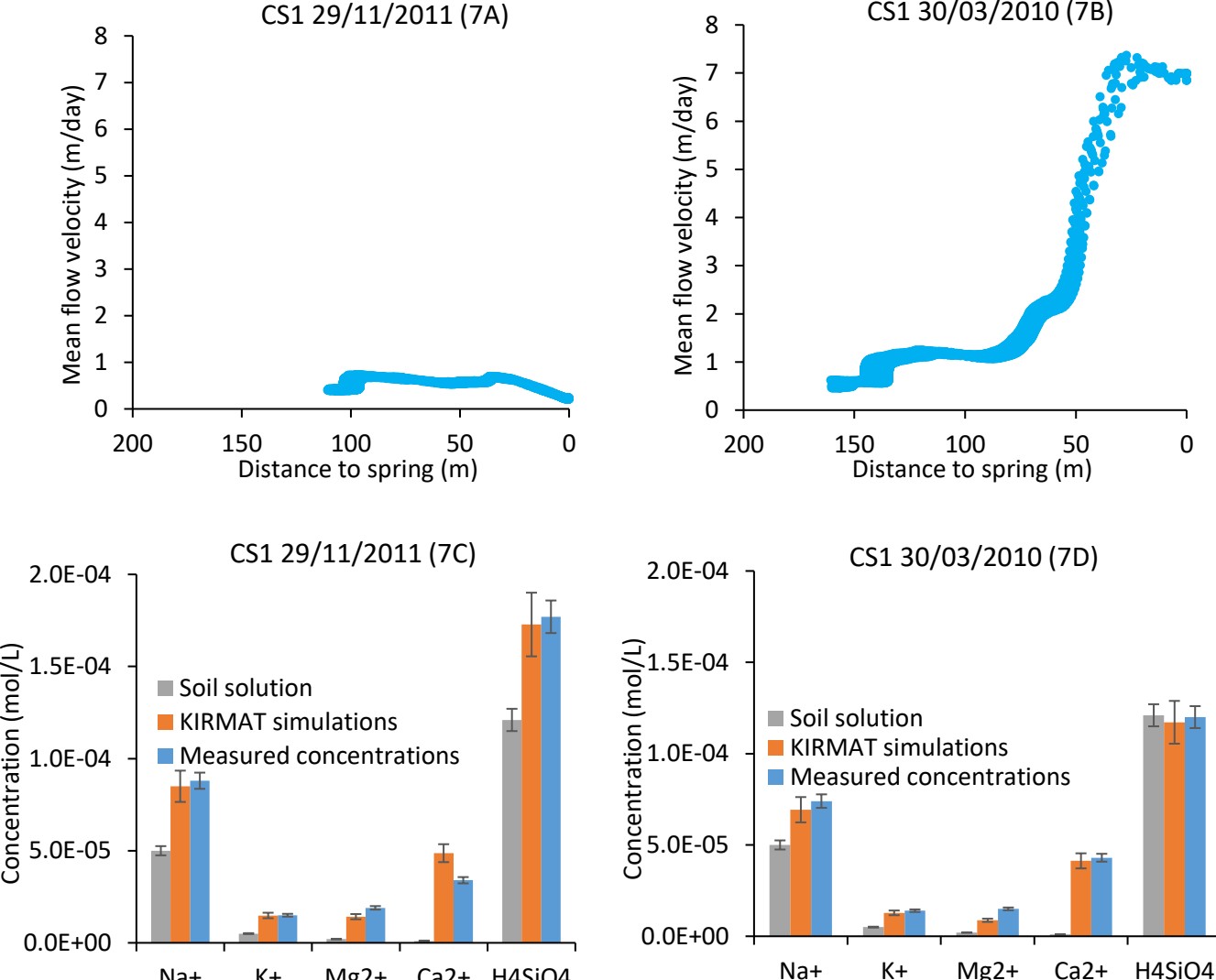

*Figure 7*

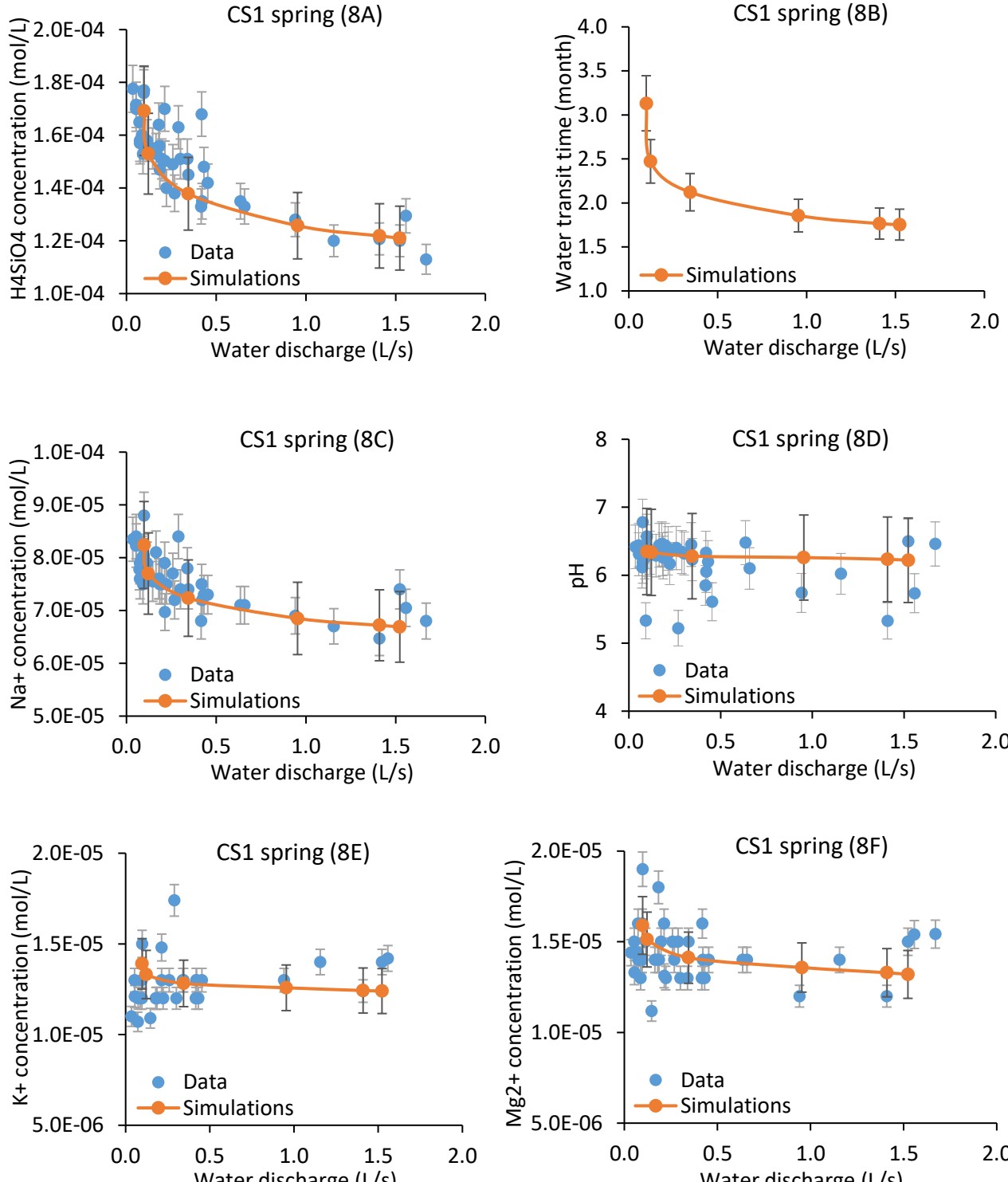

*Figure 8*

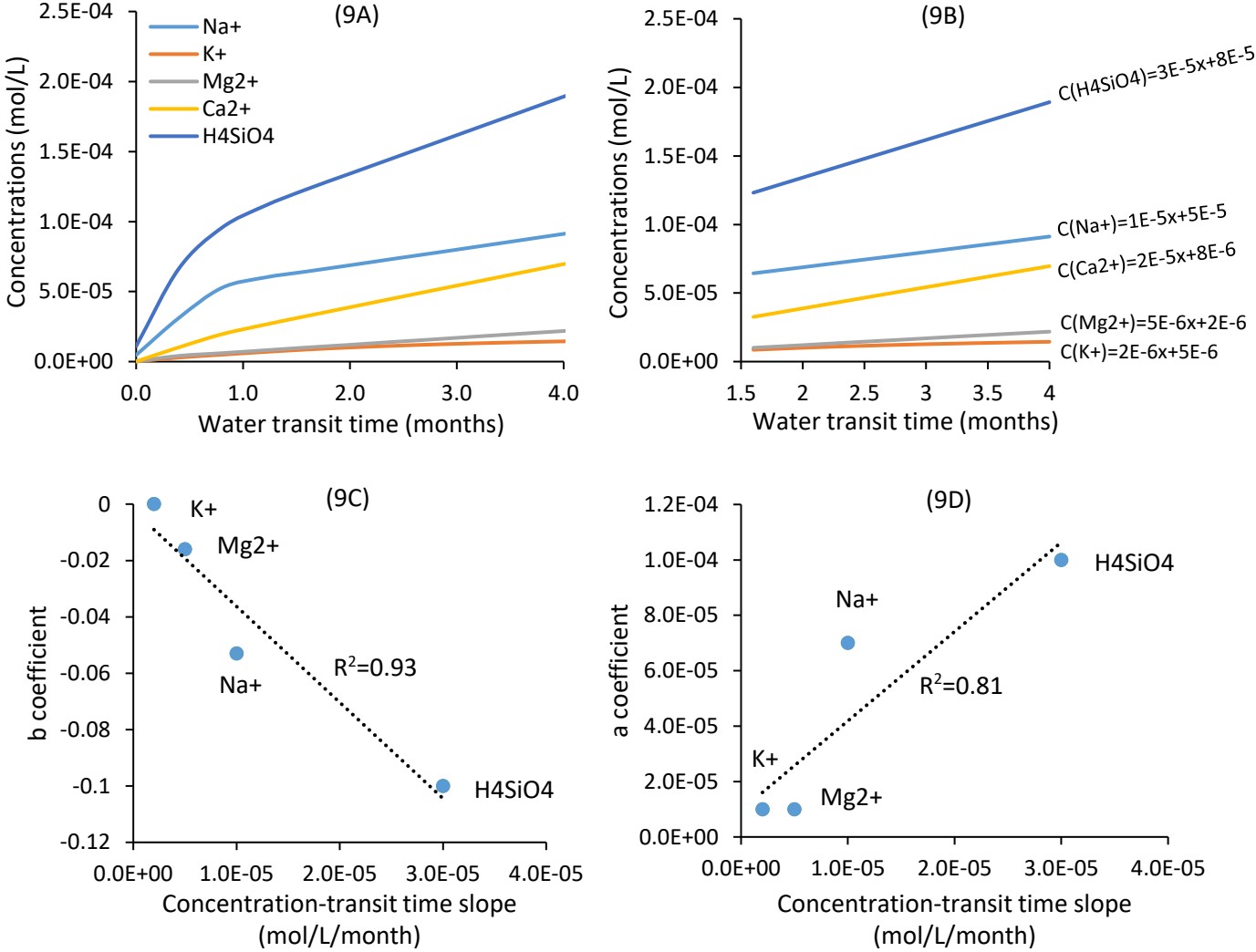

Figure 9

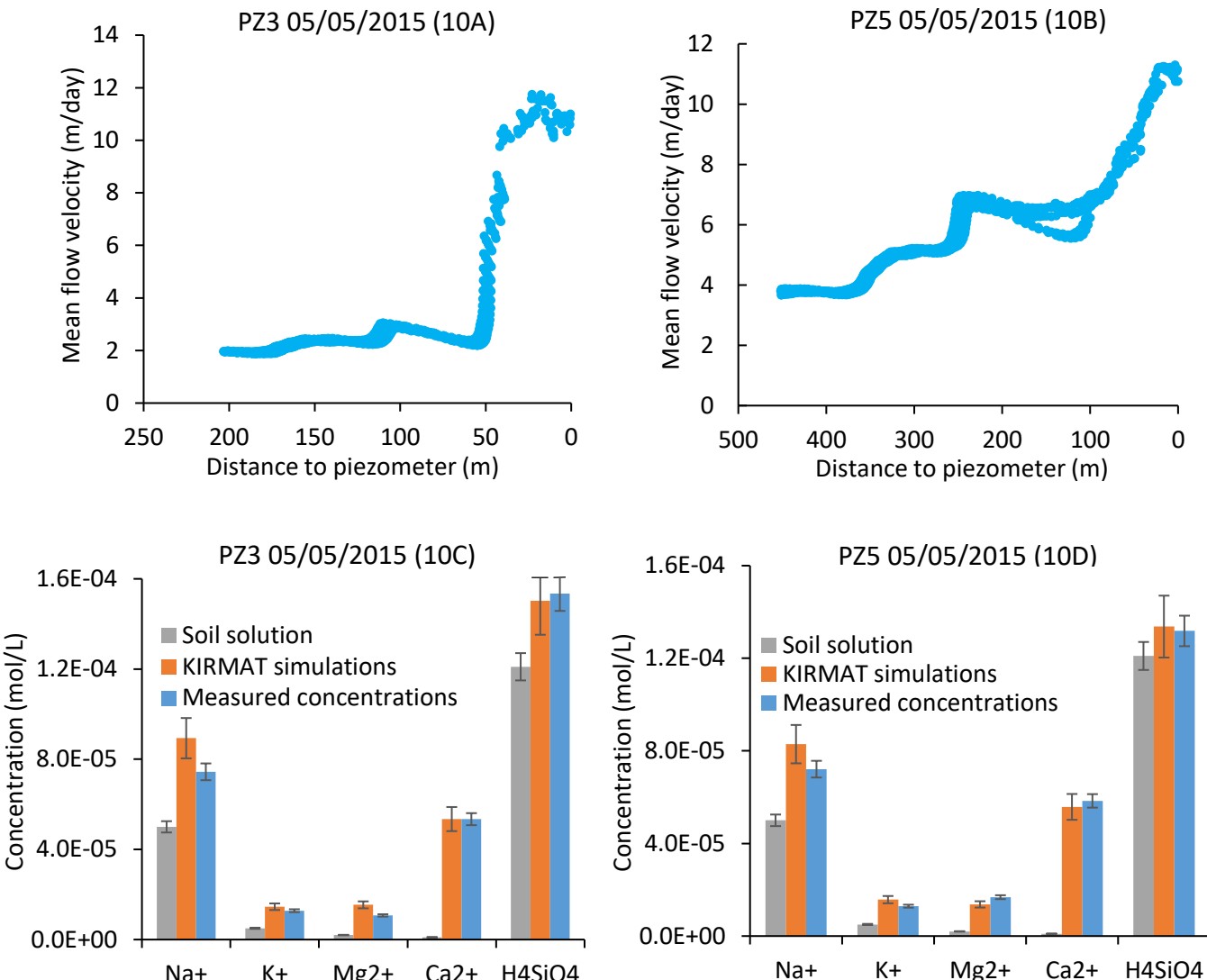

*Figure 10*

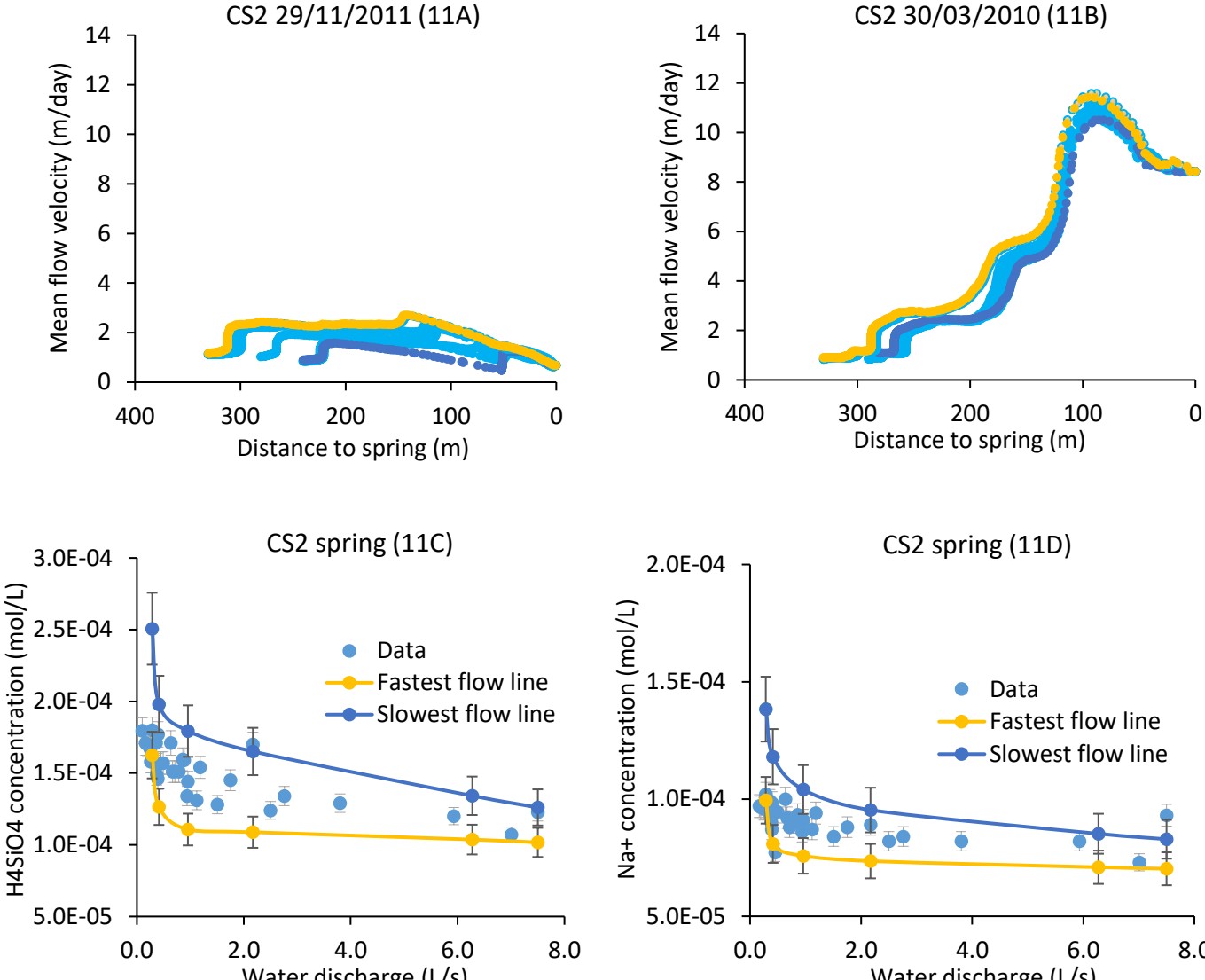

Figure 11

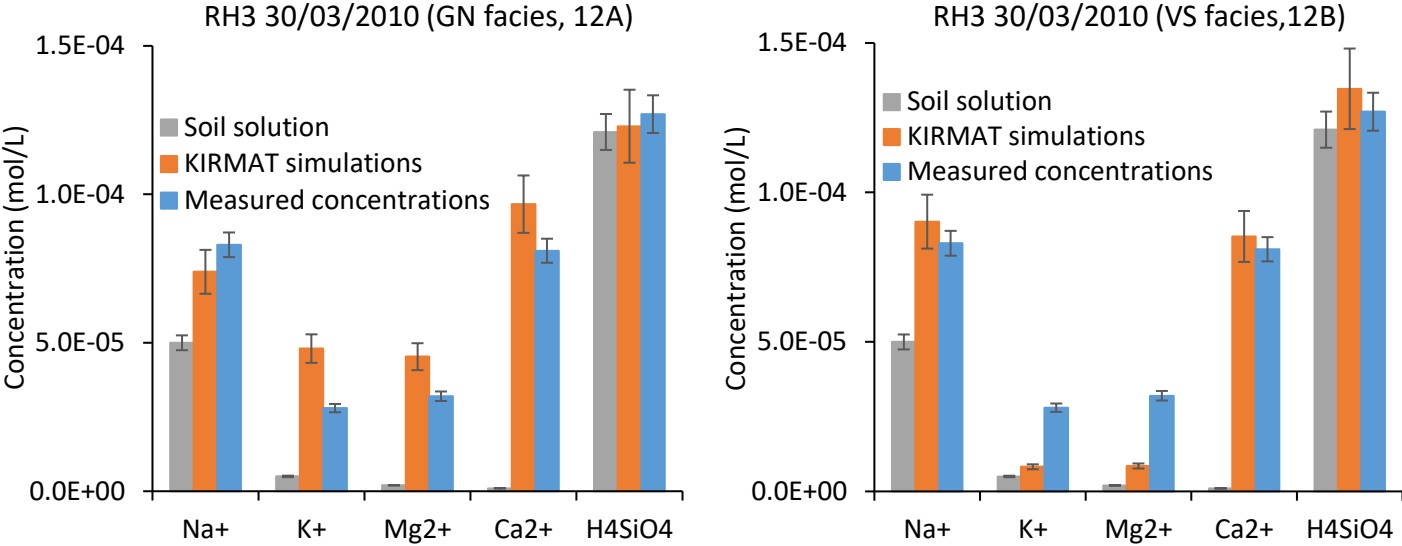

Figure 12

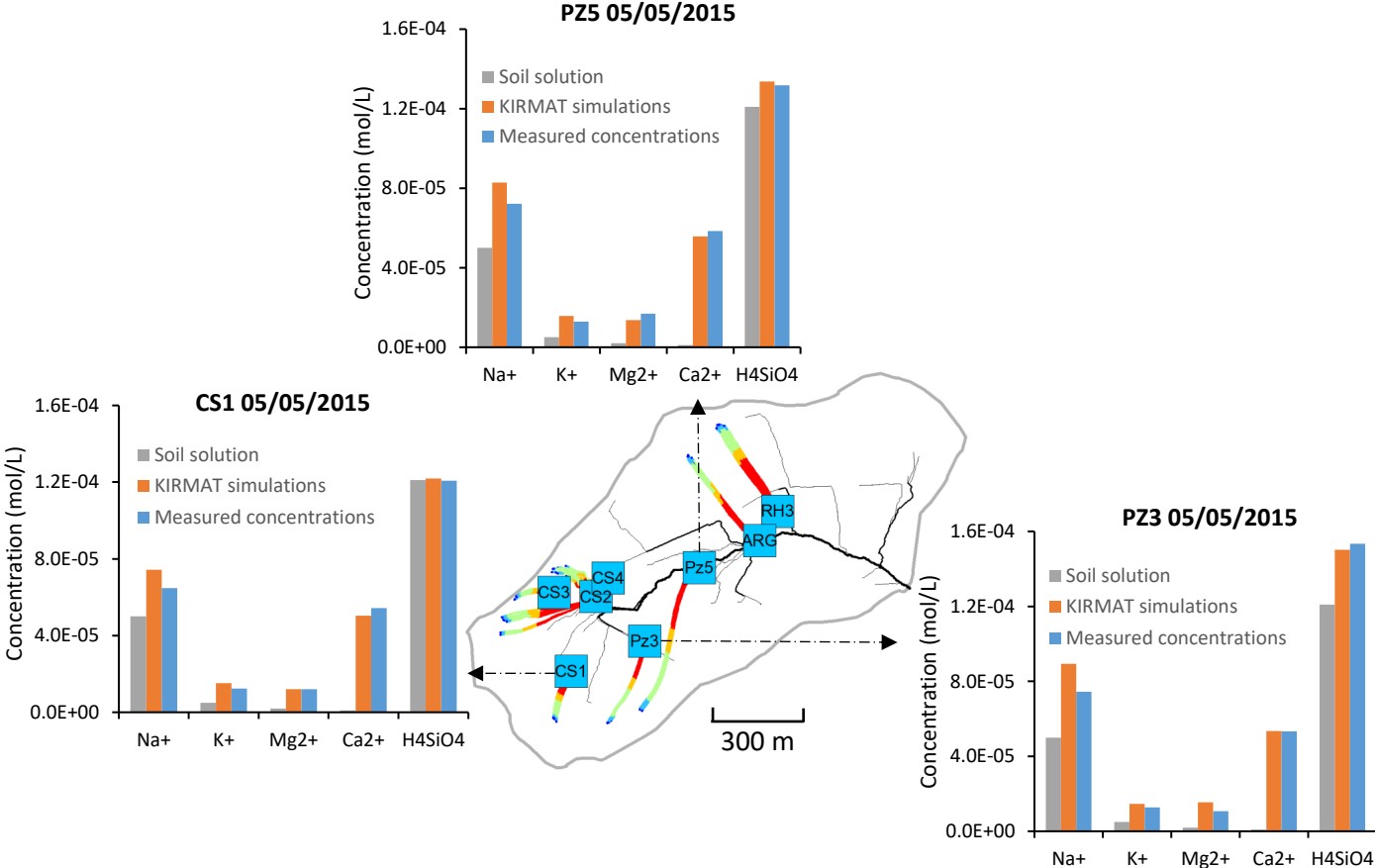

*Figure 13*