# Peer review of "Crossing hydrological and geochemical modeling to understand the spatiotemporal variability of water chemistry in a headwater catchment (Strengbach, France)"

_Hydrology and Earth System Sciences, 2018_

## Referee Comment (RC1) · Anonymous Referee #1 · 19 Feb 2019

Highlights: The study presents a coupled hydrological + reactive transport model capable of characterizing the hydrogeochemical variability in a small watershed in France, (Strengbach). This model is composed of a depth-integrated and spatially-distributed NIHM model and Kinetic Reaction and MAss Transport KIRMAT reactive transport model. The principle results of these numerical simulations can be summarized as follows: (1) chemostatic behavior is a direct consequence of water transit times and, thus, hydrologically controlled; (2) small k reactivity constants + reactive surface areas are the only parameters necessary to constrain water chemistry.

This study represents a nice contribution to catchment modeling efforts in mergeing characteristically separate hydrologic and reactivity models together in order to better understand chemical and hydrological subsurface dynamics. However, it would seem there is some general omission of the rich variety of reactive transport simulation capabilities now available and in application to hillslope – watershed scale weathering and solute fluxed. Further there is a lack of clarity regarding which parameters are adjusted in the model and in order to match the discharge data, the functional form of the rate expressions and the treatment of solid solutions, all of which must be clearly articulated in order to ensure reproducibility. Finally, the paper needs substantial revision in terms of writing, organization, and formatting. Thus, I suggest major revisions.

Abstract Line 22: "over" rather than "for the next decades" Line 36-37: Unclear what the authors mean when stating "vary from approximately 1.5 to 3 months from floods to drought events." Lines 43-45: Unclear what the authors mean by "low surfaces"? What range of surface areas would be considered "low" for this study (I think based on Table 2, this is between 0.026 m2/g – 1.420 m2/g)? Why only "low surface areas"? What about the influence of secondary mineral precipitation (that leads to the formation of high surface area phases)? Lines 48-50: This sentence doesn't say anything new. I think that the "process-based" approach to characterizing water chemistry is something that everyone is doing. The key goal for our reactive transport models is to better characterizing those water transit times, especially between the variably saturated subsurface and deep groundwater. Either way, I think the authors need to wrap up this abstract with a final sentence that states what is so unique about their hydrogeochemical models that separates it from the rest.

Introduction General comment: Introduction looks fine, but a couple points are missing that could help provide a fuller context for this study. In particular, the authors discuss concentration-discharge relationships in the abstract, but fail to return to this topic in the introduction. I think discussing C-Q relationships and their use as a diagnostic tools to understand subsurface reactivity and transport is a key context for arguing why an in-

tegrated hydrologic + RTM model is necessary. Specific comments: Lines 56-58: 'That results...' I don't understand what this sentence is trying to say. What "results from", exactly? The challenge of estimating the repercussions of climate change to the overall hydrological cycle? I suggest that this sentence be re-phrased for clarity. Lines 90-95: also water quality issues... Lines 104: I think there is something missing between the sentences " A new step is therefore necessary... " and "This is the aim of this work, which combines for the first time in this manner...". This is a key area in which the authors omit acknowledgement of important and current studies that have sought to combine RTM with hydrological models – for instance, ParCrunchFlow (Beisman et al. 2015), PHIM (Li Li & Sue Brantley), DHARA (Kumar).

Site presentation and data acquisition Line 141: with a sampling frequency sufficient to cover the entire range of water discharges... rather than "allowing for covering the entire range of water discharges...". Also it's unclear what this phrase means. Do the authors refer to daily data? Hourly data? Important to be specific on this point as it has critical implications for the model. Line 160: presented in this study rather than presented in the following". Line 164: "chemical data ... presented in Table 1". The authors' mention (line 145) that soil solutions were also sampled and provide citations to Gangloff et al. 2014 and Prunier et al. 2015 for the dissolved major cation concentrations. This is useful, but it would be better to have that information directly included in Table 1 in addition to the spring water data. Further, it is unclear if the resolution of the geochemical data is on a yearly-scale or monthly-scale, etc. based on the data presented in Table 1. It's also confusing based on the C-Q plots in Fig. 8 and Fig.10 where there seems to be more data points presented for the CS1 and CS2 springs than 12 points shown in Table 1. This needs to be cleaned up. In general, it's good practice to present the entire geochemical dataset to the reader (it can be included in a supplementary table if necessary).

Modeling methods Line 167: remove "acquisition of" Line 188: "Water exchange" rather than "the water exchanges" Line 190-198: What parameters in the model are tuned to

match the discharge data?? This must be clearly articulated to ensure reproducibility. How is it determined whether the water discharges are correctly reproduced? Is this done purely by visual comparison or is a statistical approach or some threshold of reproducibility (within a 90-95% confidence interval for instance) utilized? Line 225: What is the form of the chemical rate law used in the KIRMAT code? Is it based in Transition State Theory? Line 228: How does the model track the clay solid solution composition with time?

Results Line 343: remove "that"

Discussion Line 489-490: Clarify whether the change in reactive surface area is monitored through time as the primary minerals dissolve. Depending on how the chemical rate law is defined (again this must be reported), a change in surface area has a first order control on net dissolution rates ($R = kA (Q/K – 1$; where $k$ = rate constant, $A$ = reactive surface area, $Q$ = activity quotient, and $K$ = equilibrium constant). So being able to monitor this evolution in surface area with time is important. Also, is secondary mineral precipitation included in the simulations? I know that the composition of clays are tracked in the model through a solid solution, but it's not clear if secondary mineral formation is tracked as well. This is an important component that needs to be addressed since secondary mineral precipitation would also impact net dissolution rates.

Line 513: What are the values of the "standard" kinetic constants that were used in these simulations? These constants (and references to the associated studies from which they came from) should be shown in Table 2 (or maybe another table) for the primary minerals – there is absolutely not a set of 'standard' values that everyone uses. Without all this information included (as well as an explanation of the impact of secondary mineral precipitation), the rest of this section is too speculative.

Lines 641 - 645: OK, only now the authors clarify that the reaction rates are based on transition state theory and discuss the influence of secondary mineral precip. But, this is far too late in the paper. This needs to be addressed much earlier, preferably in the

"hydrological methods' section.

Lines 646- 648: There is no evidence provided that supports the author's argument that a clay solid solution precipitated at equilibrium can generate reasonable precipitation rates. Maybe the authors could present the numerically generated precipitation rates along with precipitation rates found in the literature in a table?

Conclusion: Overall a bit long, it could be shortened a bit without losing the key points of this study.

Figures Figures 7-11 look like they have been quickly made in excel with little formatting done. I strongly suggest that the author take the time to improve these plots and properly format and organize axes, legends, and titles. I also suggest labels in each plot A, B, C etc. and reference to these specific panels in the text (i.e Fig. 7A, Fig. 7B, etc.) to make it easier for the reader. Additionally, error bars should be shown for the measured concentrations (and a description of what the error bars represent, 1SD, 2SD, etc.). This would help justify the author's assertion that the KIRMAT simulations agree with measured concentrations.

––––––––––––––––––––

---

## Referee Comment (RC2) · Anonymous Referee #2 · 19 Feb 2019

The study 'Crossing hydrological and geochemical modeling to understand the spatiotemporal variability of water chemistry in an elementary watershed' by Ackerer et al. presents results from coupled hydrological and geochemical reactive transport models of a small watershed. The authors find that observed 'chemostatic' behavior is controlled by seasonal patterns of subsurface transit times and suggest that simple geometric representations of mineral surface areas may be sufficient to match lab and field measured effective rates. While I find this study promising and potentially interesting to a broad community of researchers, I believe there are some significant issues with the

authors' methodologies and conclusions that merit major revisions or resubmission. It is possible that I have also misunderstood some of the technical issues, as the authors do not provide adequate description of the modeling and validation methodologies, as detailed by Referee #1.

First, I'd like to echo comments offered by Referee #1 — this manuscript needs a much more detailed description of the geochemical modeling including in depth descriptions of rate parameterizations, reaction pathways, kinetic rates, and equilibrium constants used. As it stands, it is hard to contextualize any of the presented results. Additionally, there have been other studies which combine fully distributed hydrologic models with geochemical reactive transport code that should be acknowledged.

Flowpath Modeling: From what I can piece together, the back-trajectory simulations provide subsurface flow path lines that all originate at the boundary of the watershed domain. If the hydrologic model simulates subsurface/surface water connections as described, why aren't flowpath origins distributed evenly across the land surface? It seems the authors provide a quick fix to this situation by assigning an even distribution of inputs along a flowpath, but this step and its necessity should be described in more detail. In particular, this situation is characterized as 'realistic' in the beginning of the discussion, but from my understanding, other codes such as ParFlow / SLIMFast back trajectories would not need this fix because they predict the origin of waters across the land surface.

Additionally, from the diagram in Fig 5, are the measured soil solutions taken as inputs into the modeling domain? If so, it would be extremely useful to see where these fall relative to the C-Q plots in Fig's 8 and 10. What is particularly important is seeing what soil concentrations are relative to precipitation concentrations. In other words, does a significant amount of solute generation occur in soils, and if so, is that being represented in the model at all? My understanding is that it is not, which could also be one of the primary causes of observations of chemostatic Na behavior at CS1 as mentioned in more detail below. What proportion of overall solute generation from

precipitation to spring is missed by not representing soil processes?

Model Validation: As far as I can tell, the primary validation for the model is a qualitative matching of modeled v. observed solute concentration patterns across discharge. This seems fine in general to me (though would be strengthened by some level of statistical analysis); however, the implications of this matching appear oversold. Specifically, the fact that model results match observations only means that this particular combination of parameters and subprocesses (water transit times, mineral surface areas, assumed reaction networks and associated kinetic and thermodynamic constants) combine to get the right answer, but that does not validate each individual subprocess. In other words, there is the problem of equifinality: if the hydrologic model is consistently underestimating transit times, for example, then the model could still match results by overestimating dissolution/precipitation rates. Without some independent validation of water transit times (i.e. seasonal water isotope variability, tracers), the authors cannot conclude that each individual process is accurately represented. This issue needs to be discussed, particularly w/r/t Section 6. As I read it the model-data match is used to independently validate (1) water transit time simulations; (2) the fact that bedrock waters don't need representation; (3) mineral surface areas and kinetic rate constants; (4) the specific representation of clay solid solution series (which is not adequately described) — in my mind these conclusions are not sufficiently supported without independent validation of these sub-processes.

Also echoing Referee #1, it would be useful to include more description of the C-Q dynamics – both in the introduction and in the analysis. Specifically, Si seems much less chemostatic than Na, particularly at CS1. Why is this and why does it also happen in the model? Are Na concentrations diluted until they reach observed soil concentrations (i.e. point above) in which case does its representation in the subsurface even matter?

Small point is that the authors consistently characterize precipitation and drought events as 'important' without context.

I also agree that figures would benefit from formatting to make them more readable – particularly y-axis labels of concentrations.

Lastly, and in general, these subsurface transit times seem relatively fast which may make sense for such a small catchment system. However, how does this compare to measured transit times in other catchment systems (for example as a function of watershed scale)? In other words, how applicable are the conclusions that chemostatic behavior here reflects far-from-equilibrium hydrologic controls to other areas?

---

## Author Comment (AC1) · 2 Apr 2019

**Answer to Referee 1**

**Anonymous Referee #1**

Highlights: The study presents a coupled hydrological + reactive transport model capable of characterizing the hydrogeochemical variability in a small watershed in France, (Strengbach). This model is composed of a depth-integrated and spatially-distributed NIHM model and Kinetic Reaction and MAss Transport KIRMAT reactive transport model. The principle results of these numerical simulations can be summarized as follows: (1) chemostatic behavior is a direct consequence of water transit times and, thus, hydrologically controlled; (2) small k reactivity constants + reactive surface areas are the only parameters necessary to constrain water chemistry.

This study represents a nice contribution to catchment modeling efforts in mergeing characteristically separate hydrologic and reactivity models together in order to better understand chemical and hydrological subsurface dynamics. However, it would seem there is some general omission of the rich variety of reactive transport simulation capabilities now available and in application to hillslope – watershed scale weathering and solute fluxed. Further there is a lack of clarity regarding which parameters are adjusted in the model and in order to match the discharge data, the functional form of the rate expressions and the treatment of solid solutions, all of which must be clearly articulated in order to ensure reproducibility. Finally, the paper needs substantial revision in terms of writing, organization, and formatting. Thus, I suggest major revisions.

*We agree that a few recent studies geared towards the coupling between hydrological and geochemical approaches (e.g., Beisman et al., 2015; Li et al., 2017) could be duly acknowledged. Therefore, the introduction has been slightly modified to underline some of these works, and also to better emphasize the points that give originality to our present contribution (lines 110-120).*

*We also answered to Rev #1 who claimed for improved clarity regarding the adjusted parameters in the models. The fitted parameters employed in the hydrological model to match observed discharge data at the outlet of the watershed are provided in lines 200-222. A new table also gathers the initial and adjusted values of parameters in the hydrological simulations by NIHM (table 2). The treatment of solid solutions is better explained in lines 255-273. Additional geochemical data and the values of geochemical parameters used for reactive transport simulations are now reported in tables supplied as supplementary material including: minerals, reactive surfaces, thermodynamic and kinetic constants, and clay solid solution end members (tables EA1-EA13).*

Abstract Line 22: "over" rather than "for the next decades" Line 36-37: Unclear what the authors mean when stating "vary from approximately 1.5 to 3 months from floods to drought events." Lines 43-45: Unclear what the authors mean by "low surfaces"? What range of surface areas would be considered "low" for this study (I think based on Table 2, this is between 0.026 m2/g – 1.420 m2/g)? Why only "low surface areas"? What about the influence of secondary mineral precipitation (that leads to the formation of high surface area phases)? Lines 48-50: This sentence doesn't say anything new. I think that the "process-based" approach to characterizing water chemistry is something that everyone is doing. The key goal for our reactive transport models is to better characterizing those water transit times, especially between the variably saturated subsurface and deep groundwater. Either way, I think the authors need to wrap up this abstract with a final sentence that states what is so unique about their hydrogeochemical models that separates it from the rest.

*These points have been modified or rephrased to improve clarity (lines 25, 38-41). The range of values spanned by "low" surface values is now given line 47. It is also explained that the secondary minerals precipitated in the simulations are clay minerals (lines 265-266). Precipitation of other secondary minerals such as carbonates, amorphous silica, hematite, etc., have been envisioned. But in view of the hydro-climatic context prevailing at the Strengbach catchment, these secondary minerals did not precipitate in the various simulations performed (lines 270-273). The end of the abstract has been partly rewritten to better point out the originality of our modeling approach (lines 47-52).*

Introduction General comment: Introduction looks fine, but a couple points are missing that could help provide a fuller context for this study. In particular, the authors discuss concentration-discharge relationships in the abstract, but fail to return to this topic in the introduction. I think discussing C-Q relationships and their use as a diagnostic tools to understand subsurface reactivity and transport is a key context for arguing why an integrated hydrologic + RTM model is necessary. Specific comments: Lines 56-58: 'That results…" I don't understand what this sentence is trying to say. What "results from", exactly? The challenge of estimating the repercussions of climate change to the overall hydrological cycle? I suggest that this sentence be re-phrased for clarity. Lines 90-95: also water quality issues… Lines 104: I think there is something missing between the sentences " A new step is therefore necessary… " and "This is the aim of this work, which combines for the first time in this manner…". This is a key area in which the authors omit acknowledgement of important and current studies that have sought to combine RTM with hydrological models – for instance, ParCrunchFlow (Beisman et al. 2015), PHIM (Li Li & Sue Brantley), DHARA (Kumar).

*The introduction has been modified to mention the concentration-discharge relationships as a tool to qualitatively inform on subsurface reactions and transport, these mechanisms being then quantitatively re-handled via coupled hydrological and geochemical models (lines 60-69). Some points are also rephrased for the sake of clarity (lines 57-60).*

*We acknowledge that a few recent works coupling hydrological and geochemical models were omitted. Those relying upon ParCrunchFlow (Beisman et al., 2015) and PHIM simulations (Li et al.,*

*2017) are now quoted, while the introduction was slightly modified to present their content (lines 110-113). This rewriting also better mentions where to find the differences in our contribution compared with these previous works (lines 113-120).*

Site presentation and data acquisition Line 141: with a sampling frequency sufficient to cover the entire range of water discharges… rather than "allowing for covering the entire range of water discharges…". Also it's unclear what this phrase means. Do the authors refer to daily data? Hourly data? Important to be specific on this point as it has critical implications for the model. Line 160: presented in this study rather than presented in the following". Line 164: "chemical data … presented in Table 1". The authors' mention (line 145) that soil solutions were also sampled and provide citations to Gangloff et al. 2014 and Prunier et al. 2015 for the dissolved major cation concentrations. This is useful, but it would be better to have that information directly included in Table 1 in addition to the spring water data. Further, it is unclear if the resolution of the geochemical data is on a yearly-scale or monthly-scale, etc. based on the data presented in Table 1. It's also confusing based on the C-Q plots in Fig. 8 and Fig.10 where there seems to be more data points presented for the CS1 and CS2 springs than 12 points shown in Table 1. This needs to be cleaned up. In general, it's good practice to present the entire geochemical dataset to the reader (it can be included in a supplementary table if necessary).

*These suggestions have been accounted for in the revision of the manuscript (lines 151-153). It is now better specified that the geochemical database collects histories with a monthly resolution, some additional data being also available from specific sampling campaigns (lines 151-158). To avoid trimming the main text with large tables, the data from soil solutions are presented as supplementary material in tables EA6-EA7. It is also clarified in the captions of figures 8 and 10 that the data points can be found in table 1 and in supplementary table EA1. The complete geochemical dataset employed in our study is provided as supplementary tables (EA1-EA9).*

Modeling methods Line 167: remove "acquisition of" Line 188: "Water exchange" rather than "the water exchanges" Line 190-198: What parameters in the model are tuned to match the discharge data?? This must be clearly articulated to ensure reproducibility. How is it determined whether the water discharges are correctly reproduced? Is this done purely by visual comparison or is a statistical approach or some threshold of reproducibility (within a 90-95% confidence interval for instance) utilized? Line 225: What is the form of the chemical rate law used in the KIRMAT code? Is it based in Transition State Theory? Line 228: How does the model track the clay solid solution composition with time?

*These suggestions have been taken into account in the revised version. Notably, a paragraph has been added to state how the hydrological model is run to match the discharge data at the outlet of the system (lines 200-222), and which indicator is used to assess the matching between model outputs and data (KGE: Kling Gupta efficiency coefficient, lines 217-219). A new table also gives the parameters used in the NIHM simulations (table 2). Additional information on the KIRMAT model is available. It is specified in the Section on methods (lines 249-252) that KIRMAT is based on the Transient State Theory; clay solid solutions are better depicted (lines 255-262). We also refer to the previous work by Ackerer et al., 2018 where additional information about KIRMAT is available.*

Results Line 343: remove "that"

Discussion Line 489-490: Clarify whether the change in reactive surface area is monitored through time as the primary minerals dissolve. Depending on how the chemical rate law is defined (again this must be reported), a change in surface area has a first order control on net dissolution rates ($R = kA (Q/K - 1$; where $k$ = rate constant, $A$ = reactive surface area, $Q$ = activity quotient, and $K$ = equilibrium constant). So being able to monitor this evolution in surface area with time is important. Also, is secondary mineral precipitation included in the simulations? I know that the composition of clays are tracked in the model through a solid solution, but it's not clear if secondary mineral formation is tracked as well. This is an important component that needs to be addressed since secondary mineral precipitation would also impact net dissolution rates.

*"That" in line 343 has been removed.*

*The previous discussion of lines 489-490 has also been clarified. Reactive surface areas were monitored over time for primary minerals (lines 267-268), which showed that, within the short timescale of the simulations, the change in surface areas were negligible (lines 268-270). More information on secondary minerals and clay minerals are provided lines 255-262 and lines 271-273. It is worth noting that these points are now presented in the methodological Section 3, as they are related to the settings of the reactive transport model.*

Line 513: What are the values of the "standard" kinetic constants that were used in these simulations? These constants (and references to the associated studies from which they came from) should be shown in Table 2 (or maybe another table) for the primary minerals – there is absolutely not a set of 'standard' values that everyone uses. Without all this information included (as well as an explanation of the impact of secondary mineral precipitation), the rest of this section is too speculative.

*New tables are available as supplementary material to detail all the thermodynamic, kinetic constants and other geochemical parameters that are used in the modeling tasks (tables EA10-EA13). References to studies associated with the elaboration of these tables are also quoted in the supplementary material. More information on secondary minerals is provided in lines 255-262, lines 270-272, and also in lines 712-718.*

Lines 641 - 645: OK, only now the authors clarify that the reaction rates are based on transition state theory and discuss the influence of secondary mineral precip. But, this is far too late in the paper. This needs to be addressed much earlier, preferably in the "hydrological methods' section.

Lines 646- 648: There is no evidence provided that supports the author's argument that a clay solid solution precipitated at equilibrium can generate reasonable precipitation rates. Maybe the authors could present the numerically generated precipitation rates along with precipitation rates found in the literature in a table?

*These points are now clarified but presented earlier in the manuscript. The fact that a clay solution precipitated at equilibrium can render reasonable precipitation rates for the Strengbach watershed*

*case was previously discussed in Ackerer et al., 2018 (in section 6.2). To avoid useless repetition, the interested reader is referred to this recent work for more details (line 716).*

Conclusion: Overall a bit long, it could be shortened a bit without losing the key points of this study.

Figures Figures 7-11 look like they have been quickly made in excel with little formatting done. I strongly suggest that the author take the time to improve these plots and properly format and organize axes, legends, and titles. I also suggest labels in each plot A, B, C etc. and reference to these specific panels in the text (i.e Fig. 7A, Fig. 7B, etc.) to make it easier for the reader. Additionally, error bars should be shown for the measured concentrations (and a description of what the error bars represent, 1SD, 2SD, etc.). This would help justify the author's assertion that the KIRMAT simulations agree with measured concentrations.

*The conclusions have been slightly trimmed (lines 729-748), but it was found difficult to shorten them more. The figures 7 to 11 have been improved (better organization of axes and titles). The plots are now individually labeled (e.g., 7A, 7B, 7C, 7D) and quoted as such in the main text to improve readability (example lines 669, 676). Error bars are drawn in the figures and their interpretation is given in the captions.*

---

## Author Comment (AC2) · 2 Apr 2019

April 2d, 2019

To: Pr. Jan Seibert
**Handling Editor -HESS**

**Re: Answers to the reviewer comments: Manuscript HESS-2018-609** " Crossing hydrological and geochemical modeling to understand the spatiotemporal variability of water chemistry in an elementary watershed (Strengbach, France)" by Julien Ackerer et al.

Dear Editor,
 We appreciate the efforts the reviewer has invested in our manuscript, which resulted in a set of revisions. Please find attached the a pdf with the answers to the reviewers' comments in red, to distinguish them from the reviewers' comments. If required the revised manuscript with the changes marked in red can be send for your kind consideration.
 We are grateful to the reviewer for providing stimulating comments and advices to improve the initial version of the manuscript.

Yours Sincerely

François Chabaux, for the authors
University of Strasbourg
UMR CNRS 7517 – LhyGeS
1, rue Blessig
67000 Strasbourg – France

**Answer to Reviewers**

**Answer to Reviewer 2:**

**Anonymous Referee #2**

The study 'Crossing hydrological and geochemical modeling to understand the spatiotemporal variability of water chemistry in an elementary watershed' by Ackerer et al. presents results from coupled hydrological and geochemical reactive transport models of a small watershed. The authors find that observed 'chemostatic' behavior is controlled by seasonal patterns of subsurface transit times and suggest that simple geometric representations of mineral surface areas may be sufficient to match lab and field measured effective rates. While I find this study promising and potentially interesting to a broad community of researchers, I believe there are some significant issues with the

authors' methodologies and conclusions that merit major revisions or resubmission. It is possible that I have also misunderstood some of the technical issues, as the authors do not provide adequate description of the modeling and validation methodologies, as detailed by Referee #1.

First, I'd like to echo comments offered by Referee #1 — this manuscript needs a much more detailed description of the geochemical modeling including in depth descriptions of rate parameterizations, reaction pathways, kinetic rates, and equilibrium constants used. As it stands, it is hard to contextualize any of the presented results. Additionally, there have been other studies which combine fully distributed hydrologic models with geochemical reactive transport code that should be acknowledged.

The revised manuscript brings now more details on the modeling and validation exercises. The hydrological modeling and its validation are presented in lines 200-222, and a new table giving the initial and adjusted parameter values of the hydrological simulations has been added (table 2). Clay minerals and secondary phases are detailed in lines 255-262, lines 270-273, and lines 712-720. The whole geochemical database and the geochemical modeling parameters (minerals, reactive surfaces, thermodynamic and kinetic constants, clay solid solution end members, etc.) have been reassembled as supplementary material associated with the revised manuscript (tables EA10-EA13). We also refer to the recent work by Ackerer et al., 2018, concealing detailed information on the KIRMAT model (line 716).

Recent works coupling hydrological and geochemical models, including ParCrunchFlo (Beisman et al., 2015) and PHIM (Li et al., 2017) are now cited in the Introduction of the revised manuscript. The general content of these studies is presented at the end of the Introduction, which also allows us to better raise the points that give originality of our contribution compared with these previous works (lines 110-121).

Flowpath Modeling: From what I can piece together, the back-trajectory simulations provide subsurface flow path lines that all originate at the boundary of the watershed domain. If the hydrologic model simulates subsurface/surface water connections as described, why aren't flowpath origins distributed evenly across the land surface? It seems the authors provide a quick fix to this situation by assigning an even distribution of inputs along a flowpath, but this step and its necessity should be described in more detail. In particular, this situation is characterized as 'realistic' in the beginning of the discussion, but from my understanding, other codes such as ParFlow / SLIMFast back trajectories would not need this fix because they predict the origin of waters across the land surface.

The backtracking of a particle, which ended in coordinates ($x, y, z$) at time $t$ consists in following particles through flow fields with reversed directions. The stream lines drawn by this procedure indicate the time (prior to $t$) at which a particle passed at a given location or entered the flow field. In the case limiting the backtracking to various successive subsurface flow fields, no stream line will reach the surface, with the exception of distorted flow fields generated by a continuous massive infiltration at a few locations in the system. It must also be raised that the studied watershed with its steep slopes mainly generates gravity driven flow from top hills to valleys, which renders steam lines extending from the boundaries of the domain to the stream. To observe stream lines originating

from various locations over the watershed, one should proceed with a forward tracking (with the velocity fields in their actual directions) following particles initially spread over the surface of the system at a given time *t*, and then moved by the successive velocity fields posterior to *t*. Our Modeling tool NIHM allows for this forward tracking but the results are not that sought to condition geochemical interactions in the subsurface. We seek the various flow paths and associated travel time distributions in the subsurface that are collected at point locations in the domain where water chemistry is monitored. It is far easier to proceed with a backtracking technique initiating the (back) travel of particles from the sampled points and the sampled times of geochemical data.

Other codes such as ParFlow/SlimFast with a three-dimensional approach to the subsurface compartment could eventually backtrack particles up to the land surface in the case discussed above of a massive local infiltration area. In the case of gravity driven flow along slopes associated with widespread infiltration from the surface, almost all backtracked stream lines should also reach the boundaries of the system. That being said, it is right that backtracked stream lines, irrespective of the code employed, do not inform on the elementary water flux conveyed along each lines. They also do not distinguish between the time a particle passed at a given location and the time the particle entered in the subsurface at this location. Therefore, extracting transit time distributions from backtracked stream lines implies that these distributions are an approximation that gives exactly the same weight to all the times retrieved along all the stream lines. This approximation goes with uniform elementary water fluxes along each line and uniform inlets from the surface. Proceeding differently would imply to solve a transport equation in transient flow regime and for transient source terms, a cumbersome procedure probably resulting in as much uncertainty as backtracking for a poorly known hydrosystem only conditioned by the stream flow rates at the outlet of the system.

With regard to geochemical modeling, it is right that the calculations assume uniform inlets of soil solutions along the subsurface flowlines and spatially homogenous water-rock interactions within the watershed (rephrased in lines 307-310). These assumptions are supported by the good agreement between measured and modeled solute concentrations (this study), and the comparison between regolith profile data and water chemistry data (Ackerer et al., 2016).

Additionally, from the diagram in Fig 5, are the measured soil solutions taken as inputs into the modeling domain? If so, it would be extremely useful to see where these fall relative to the C-Q plots in Fig's 8 and 10. What is particularly important is seeing what soil concentrations are relative to precipitation concentrations. In other words, does a significant amount of solute generation occur in soils, and if so, is that being represented in the model at all? My understanding is that it is not, which could also be one of the primary causes of observations of chemostatic Na behavior at CS1 as mentioned in more detail below. What proportion of overall solute generation from precipitation to spring is missed by not representing soil processes?

The deepest soil solutions are that used as inputs into the modeled subsurface compartment. This is clarified in lines 284-287. The concentrations in soil solutions are presented in figures 7, 9, 11, and 12. In the case of the Strengbach catchment, soil solution concentrations are by far higher than precipitation concentrations, and the aim of this study is to grasp the hydro-geochemical processes that occur in the subsurface between soil solutions and spring or ground waters. Modeling what occurs between precipitations and soil solutions would imply to cope with another scale of

investigation, relying upon precise modeling (the cm scale, or less) of transient unsaturated flow coupled with reactive transport.

Since the soil solutions are injected into the modeled subsurface, the chemical temporal variability of these solutions and its impact on the water-rock interactions are accounted for in the modeling exercises. This point is better specified in lines 290-293. It is worth noting that the impact of the soil solution chemistry on spring waters is detailed in Ackerer et al., 2018.

Na concentrations are much higher in spring waters than in soil solutions (figures 7, 9). The Na concentrations in spring waters are obtained from a combination of an initial load in soil solutions, primary mineral dissolution (albite), and Na incorporation into clay solid solutions. There is no solute generation in spring waters missed by our approach, as the deepest soil solutions aggregate the surface processes, then they enter the aquifer, and water-rock interactions within the aquifer are modeled via KIRMAT.

Model Validation: As far as I can tell, the primary validation for the model is a qualitative matching of modeled v. observed solute concentration patterns across discharge. This seems fine in general to me (though would be strengthened by some level of statistical analysis); however, the implications of this matching appear oversold. Specifically, the fact that model results match observations only means that this particular combination of parameters and subprocesses (water transit times, mineral surface areas, assumed reaction networks and associated kinetic and thermodynamic constants) combine to get the right answer, but that does not validate each individual subprocess. In other words, there is the problem of equifinality: if the hydrologic model is consistently underestimating transit times, for example, then the model could still match results by overestimating dissolution/precipitation rates. Without some independent validation of water transit times (i.e. seasonal water isotope variability, tracers), the authors cannot conclude that each individual process is accurately represented. This issue needs to be discussed, particularly w/r/t Section 6. As I read it the model-data match is used to independently validate (1) water transit time simulations; (2) the fact that bedrock waters don't need representation; (3) mineral surface areas and kinetic rate constants; (4) the specific representation of clay solid solution series (which is not adequately described) — in my mind these conclusions are not sufficiently supported without independent validation of these sub-processes.

More information is now available regarding the hydrological model validation (lines 200-222). Regarding the solute concentrations, the validation is simply performed by checking that modeled concentrations are close to chemical data up to the addition of eventual analytical uncertainties on measures.

We underline that the hydrological modeling is performed independently and before any geochemical simulation. If the hydrological model were set up to significantly underestimate water transit times, a correct reproduction both in fluctuations over time and amplitude of the water flow rates at the outlet of the watershed would not be possible. The fact that the flow rates are well reproduced over the whole hydrological context between 2010 and 2015 supports the idea that the water transit times inferred by NIHM are realistic. This point is mentioned in lines 695-701.

The fact that the chemical composition of waters is well captured indicates that the combination of geochemical parameters and water transit times in KIRMAT is able to generate realistic reaction rates. This is rephrased in lines 683-686. We simply emphasize that non-modified kinetic constants and geometric reactive surfaces form a combination able to reproduce the evolution of water chemistry in the catchment (line 686-690). It is right that eventual equifinality between parameters in any model might result in multiple solutions fitting observed data. In that sense, we cannot assert that each single geochemical parameter is set up at its right value. But we support the idea that the inferred set of geochemical parameters produces realistic reaction rates, and is relevant for this kind of geochemical studies (lines 683-686).

We agree that a comparison of our results with water isotope variability could be interesting, even if the seasonal variability of isotopes content might be not accurate enough to be faced with the short transit times that characterize the watershed. It is reminded that the steep slopes of the subsurface compartment generate rapid and very transient flows mainly controlled by the variability over time of hydro-meteorological inlets and the wet versus dry character of some portions of the system. In any case this comparison is beyond the scope of our study, but a sentence has been added to suggest that seasonal variability of isotopes could be informative (lines 718-720).

Also echoing Referee #1, it would be useful to include more description of the C-Q dynamics – both in the introduction and in the analysis. Specifically, Si seems much less chemostatic than Na, particularly at CS1. Why is this and why does it also happen in the model? Are Na concentrations diluted until they reach observed soil concentrations (i.e. point above) in which case does its representation in the subsurface even matter?

The introduction has been modified to reintroduce this notion of concentration-discharge relationships (lines 60-69). We explain that these relationships are mainly qualitative and do not substitute to quantitative evaluations for which the coupling between hydrological and geochemical modeling is an important new step. This feature is also reminded later in the discussion (lines 660-666).

Na concentrations in spring and piezometer waters are higher than in soil solutions (see figure 7 and 9). Na concentrations in spring and piezometer waters cannot be explained by a simple dilution. A gain of Na load occurs in subsurface waters because of the dissolution of albite minerals. The relative response of each chemical element to a change in water discharge is also associated with the initial soil solution concentrations, the nature of primary minerals leached by waters and the degree of incorporation of the element in the clay solid solutions. Na is only provided by one single primary mineral (albite) and is weakly incorporated into clays. Si is provided by several primary minerals (albite, biotite, K-feldspar…) and strongly incorporated in clay minerals that are characterized by varying precipitation rates with hydrological conditions. These points explain why the response of solute concentrations to hydrological changes is variable for each element. This is now specified in lines 401-408.

Small point is that the authors consistently characterize precipitation and drought events as 'important' without context.

I also agree that figures would benefit from formatting to make them more readable – particularly y-axis labels of concentrations.

It is now clarified that these important flood and drought events are defined with reference to the maximum and minimum flow rates at the outlet of the watershed (line 346-347).

The figures 7 to 11 have been improved in the revised manuscript (better organization of axes and titles). The plots are now individually labeled (e.g., 7A, 7B, 7C, 7D) and quoted as such in the main text to improve readability (example in lines 684, 691). Error bars are drawn in the figures and their interpretation is given in the captions.

Lastly, and in general, these subsurface transit times seem relatively fast which may make sense for such a small catchment system. However, how does this compare to measured transit times in other catchment systems (for example as a function of watershed scale)? In other words, how applicable are the conclusions that chemostatic behavior here reflects far-from-equilibrium hydrologic controls to other areas?

Yes, the small size of the watershed, but mainly the small storage capacity of the aquifer (thickness less than 10 m) and the steep slopes generating gravity driven flows, induce short water transit times. This is rephrased in lines 370-371. The chemostatic behavior is in our study explained by short transit times, precipitations (and thus percolations of soil solutions) regularly distributed over time and space and few mixings of waters from contrasted origins. There is no reason to suggest that our conclusions could not extend to systems with similar dynamics. This is clarified in lines 725-727.

---

## Referee Report (RR1)

This work is of great interest and provides new insights in coupling hydrological and geochemical processes at watershed scale. The authors explore the development of a dimensionally-reduced model coupled with a reactive transport model. The modelling approach is validated by field data collected in springs and piezometers at wet and dry periods, which allows for the investigation of the spatial and temporal variability of transit times and reaction rates. The authors conclude on a hydrological control on the chemostatic behaviour of the watershed, due to fairly constant transit times despite highly varying flow dynamics. The conclusions of the paper thus strongly rely on the hydrological model simulating a low variability of water transit times, from which the geochemical model logically simulate a low variability of geochemical concentrations. To my point of view, a more detailed description of the hydrological model is needed to strengthen the confidence in the results (see specific comments). I recommend moderate revisions before publication of this interesting work, mainly because information is missing in the method section and because the study would gain from a more detailed analysis of the results.

Specific comments:

Figure 1. Please add a x scale.

l. 175 refer to Figure 1

METHODS

l.179 A lot of geochemical data seem available, both for springs and piezometers, but only some of them are used to validate the models. Why are you only using some of the available data? The model would be improved with a validation on all available data that span a long period of time. And if not, the authors should justify why they use only a limited part of their dataset and how they chose the data used.

l.200 "The exchange of water between the surface and subsurface flows are addressed via the hydraulic head differences between the compartments." A hydraulic conductivity value of the interface between the two compartments must be also considered. Which value was used? Was it calibrated or fixed? Please clarify.

The model parametrization paragraph needs clarification and additional precisions.

- "several zones of heterogeneity" please refer to the Figure
- Please specify all parameters instead of "other parameters" l.206 for clarity
- I guess the aquifer thicknesses given Fig. 2 correspond to the values obtained after the calibration. If so, please be specific on the figure or in the legend. Also add in the legend that the grey lines show the mesh grid of the hydrological model (if it is the case).
- The hydraulic conductivity is calculated using with the Van Genutchen model (which involves parameters such as the saturated hydraulic conductivity, n and alpha), and therefore is a result of the simulations at the grid size, is it correct? I don't think it is clearly said in the paper. As the temporal variability of the hydraulic conductivity is an important point of this study, I think a very clear explanation of how the depth-integrated hydraulic conductivity is calculated would make it easier for the reader to follow the logic. Fig.6, do the red colour (highest hydraulic conductivity) correspond to the value of the calibrated saturated hydraulic conductivity? If yes, it might be worth saying it, as it shows that the subsurface compartment is fully saturated at high flows.
- How is the calibration realized? Which algorithm was used? Could the authors add some uncertainty estimates on the calibrated parameters?

- The hydrological model over the whole catchment is calibrated only on the discharge time series located at the outlet of the catchment. Do the authors have other hydraulic data they could use to strengthen their calibration, such as piezometric heads or spring discharge rates? If not, I am worried that equifinality might not be negligible, which also points for a serious estimation of parameter uncertainties and/or sensitivity analysis (see previous comment).

The hydrogeochemical modelling strategy presented Figure 5 deserves more details. For each flow line, how much is "several"? What is the value of the "constant distance along the flow line" that is used? Do the boxes drawn Figure 5 correspond to the grid of the NIHM model or not? KIRMAT simulations were performed for different flow lines independently and then mixed at the outlet of the flow lines, if I got it correctly (Figure 5). The mixing could also occur all along the flow line, each percolated soil water mixing with the water coming from the upstream "box". Would the results be different? What is the justification for no mixing between flow lines within the subsurface compartment?

RESULTS

l. 320  Where are the results of the water velocities?

l. 328 Please specify which characteristics of the flow lines you consider similar (geometric characteristic? length, position…), as the velocity along the flow lines for instance differ from given dates (4.2). Maybe not talking about water velocity along flow lines in this paragraph might be less confusing.

l. 341 "or parameters" Parameters should not change under transient conditions as they have been previously calibrated, right?

l. 344 Please describe the driving factors of the spatial and temporal change in the simulated hydraulic conductivity. How much is this result related to the geometry of the watershed (small thickness, steep topography…)?

l. 385 refer to Figure 7

Figure 7. Where do the uncertainty bars come from in the KIRMAT simulated concentrations? Please, clarify what you mean by "induced" in the legend. A priori it could not be induced by the hydrological model as uncertainties are not taken into account. Is it only coming from the propagation of soil solution uncertainties?

Figure 8. The CS1 geochemistry was simulated at 6 different dates, but we cannot tell which of the observation blue point corresponds to each simulated orange point on the Figure. This information is needed to assess the reliability of the modelling. Maybe using different colours, please link each simulated point to the corresponding observation point.

Figure 9. and l. 403-405. The sentence is quite vague. The authors chose to present the differences between elements in a figure, which is very interesting. But then their explanation for the differences remain very vague and not specific to any element. Could the authors expand on the geochemical mechanisms yielding to the different C-MTT relations?

PZ3 and PZ5 piezometers. You already specify paragraph 4.2 that, such as for CS1 and CS3 springs, single flow lines can be used, so you can shorten the first part of the section. Maybe this section could be merged with the previous one as the approach and the results are similar.

CS2 and CS4 springs. Same comment as above, you can probably shorten the justification of the scattered flow line distributions as it is a repetition of paragraph 4.1

DISCUSSION

Have the standard kinetic constants used here been determined on minerals all coming from the Strengbach catchment? Then would you recommend to use site-specific kinetic constants to account for local aging effects?

I am not sure if Figure 13 brings anything new. It might be more interesting to show the distribution of dissolution rates over the catchment.

The discussion on the chemostatic behaviour would gain from a more concise and straightforward argumentation. Some repetitions with the result section could be avoided. For instance, the whole paragraph describing changes in the simulated hydraulic conductivities, in water velocities and in mean transit times should not be detailed as much or details should be moved to the result section 4.2. Lines 693-699 aim to justify the modelling approach and results (same holds true for the discussion on the clay solid solution), which could be moved elsewhere for clarity. I would advise to refocus the whole paragraph only on the discussion on the potential origins for the chemostatic behaviour.

---

## Referee Report (RR2)

**Crossing hydrological and geochemical modeling to understand the spatiotemporal variability of water chemistry in a headwater catchment (Strengbach, France)**

Julien Ackerer, Benjamin Jeannot, Frederick Delay, Sylvain Weill, Yann Lucas, Bertrand Fritz, Daniel Viville, François Chabaux

The authors addressed previous comments and answered most of my questions. In particular, the authors added an interesting focus on the implications of their work for the Strengbach watershed. I believe the manuscript has improved and can be published after few minor changes.

The modelling approach proposed by the authors is novel and very interesting, and has the great advantage of combining a low-dimension hydrological model with a reactive transport model. The description remains slightly confusing sometimes. Please consider clarifying the following points.

- Kirmat is derived from the Transition State Theory (l. 294) but are all chemical reaction (dissolution of primary minerals, oxido-reduction reactions and clay precipitation l.298) rates calculation based on the kinetic laws derived from the TST? Clays precipitated at the thermodynamic equilibrium (l. 307), therefore I would expect no kinetic laws. Please clarify if TST is only applied for mineral dissolution. Which oxido-reduction reactions are taken into account in this study?
- Clay precipitation rates are said to be realistic (l.768) but I could not find the rates in the paper. Maybe this could be added in the supplementary materials (in a table such as for the dissolution rates).
- Simulated chemical compositions are far from a state of chemical equilibrium with respect to primary minerals (l. 756). What would be the order of magnitude of water transit time (or chemical equilibrium length) to reach equilibrium with respect to primary minerals in the watershed?
- Back tracking was used to constrain the origin of subsurface water exiting the system at certain points, as simulated by the hydrological model (l.243), meaning that water originate from the tipping point of the flow line. However in the geochemical model, the authors consider water percolating all along the flow line. How can the back tracking be used to differentiate between water particle entering the subsurface or just passing at a certain point?

---

## Author Response (AR2)

Julien Ackerer                                          2019, October 6th
François Chabaux
Laboratoire d'Hydrologie et de Géochimie de Strasbourg
Université de Strasbourg-CNRS
rue Blessig - 67084 Strasbourg Cedex – France
fchabaux@unistra.fr

Pr. Jan Seibert
Handling Editor – Hydrology and Earth System Sciences (HESS)

Dear Editor,

The second revised version of the manuscript we have submitted for publication to HESS (Manuscript HESS-2018-609) has been uploaded with this file.

Please find below the answers to the reviewers comments, written in green to distinguish them from the comments, along with a marked-up manuscript version showing the changes made compared to the first revised version (changes marked in green).

We thank the reviewers for the comments and suggestions, made on our manuscript. We hope that the new version sent with this letter and the different answers to the reviewer's comments make the manuscript suitable for publication in HESS.

Yours Sincerely

François Chabaux, on behalf of the authors.

**Reply to Reviewers – Second round of revisions**

**Reviewer 1:**

Ackerer et al. coupled the catchment hydrology model NIHM with the geochemical model KIRMAT, enabling the much-needed connection between hydrological processes and geochemical reactions. Such connection is particularly important as the hydrology and biogeochemistry fields advance to resolve pressing issues at the interface of water quantity and quality. In addition to the model development, the authors also validated the model with field data, and explored the connections between transit time distribution and reaction rates, another important missing link. While I applaud these novel aspects of the work, there are issues that need to be addressed before publication.

First, with the advanced modeling tool and particle tracking technique that quantifies the travel time, the scientific results from this work appear weak. In particular, the conclusion that "durations of water rock interactions exert a first order control on the chemical composition of waters and that the acquisition of the water chemistry can be explained by weathering processes that are spatially fairly homogeneous over the catchment." This seems a fairly obvious conclusion that does not need a coupled watershed modeling tool. In fact, the dependence on residence time and the closely related concept of Damholker number have been discussed extensively in existing literature. See, for example, (Maher, 2010; Wen & Li, 2018) and the literature therein.

We acknowledge that the ideas of a first-order control on water chemistry by water transit times, and occurrences of fairly homogeneous weathering processes over small catchments are not particularly novel in the context of hydrogeochemical modeling. We simply checked that these hypotheses were valid at the Strengbach catchment. It remains interesting to check that the chemical composition of the spring waters, irrespective of their locations over the system, respond to a homogeneous scheme of water-rock interactions in a shallow regolith (less than 8 m thickness). This leads to the conclusion that eventual circulations of deep-water in the fractured bedrock do not control the spring chemical composition in the watershed. This is an important result even though it can be specific to this type of widespread environments (small catchments of mid-mountain countries). That being said, the revised manuscript better mentions the previous studies that contributed to assess these hypotheses of timely-controlled and homogeneous weathering processes (lines 588-590).

We defend however that the novelty of our study lies in the dynamic of the flow pathways in the subsurface between wet and dry conditions, this dynamics limiting the overall variability of water transit times. The variability of flow and associated water transit times is mainly explained by the seasonal fluctuations of water contents and hydraulic conductivities in the watershed (lines 352-359).  Translating the flow dynamics into mean times of water-rock interactions and introducing these times into a reactive transport model allows for simulating the observed geochemical signatures of waters. To our knowledge, it is the first time that such hydrological control is shown via a complete modeling of the flow dynamics over a watershed and subsequent analysis of water transit times. These points are highlighted in lines 654-664. We also bring new insights on the understanding of element-specific C-Q relationships (C=concentration, Q=flow rate), linking the C-Q shapes with the trends in the C versus transit time relationships (lines 665-685, and the new figure 9).

We also defend that our approach is able to successfully couple hydrological and geochemical processes in an innovative way, involving depth integrated, dimensionally reduced hydrological models and limited computation times. We recall this point in lines 115-122.

The authors also concluded that "… the chemostatic behavior of the water chemistry is a direct consequence of the strong control exerted by hydrological processes on water transit times." This very general statement does not provide specific insights on how water transit time control water chemistry. We all know that transit time controls water chemistry. But how and via what mechanism it leads to chemostatic behavior? Transit time influences dissolution rates but that do not necessarily would have chemostatic behavior. This needs to be better explained and discussed from view point of how processes occurred. The real strength of process-based model like the one presented here is its power in linking observations to processes and mechanisms.

It is commonly agreed that water transit times partly controls water chemistry. Previous studies that contributed to generalize these hypotheses are now cited lines 588-590. The innovative part of our work is precisely to show why transit times span a much narrower variation range than water discharges in the catchment (lines 352-370). Modeling spatial and temporal variability of the flow patterns in the system renders information on the origin of the water feeding sampling points and how this origin evolves with the hydrological conditions. During wet periods, faster flow occurs along extended flow lines active from the valley up to the crests limiting the watershed. During dry periods, slower flow occurs along shorter flow lines that only stretch between the mid-slopes and the valley. This covariation between flow velocity and flow length attenuates the overall variability of water transit times, which in turn, results into a stable geochemical signature of waters. This point is rephrased in lines 654-659.

Second, despite of the comments from previous reviewers about earlier work, the authors are still not up to speed about literature. For example, the authors still state that (in Line 186 – 187) "To the best of our knowledge, this is the first Time that such a coupling between hydrological and hydrogeochemical modeling approaches has been attempted at the watershed scale." This work is obviously NOT the "first time" such coupling has been done, as the previous reviewers have pointed out. Although Beisman et al (2015)'s model does not consider the surface hydrology processes and temporal dynamics and therefore is not strictly a watershed scale hydrological and biogeochemical model, RT-Flux-PIHM (Bao et al., WRR, 2017; Li et al., WRR, 2017) is a coupled hydro-biogeochemical model with the relevant surface hydrology processes. They in fact have a string of new papers out based on this model, including (Wen et al., 2019; Zhi et al., 2019). In that context, the authors really cannot claim the "first time". Even not being the first time doing this, this paper is still valuable and publishable. It is better not to claim "the first time" when it is not. The series papers from RT-Flux-PIHM did not link water transit time with geochemical reactions, which may be the angel that the authors CAN claim as the major novelty of this work.

We agree that our statement was not precise enough. Our work is obviously not the first-one trying to couple hydrological and geochemical models. It is the first attempt coupling a dimensionally reduced hydrological model with reactive transport. The reduction of dimensionality (in the subsurface compartment of the watershed) and the introduction of transit times (instead of velocity fields) in the transport problem are at the origin of a strong reduction of computation costs. The methodology still captures the flow dynamics, water transit times, and chemical variability over an entire watershed. The rewriting (lines 182-187) in the revised version clarifies this feature and also includes new references (e.g. lines 58-63), even if some of them were not published at the time of the submission of our manuscript (December 2018).

Third, although a major focus is on concentration discharge (CQ) literature, the authors seem not aware of the most recent CQ literature. For example, (Musolff et al., 2017) explored relationships between travel time and emergent CQ patterns, although they used a different approach for quantification of travel time. Zhi et al (2019) showed that the contrasts between shallow and deeper water composition governed CQ patterns, which in fact can explain the chemostatic behavior observed in this work, as the dissolving minerals are homogeneously distributed here (if I understand correctly). It would be meaningful and increase the readership of the paper if the authors can discuss results from this work in the context of previous topics on similar topics. Other relevant papers include, for example, (Diamond & Cohen, 2018; Herndon et al., 2018; Musolff et al., 2015).

We updated the references, especially in the introduction (lines 58-63). A new discussion is proposed to compare our interpretations with previous studies (lines 708-723). We propose an alternative to the conclusions by Zhi et al.,2019 who evoked that the chemostatic behavior resulted from the mixing of various sources of water (superficial soil water vs groundwater). In the Strengbach catchment, recent studies indicate that both the spring and stream waters show a completely different geochemical signature than the deep groundwater sampled in the fractured bedrock (Chabaux et al., 2017). In addition, deep groundwater is not an important contribution compared with shallow groundwater regarding the feeding of springs and streams (Pierret et al., 2018). The soil solutions sampled in the watershed are also very different from the spring and stream waters (Prunier et al., 2015). It is not necessary to mix soil and deep groundwater to generate chemostasy. As we show, chemostasy can result from water percolation into a single regolith layer with homogeneous mineralogy if the dynamic of flow lines limits the overall water transit time variability (lines 654-659). Musolff at al. 2015, associated the chemostatic behavior of subsurface water with the accumulation of a large mass of weathered material. We add to this the need for relatively stable water transit times through the various hydrological periods experienced by the watershed (lines 709-712). We defend that chemostasy is not explained (in the studied context) by a modification of the reactive surface of minerals in the subsurface (i.e., Clow and Mast, 2010), or by the absence of chemical contrast between slow and rapid flows (i.e., Kim et al., 2017). The precipitation of clay minerals is essential to correctly capture the water chemistry in the Strengbach watershed, but the dissolution or re-dissolution of clays is not a key process to explain chemostasy (i.e., Li et al., 2017). Our results clearly support the idea that a spatial variability in the flow paths compensated by the variability of fluid velocities is the key process to explain the chemostatic behavior (Herndon et al., 2018). In addition, our approach quantifies the mean transit times and the seasonal variability of water chemistry. We updated this discussion in lines 708-723.

I generally believe that more insights can be gained via more detailed analysis. For example, when and where the dissolution rates are highest in the watershed at dry and at wet times?

The pattern of dissolution rates for primary minerals is mainly controlled by the variability over time and space of the flow lines. During wet conditions, the maximal dissolution rates of primary minerals occur on the upper parts of the catchment. During dry conditions, the dissolution rates are maximal at mid-slopes, as the upper parts of the catchment are simply dry. This information has been added in the revised manuscript, lines 599-604.

In addition to the spatial patters of conductivity, can you show rates or concentration spatial patterns over the entire watershed?

Rates or concentration patterns are not available over the entire watershed, simply because the modeling approach, that renders the calculations tractable and with results that can be faced with data, focuses the simulations of reactive transport along the flow paths feeding each sampling site (lines 276-278). Nevertheless, we added a new figure in this revised version to exemplify the changes in concentration values as a function of water transit times (new figure 9). We also updated the discussion to clarify how the relationships between concentration and transit times are useful to capture the C-Q patterns of the dissolved elements (lines 665-685).

I also think the writing of the manuscript can be improved by being more specific and concise. Some of the discussion appears lengthy and diffusive. For example, 6.1 and 6.2.

We slightly reduced the discussion in sections 6, 6.1, and 6.2. Some repetitive paragraphs have been removed in the revised manuscript. However, Section 6.3 has been slightly extended to better compare our results and interpretations with previous studies (lines 708-723).

Detailed comments:

title: what is "elementary" watershed? it seems an unusual name.

By 'elementary' we mean here headwater watershed. We use the term headwater watershed in this new version to make this clearer (ex: in the title).

Introduction: Motivation for coupling is still not strong. It can use literature review of CQ to motivate the need of coupling, as previous reviewers have pointed out.

New additional references have been added in the introduction to better argue on the clear need today for coupled approaches modeling hydrology and geochemistry (lines 58-68). We quote recent studies to motivate some developments that would open the "black box" of the concentrationdischarge (C-Q) relationships. Emphasis is also put on better deciphering the variability of flow paths and transit times over water catchments (lines 63-68).

Line 62, Ameli et al 2017 is not at the watershed scale. it is hillslope scale.

This is corrected and outlined (line 56).

Line 85-87: please define "depth-integrated models". Do you mean there is no resolution in the vertical direction and there is only one grid in the vertical direction?

The notion of low-dimensional hydrological model is here associated with a normal-to-bedrock integration of the subsurface compartment of a watershed. The subsurface is therefore represented by a single two-dimensional grid (lines 231-233). NIHM, the model employed in this study, is of the type integrating the subsurface. It has been developed and tested in Pan et al., 2015; Weill et al., 2017; Jeannot et al., 2018. We mention (lines 189-197) these contributions for further details on how the model is built and then calculated, when the present study only reminds us on the type of model employed.

Line 103 – 107: do you need "if" at the beginning of the sentence. Reads awkward. Can be separated into 2 sentences.

This sentence has been rephrased for clarity (lines 111-114).

Line 114 – 115: please define "dimensionally-reduced" here and later in approaches. What specific did you do? again related to the "depth-integrated". Are these two terms equivalent here? if so, maybe stick to one term? Do you mean you only have two grids in vertical direction with unsaturated and saturated zones?

Dimensionally-reduced and depth-integrated are almost synonymous; it is because the hydrological model NIHM is depth-integrated for its subsurface compartment that it becomes dimensionally-reduced. Specifically, after integration NIHM solves a two-dimensional Richards equation valid for both the vadose and the saturated zones merged together into a single layer. This is better specified in the revised version of the manuscript, but the interested reader is referred to founding papers (Pan et al., 2015; Weill et al., 2017; Jeannot et al., 2018) for details. Notably, after a rapid discussion on the depth-integrated and dimensionally-reduced nature of NIHM, we stick to the term "dimensionally-reduced" when needed.

Line 202-245: this is a rather long paragraph. I suggest separate the particle track part as a separate paragraph and with its own subtitle (starting from line 222). this would help give attention to this important section.

We increased the length of this paragraph in answer to the questions raised by a previous round of review. This is why this new version provides more details on the hydrological model calibration and validation.

Line 222: are there existing references for backtracking approach? does figure 3 indicate that the backtracking only tracks through flowing water? what about the areas that were dry and disconnected at dry times and reconnected to the stream at a later time. Are mineral phases homogeneously distributed? I think they are. Please be explicit.

The backtracking approach is directly performed from the outputs of the NIHM simulations, and reverting the direction of the transient flow velocity fields.  If an area initially dry is not connected to the stream (or to any other location when delineating flow lines feeding a sampling point) at the reference time where the particles are launched, it does not mean that the area is definitively unseen by the tracking. If the area has been wet prior to the reference time, with non-negligible velocities, particles moved backward in time and located at the boundary of the area will pass through the wet area. Incidentally, for a dry area, the particles would wait at the boundary of the area with null velocities. In that sense, backtracking also tracks non-flowing water of temporarily dry areas. It must be understood that backtracked particles render flow lines conditioned by an exit point at a given time of reference. If there is no flow at this point and at the time of reference, the particle will stay immobile, simply increasing its duration of stay in the system. When flow occurs (more precisely, flow occurred prior to the reference time) the particle moves backward. The same comment applies to a particle located at a given time along a streamline. It either wait at this location or move backward according to the current water velocity field.
To answer the question about mineral phases, the mineralogy of the regolith is assumed homogeneous. This is specified in line 293.

Section 3.2: it sounds like the concentrations are calculated based on TST rate law. but the description also sounds like the calculation is based on travel time. The TST itself does not have a time component to take into account of travel time. so I am confused about how exactly the rates were calculated. Please clarify.

The concentrations are calculated with the reactive transport code KIRMAT, which is a thermo-kinetic code based on the TST rate law. The concentrations are not calculated with the transit times. Our discussions would be meaningless if the concentrations were directly calculated from transit times. The rates of primary mineral dissolution are calculated following Ackerer et al., 2018, eq.1 and Ngo et al., 2014, eq.1. These calculations involve reactive surfaces, thermodynamic and kinetic constants, and the calculation of a distance from chemical equilibrium. The transit times are only constrained by the hydrological simulations rendered by NIHM, and before any geochemical simulation. This is why we defend that water discharges and transit times are constrained independently and before dealing with water chemistry, leading to the discussions developed in section 6.

Figure 7, 11: these bar figures are unnecessary and not effective. Why not plot lines for conc vs time for different solutes? you can still add the measurement data for comparison. It would be nice to also include conc. vs. mean travel time, as they may reveal different trend as conc vs discharge.

It is useful to show somewhere how the simulated concentrations are matching data. The bar figures allow for the comparison between model results, soil solutions, and measured concentrations in water for five elements in a single plot. However, we agree that a plot of concentrations vs mean transit times could be useful. This type of plot has been added in the new Fig. 9. We also added in Fig. 8 a few plots increasing the number of elements discussed in terms of the C-Q relationships. Y-axes in Fig. 8 have also been re-scaled to better visualize the different trends between the elements. Finally, We also provide more information on how the trends in concentration vs mean transit time impact the C-Q relationships of the elements (lines 665-685).

518-523: "give weight to" were used for a few times. Suggest rephrasing
The discussion about geometric surface area is applicable together with lab-measured reaction constants may need further consideration. The text describing Table 3 emphasizes similar geometric surface area and BET surface area. But the geometric surface area has a large range, often by orders of magnitude. if one takes log average instead of arithmetic, the geometric surface area is in fact much lower than the BET surface area, which mean a much lower surface area is needed to reproduce the concentration data. This in fact is consistent with many previous studies showing that lower surface area needs to be used in order to directly use TST rate law at the field scale, see for example (Heidari et al., 2017; Moore et al., 2012).

We shortened this section and removed the repetitive sentences (lines 513-516). Our calculations of the geometric surfaces in dissolution experiments indicate that the geometric and BET surfaces are less different that we could think. We agree that low values of reactive surfaces are necessary to reproduce realistic water chemistry. But we show that the simple and raw geometric surfaces are low enough to generate realistic water chemistry when implemented in reactive transport approaches (lines 568-574).

657-659: "the study of concentration discharge relationships has been intensively used to assess the chemostatic behavior of waters (Godsey et al., 2009; Kim et al., 2017; Ameli et al., 2017)." Oddly phrased sentence. Please rephrase.

This sentence was repetitive and has been removed from the revised manuscript.

**Reviewer 2:**

Dear authors,

The overarching framework of this contribution is the need for developing innovative hydrogeochemical modelling approaches for the watershed scale, able to integrate both the complexity of water flow paths and the diversity of water-bedrock interaction processes. The presented work is novel and timely in the context of existing literature on this topic. The submitted work is equally well aligned with the scope of the journal.

As a way forward, the authors propose – like other recent studies – to merge hydrological and geochemical codes. Instead of solving fully dimensioned problems, they here explore an alternative avenue, consisting of a so-called dimensionally-reduced approach with modest computational needs (even when applied to an entire watershed). Their aim is to model the spatial and temporal distribution of water flow paths (or trajectories), weathering reactions and the subsequent evolution of water chemistry. The authors expect to improve their knowledge of water flow paths – including their variability – between wet and dry seasons, as a prerequisite to better assess water transit times and eventually to better understand water chemistry dynamics.

While the overall goal and proposed approach are highly valuable, this contribution could have its impact certainly increased by further highlighting and leveraging the more than three decades of research into hydrological processes in the Strengbach catchment. While mentioned throughout the manuscript, many of the historical investigations in the area of interest are not presented clearly enough as contributions to what certainly is a textbook case of an evolving perceptual model of a key experimental watershed in critical zone research. By developing (even briefly) in the introduction on what is the perceptual model of fundamental catchment functions of water (and matter) collection, storage, mixing and release in the Strengbach catchment, the authors could build a much stronger case (e.g. in the discussion and conclusion) for how their (unquestionably) important work is a major milestone towards improving the understanding of water chemistry dynamics – in the Strengbach watershed and elsewhere.

The introduction is already relatively long, and we preferred to introduce our study under the perspective of understanding the concentration-discharge relationships in watersheds. As our methodology is generic because physically-based models are applicable to various contexts, we are inclined to keep an introduction that is not too much site-specific. The introduction is modified to give a better overview of previous studies in this field of research (lines 58-63). We acknowledge that we can also refer to the synthesis work of Pierret et al., 2018 in the introduction, where an update of the general comprehension of the watershed can be found This is done line 129. We take the opportunity of this review to highlight the renewed perception of the Strengbach catchment, stemming from the ability to model the complete watershed on its hydrological behavior from surface to subsurface over large periods of times (section 6.2). This modeling exercise concluded that the active system extends from soil to a shallow regolith, which results in low storage and rapid water transit times that should renew the view on how surface and subsurface waters acquire their geochemical signature.

A new improved perceptual model – leveraging the work presented in this contribution – could (for example) be introduced in the discussion. Since building on previous hydrological and geochemical modelling work, the authors rather marginally develop on the modelling concepts and mostly refer to existing literature. For aspects related to model parameterization, choices are backed in most cases by references to past investigations and/or field data. However, a major problem for the hydrological modelling in the Strengbach watershed is related to the fact that no O and H isotope data in precipitation and stream water is available. Therefore, mean transit time estimations obtained through the hydrological model cannot be validated by experimental data. While certainly not perfect, a (simple) way to consolidate the outcome of the hydrological modelling could consist in comparing the MTTs obtained through the NIHM model to calculations of the hydraulic turnover of the watershed.

It is right that the methodological aspect of this study is mainly a work crossing information provided by previously developed models KIRMAT and NIHM (Ngo et al., 2014; Ackerer et al., 2018; Pan et al., 2015; Jeannot et al., 2018). We acknowledge that further samplings, like O and H isotopic data, or cosmogenic isotopes in stream water, could be very interesting to obtain new constrains on transit times. But, in the same way as for hydrogeochemical modeling, interpretations of isotopic data are not straightforward, and do not give direct estimates on transit times.

It is not totally true that the transit times are not validated by the data, because matching water discharges at the outlet for any hydrological conditions implies that the inferred transit times are realistic (section 3.1). Unrealistic transit times would over-predict or under-predict the measured flow rates at the outlet, which is not the case in figure 2.

Other (albeit more sophisticated and time-consuming) approaches rely on time-variant transit time concepts (e.g. Hrachowitz et al., 2016; https://doi.org/10.1002/wat2.1155). Surprisingly, the range of MTT values is 1.5 to 3 months in the abstract, while in section 4 the range given for MTTs is 'approximatively 1.75 to 4 months between the strongest flood and the driest conditions.' These MTTs also differ slightly from those given for the same watershed by Pierret et al. (2018; 'The Strengbach Catchment: A Multidisciplinary Environmental Sentry for 30 Years'; Vadose Zone Journal): 100 to 200 days (i.e. ~3 to 6 months). While not contradicting per se, these MTT values nonetheless are not totally coherent and should be homogenized in the contribution, if not discussed at some point in the manuscript for if and why they differ from previous work.

In our approach, the transit times are also 'time' and 'seasonally' variant, due to the dynamic of the flow lines. For example, in Fig. 8B we determine the variance of mean transit times for the CS1 spring water between dry and flood conditions.

Relying upon the analysis of time-varying transit time and residence time distributions in the watershed mainly finds out its worth in purely hydrological applications, for example by assessing how young and old waters contribute to feeding a stream, etc. In our study, the point is to extract transit times of water that feeds a sampling point at a prescribed sampling time. There is no reason to duplicate calculations of transit times, even though we did so at the margins of this study to check that the distributions used in the geochemical modeling were not outliers.

These small differences in inferred transit times are simply explained by the different locations (exit points) where the transit times are calculated. Sites at high elevations have shorter transit times compared to sites at lower elevations. Values between 1.75 and 3 months are valuable for the CS1 spring, which is located on the upper part of the catchment (figure 1). Values between 1.75 and 4 months are given for all the springs and piezometers in this study (lines 368-370). The values given in Pierret et al., 2018 are average values at the outlet of the watershed, for which transit times are slightly longer.

Another important point is how the results of this contribution relate to the non-stationarity documented in Pierret et al. (2018) for various parameters in the Strengbach watershed. They have described long-term (i.e. over several decades) significant increasing and decreasing trends in pH and sulfate concentrations in precipitation, spring water and stream water in the Strengbach watershed. As an example of this non-stationaity, mean annual pH in precipitation still remains lower than at pre-industrial levels. In the Strengbach watershed, stream water pH is strongly correlated to precipitation acidity. The slope of pH vs. time for precipitations is more than twice that for stream water, suggesting that some protons are neutralized during their transfer through soils, saprolite and bedrock via exchanges and mineral weathering processes. Pierret et al. (2018) also hypothesize long-term signal fluctuations being dampened during this transfer process – translating in lower standard deviations at the watershed outlet, confirming the neutralisation reactions. Alongside other examples of past research carried out in the Strengbach watershed (e.g. on the interception process and how tree species influences chemical concentrations of atmospheric inputs reaching soils, on the reported 7-year periodicity in precipitation and outflows, on a reported increasing difference between annual inflows and outflows over the past 3 decades). Discussing the results of this contribution in the light of three decades of research in the area of interest and especially how the proposed combination of hydrological and geochemical modelling may help anticipating non-stationarity in hydrological catchment functions of water and matter collection, storage and release in the Strengbach watershed would certainly contribute to further increase the impact of the presented findings.

A previous work by Ackerer et al., 2018 emphasized on the long-term evolution of water chemistry, and its links with superficial perturbations, pH changes, and non-stationarity in the watershed. We definitely agree that some trends recorded in the atmospheric depositions and in the soil solutions are dampened during water transfer within the regolith. This is reported in Pierret et al., 2018, and also in Ackerer et al., 2018, where modeling results showed that the spring chemistry was affected by changes in the soil solution chemistry. The study of Ackerer et al., 2018, was also aimed at the multi-year evolution of spring chemistry and the non-stationarity of $Ca^{2+}$ vs the relative stationarity of $H_4SiO_4$ concentrations. This is why we prefer here to discuss in more details the other new implications of the study. That being said, we expanded the discussion to better compare our results with previous studies (lines 708-723). We also provide new insights on how the specific trends of concentration versus mean transit times are impacting the C-Q relationships of the elements (665-685).

I hope that these suggestions – for what I consider minor changes – will help to further improve what I consider as a very interesting and valuable contribution.
Best regards,
Laurent Pfister

**Reviewer 3:**

General Comment: I commend the authors for their efforts to address the recommended revisions by both myself and Reviewer #2. Clearly, the authors addressed the hydrological concerns with a lot of thought and effort. I particularly appreciated the authors' incorporation of a geochemical database and modeling parameters that are included in the supplementary material of the modified manuscript.

However, after this first round of review I still have some concerns on the geochemical portion of the model and its reproducibility. In particular, the saturation states for the primary and secondary minerals are not presented in the current manuscript. Additionally, the specific clay phases precipitated are not identified (i.e kaolinite vs. smectite, etc.). Without this information, which was requested in the last round of reviews, it's hard to justify the authors claims that only secondary clays were precipitated (and not other phases) as stated in Lines 270-273 and the representation of clay solid solution series. In general, treatment of the secondary phases still needs to be more adequately described. This is rather crucial for interpretations on C-Q relationships for Na and Si. If these issues can be addressed in the next round of revisions then I believe the manuscript will be suitable for acceptance.

A lot of details about the geochemical model KIRMAT are available in the quoted references (Tardy and Fritz, 1981; Gérard et al., 1998; Ngo et al., 2014; Lucas et al., 2017; Ackerer et al., 2018). It is right that the saturation states of primary and secondary minerals were not provided. A new table is now available as supplementary material (table EA14) with the log(Q/K) mean values for the primary and secondary minerals along a typical flow path in the watershed. We also provide typical values of dissolution or precipitation rates for the minerals in table EA14. In the work by Ackerer et al., 2018, the spring chemistry was already correctly captured relying upon the precipitation of clay minerals, but without any precipitation of calcite, hematite, amorphous silica…

The chemical compositions and the thermodynamic data for the clay solid solution end members are available in table EA13 (supplementary material). We remind this point in this revised manuscript, and we explicitly mention the different clay end members in lines 254-256. Only a clay solid solution is precipitated at the thermodynamic equilibrium in our approach, but the composition of the solid solution evolves over time and space along a flow line (see figure 7 in Ackerer et al., 2018). We also recall that all the thermodynamic and kinetic data for the primary minerals are given as supplementary material in tables EA11 and EA12 (lines 248-250).

We provide a new figure (Fig. 9) in the revised manuscript to better highlight the differences in the trends of concentration vs mean transit time for the different elements. We also added a few plots in Fig. 8 to compare modeled and measured C-Q for more chemical elements. Y-axes are also re-scaled to better visualize the different trends between elements (figure 8). It is also better explained that the species with significant slopes in the plots concentration vs transit time are slightly chemodynamic (Fig. 9, H4SiO4 and secondary Na+) while the species with flat slopes are almost perfectly chemostatic (K+ and Mg2+, lines 665-685). These points explain the differences of element-specific C-Q relations, while the overall modest variability of water transit times explains the weak variability of concentrations over the catchment.

Specific Comments:

Lines 50-52 (Abstract): I'm not sure "originality" is a good word to start this sentence. Also it's still not clear what is original about this study: is it the approach? – i.e. using a hybrid hydrological + geochemical model to characterize a mountainous catchment? Or is it the hybrid model itself? – i.e. the fact that its "dimensionally" reduced and thus unique from other hybrid models?

This sentence has been rephrased for clarity. The novelty of this study is to couple hydrological and geochemical modeling approaches to understand the spatiotemporal variability of the water chemistry in a small watershed with low dimensional approaches and fairly short computation times (lines 115-122). The innovation is also to provide new constraints on the flow line variability between dry and wet seasons, water transit times, and understanding of C-Q relations. These points are developed in the section 6.

Lines 57-60 (Introduction) C-Q relationships are not qualitative. How they are used and interpreted can be qualitative. In this regard, I'm assuming that the authors were speaking specifically about their interpretations based on simple 3 end member mixing relationships (Hornberger et al. 2001). But, C-Q relations themselves show how measured, dissolved solute concentrations evolve with measured changes in discharge. The goal is to explain these observed C-Q relations or, in other words, the observed behavior of dissolved solute concentrations with discharge in watersheds. This is the motivation for using hybrid hydrological and geochemical models – to properly constrain important parameters such as water transit times and chemical reaction times that influence the overall fluid chemistry in order to accurately predict C-Q relationships. This is what was missing in the introduction and I highly suggest that this be rewritten to emphasize this point considering that C-Q relationships represent a large portion of the results and discussion section.

We agree that this sentence was unclear and not precise enough. The introduction has been modified to better quote recent studies and better define the overall framework of our study (lines 58-63). More details are given to introduce the C-Q relationships, and to motivate the evaluation of the transit time variability as a key feature to understand these C-Q relationships (lines 63-68).

Lines 114-115 (Introduction): Nit-picking here, but what is meant by a "dimensionally-reduced" approach and how does it differ from a "fully dimensionalized" approach? I'm assuming that this is referring to 2-D vs 3-D approaches, but deserves a clear explanation.

We mean here by "low-dimensional" that the hydrological model simulates both the vadose and saturated zones of the subsurface as a single two-dimensional layer in which appropriate integration over the direction normal to bedrock reduces the dimensionality of flow and of the computation grid. NIHM is a low- dimensional hydrological model that has been developed and tested in Pan et al., 2015; Weill et al., 2017; Jeannot et al., 2018. We mention in lines 189-197 these studies where all the characteristics of NIHM can be found. NIHM solves subsurface flow via an integrated Richards equation (manipulating parameters integrated along the direction normal to bedrock) over a single two-dimensional grid. NIHM was shown to reproduce the main results of fully dimensioned (3-D for the subsurface) hydrological models, while significantly reducing computation times (Pan et al., 2015; Weill et al., 2017; Jeannot et al., 2018). We cite these specific publications for the readers who are not used with this notion of low-dimensionality in hydrological models (lines 189-197).

Lines 267-268 (Hydrogeochemical modeling): How is the reactive surfaces of the primary minerals tracked? What was actually tracked- changes in the specific surface area (SSA, m2/g) or the bulk surface area (BSA, m2mineral m-3porous media)? Is a shrinking sphere model used (an example can be seen in eqn.3, Navarre-Stichler et al. 2011)? What assumptions are made in the calculation? This needs to be outlined in detail, especially considering that (1) estimating the SA evolution is generally challenging and there are several approaches that can be taken, each with their own assumptions, (2) the authors conclude that "changes in the reactive surfaces of primary minerals became negligible" (line 270) and (3) one of the principal conclusions is centered on surface area estimations.
I highly recommend including in the appendix how the model calculates the evolution of the surface area and present the associated parameter values.

In the KIRMAT model, the reactive surfaces are expressed in m2/kg H20, and it is a shrinking sphere model that is used to track the evolution of the reactive surfaces (Schaffhauser, 2013; equation IV-71). The calculation considers the competition between mineral dissolution and porosity evolution through time. It is clear that the evolution of reactive surfaces for the dissolving primary minerals is negligible in this study, simply because the timescale is very short. For example, reactive surfaces of albite vary between 12.92 m2/kg H20 and 12.91 m2/kg H20, and reactive surfaces of biotite vary between 10.23 and 10.22 m2/kg H20 throughout the geochemical modeling exercise. Reactive surfaces can evolve significantly at a millennial timescale for these types of primary minerals, but not for simulations at a monthly or yearly timescale. This is explained in lines 267-269. This is why it is not very useful to give a lot of details on this calculation, because reactive surfaces nearly invariant in our study.

Lines 270-273 (Hydrogeochemical modeling): It's unclear what is meant here by "hydroclimatic context" in regard to the precipitation of secondary minerals other than clays. It's the saturation state with respect to these secondary minerals specifically that dictates whether they form or not. This sentence should be changed to something along the lines of: "these secondary phases were not formed based on calculated saturated states…" Along these lines, how is the saturation state calculated in KIRMAT? A quick statement to address this would suffice. What are the calculated saturation states for both the primary minerals dissolved and clays precipitated? Also, unless I am missing something, the type of clays precipitated is not described in the paper. Was the clay mass fraction mostly kaolinite? Or some mixture with other commonly formed clay phases like smectite? This information must be provided in the manuscript so that this model is reproducible. A good place to put it would be in Table EA11. Otherwise the interpretations on the geochemical variability in the fluid phase as well as tracking clay mass fractions as solid solutions are unsubstantiated. It's possible that these concerns are all considered in the model, but the point is that the authors do not provide any descriptions of these important chemical parameters and thus makes it hard for the reader to follow the logic of the paper.

We changed this sentence to specify that these secondary phases were not formed based on the calculated saturation states (lines 269-272). The saturation states are calculated with the ratio log (Q/K), with Q the ion activity product of the minerals and K the thermodynamic equilibrium constants of the dissolution reactions. These terms are explained in Ngo et al., 2014 and Ackerer et al., 2018 and we quote these studies. We provide in the new table EA14 the values of log(Q/K) for the primary and secondary minerals tested in precipitation. The end members of the clay solid solution and the mean log (Q/K) values of the end members are given in table EA13. The clay solid solution is made of Illite and Montmorillonite, these types of clay having been observed by XRD analysis of bedrock and regolith samples (Ackerer et al., 2018). Kaolinite is not revealed by the XRD analysis. The chemical composition of the clay solid solution (the fraction of the different end members) is changing over time and with the distance along a flow path (details in Ackerer et al. 2018; figure 7).

Lines 642-643: Chemostatic behavior in this study hasn't yet been justified. I would suggest validating chemostatic behavior using a power-law relationship to fit the C-Q relationships in log-log space and reporting the value of the exponent, "b" (Godsey et al. 2009). Generally, a b close to zero indicates chemostatic behavior in the catchment.

We agree that it is useful to provide the coefficients 'a' and 'b' of the C-Q relationships to quantify the degree of chemostatic behavior. We provide in the new caption of Fig. 8 the 'a' and 'b' coefficients of the expression $C=aQ^b$ for the different elements. We also provide new insights on the link between the concentration vs transit time slopes and the C-Q shapes for the different elements in the new Fig. 9. We updated part of the discussion to better explain these findings in lines 665-685.

Figure 8: I would suggest having the data points color coded according with time in a gradient that goes from oldest (2005) to youngest (2015) with the date of each sampling point presented in the legend. This would help the reader understand the evolution of the C-Q relationships with time and help the reader follow the results (lines 647-650) for dissolved Si and Na behavior.

A color code varying for times between 2005 to 2015 would be confusing for most of the species, because within each year there exist periods of low and high water discharges, resulting in color coded dots simply scattered all along the C-Q relations. But we updated Figs 8 and 9, and the associated discussion to make clearer our findings on the specific control of the C-Q shapes for the different species (lines 665-685).

References:

Schaffhauser, T. (2013). *Traçage et modélisation des processus d'altération à l'échelle d'un petit bassin versant, le Ringelbach (Vosges, France)* (Doctoral dissertation).

[revised manuscript text omitted]

---

## Author Response (AR3)

François Chabaux                                               2020, February 12th
Laboratoire d'Hydrologie et de Géochimie de Strasbourg
Université de Strasbourg-CNRS
rue Blessig - 67084 Strasbourg Cedex – France
fchabaux@unistra.fr

      Pr. Jan Seibert
      Handling Editor – Hydrology and Earth System Sciences (HESS)

      Dear Editor,

      The new revised version of the manuscript we have submitted for publication to HESS (Manuscript HESS-2018-609) has been uploaded with this file.

      Please find below the answers to the reviewer comments along with a marked-up manuscript version showing the changes made.

      We hope that the new version sent with the letter and the different answers to the reviewer's comments make the manuscript suitable for publication in HESS.

      Yours Sincerely

François Chabaux, on behalf of the authors.

**Reply to Reviewers – Third round of revisions**

**Reviewer 1:**

The authors have comprehensively addressed previous comments. in particular, the manuscript has been improved with more precise language about novelty of the work, in-depth discussion on concentrations, rates, and transit time, and the positioning of this work in the context of other existing work. I support the publication of the manuscript.

**Reviewer 2:**

General Summary: This study presents the application of a coupled hydrological—reactive transport model capable of characterizing monthly to annual scale hydrogeochemical variability in Strengbach, a small granitic headwater catchment in France. The model proposed is composed of both a depth-integrated and spatially-distributed NIHM hydrologic model and a Kinetic Reaction and Mass Transport KIRMAT reactive transport model. Although there are several combined hydrologic + geochemical models that have been developed in the literature, this particular model has some unique features that make it quite appealing in application to watershed scales. One feature in particular is the low-dimensional approach to the hydrologic model, which consists mainly of combining vadose and saturated zones into a single subsurface compartment that can be modeled as a simple 2-D layer. Reducing the "dimension" of the hydrologic model has significant advantages in that it reduces the complexity of solving numerically for both unsaturated and saturated flow and the computational costs. This is not the first study to take such an approach, but it is novel in that it's one of the first studies to combine this "non-dimensionalized' hydrologic model with a fully realized reactive transport model. This model was further validated with field data where independently constrained water transit times through the hydrological simulations were used to accurately predict observed geochemical variability.

General Comments: This last round of revisions has shown large improvements to the overall quality of the manuscript and I thank the authors for their efforts. In light of the corrections that have been made by the authors, I believe that this article is now suitable for publication granted that certain minor modifications are made beforehand. I suggest the authors try to concentrate more on what these model simulations tell us about the catchment dynamics in Strengbach and how this build on findings from other studies of this catchment. As one of the other reviewers noted, Strengbach is a well-studied catchment and, thus, articulating how the results from these novel hydrological + reactive transport models provides a (large?) step forward in our understanding of the Strengbach catchment in particular would boost the impact of this paper in my opinion. While I think it is good to explore what these findings might mean at a global scale or in comparison to other similar catchments, it shouldn't be the centerpiece of the present conclusions; rather it should be used as motivation for conducting similar types of studies in other catchments in the future.

We updated the section 6.2 discussing the implications of this work for the Strengbach watershed in particular (lines 641-689). More information is available on how our results propose a new step to validate several hypotheses made by previous studies conducted in the Strengbach watershed (i.e. Viville et al., 2012; Pierret et al., 2014; 2018; Pan et al., 2015; Ackerer et al., 2016; 2018; Chabaux et al., 2017; Weill et al., 2017). The modeling work first emphasizes the key role of times of water-rock interactions (lines 647-661). The modeling work also reveals that the spatial distribution of the weathering processes is relatively homogenous within the catchment (lines 662-672). This latter point explains the previous observations of similar chemical fluxes in the stream, springs and regolith profiles (lines 669-673). In addition, our results bring the important conclusion that the hydrogeochemical functioning of the watershed is properly simulated by water circulations in the very shallow subsurface (saprolitic aquifer, lines 674-689). The contribution of waters circulating in the deep fracture network of the granitic bedrock is not necessary to explain both the dynamic of the stream and the chemical composition of springs (lines 674-689). The independent flow paths feeding the springs are also confirming hypotheses from previous isotopic studies (lines 683-689). Finally, section 6.2 underlines how this study improves the understanding of the Strengbach headwater catchment.

However, we believe that it is important for the conclusion of the manuscript not being too much "Strengbach centered", with the meaning that many headwater catchments could behave as we show. This incline us to let room in the conclusions for general implications that our study suggests (lines 787-807).

Additionally, I found quite a bit of spelling and grammar mistakes as I was reading through this latest version of the manuscript. There are also areas where I still believe the writing can be paired down. I strongly urge the authors to address these issues before publication.

We corrected the last mistakes and reduced slightly the text where it was feasible (e.g., section 6.2). Some repetitions between sections regarding "Results" and "Discussions" have been removed (e.g., between sections 6.2 and 4.2, lines 712-724).

Specific Comments:

Lines 36-37: "explains why transit times span much narrower ranges of variation than that water discharges." Does this statement apply to Strengbach specifically or is this a more general observation? Please provide citations for the latter.

This finding is a result from our modeling work and is specific to the Strengbach watershed. This is clarified in line 40. The point is that flow over the catchment is mainly controlled by gravity (the slopes) irrespective of the local pressure head gradients. As an example, 1 or 7 m of water-saturated thickness (and therefore of pressure head) in the system does not really influence water velocity when the elevation gradient is of 30%. This renders water velocities almost invariant over time, with pressure heads simply being an indicator of stored water. Water storage highly fluctuates, with consequences on volumetric fluxes and on instantaneous stream discharges, when water velocity fluctuations are smooth.

Line 50: remove "the" in " effects of the ongoing climatic changes…"

Done line 52.

Lines 87-89: How the flow is treated in a dimensionally reduced hydrologic model should be quickly added here. Even the explanation that was provided to one of my comments in the last round of revisions is suitable as shown below:
"NIHM solves subsurface flow via an integrated Richards equation …"

We added this information in the text lines 91-94.

Lines 607-610: Is the model able to track the areas of maximal clay formation in the catchment and how this varies between wet and dry seasons? This would be very useful information to complement the predicted areas of max dissolution rates proposed by the model.

Yes, the model is able to track the clay formation rates and the clay compositions along the flow lines. Clay formation rates are slightly higher at the upstream tips of the flow lines, where the dissolution rates of primary minerals are also higher. With the seasonal variability of the extend of flow lines (flow lines extending up to the crests during wet periods and retracting to middle elevations during dry periods), the clay formation rates are maximal near the crests during floods but maximal at middle elevations during dry periods. We added this information lines 469-475. We moved this part dealing with primary mineral dissolution and clay formation rates in the section on results (lines 469-475), as this result is a direct conclusion of the modeling task.

697-699: Yes, but only on a monthly to annual timescale resolution. It would be interesting to see if this model can replicate observed water chemistry in more high frequency daily or hourly geochemical datasets during major flooding events.

Yes, the modeling exercise in this work is definitely dealing with monthly timescale data, even though hydrological simulations, due to the variability of rainfalls events, infiltration and stream flow velocities sometimes use very short time steps (sometimes less than 15 min). Testing our modeling approach with high frequency geochemical data would be an interesting new step for the future. But this exercise is currently infeasible because hourly collected geochemical data for the Strengbach spring waters are lacking.

Line 713: Be specific in regard to the explanation of this thermodynamic equilibrium, which is (I think) the primary mineral dissolution reaction. I think I found this elsewhere throughout the manuscript.

We underline here the difference between the chemical equilibrium state of the water as in the sentence: " It is also important to emphasize that the simulated chemical compositions of waters remain far from a state of chemical equilibrium with respect to primary minerals."(lines 756-757), and the way to handle precipitation of clay minerals in the KIRMAT code as in "a clay solid solution precipitated at the thermodynamic equilibrium is able to generate reliable water chemistry (this study) and realistic clay precipitation rates (more detail in Ackerer et al., 2018)" (lines 765-768).

The precipitation of a clay solid solution at thermodynamic equilibrium means that no kinetic data are used for the clay solid solution end members. Only the thermodynamic constants log(Ki), log (Qi/Ki) are used for the precipitation of the clay solid solution end members (supplementary table EA13).

With these model properties, spring waters remain far from a state of chemical equilibrium with respect to primary minerals, resulting however in correctly captured geochemical composition of spring waters (lines 765-768). The overall clay precipitation rates are also realistic compared with the determination of clay mass fractions in regolith samples (lines 765-768, details in Ackerer et al., 2018).

Lines 727-729: "…spatial and temporal variability in flow paths is a key process to explain C—Q relations" I would add for this particular catchment.

We added "in this type of headwater catchment" in line 783. Our results support that this finding is probably applicable for other headwater catchments in a similar context.

Lines 748-752: "…good estimate of the reactive surface within the natural environment" is a little too broad. While I agree that this geometric approximation of the minerals is most likely a better approach in the models, I don't think you can generalize these conclusions to the "natural environment". In this case, for this particular catchment, these estimated low surface areas along with the rate constants did provide good replications of the observed geochemical data. This might also be the case for other similar granitic headwater catchments, but not perhaps the case for volcanic or karstic systems.

We agree, "natural environment" is a probably a too large perspective. We replaced "natural environment" by "in this type of granitic catchment" to be more specific (lines 805).

Figure 8: The axes for the H4SiO4, Na+, K+, and Mg2+ should really be shown here in mmol L-1, rather than mol L-1 that way it's easier for the reader to see the variability.

All our figures are using mol/L, making that we prefer to keep the axes in mol/L for consistency. But we updated the variation ranges of the axes to improve readability (figure 8).

**Reviewer 3:**

This work is of great interest and provides new insights in coupling hydrological and geochemical processes at watershed scale. The authors explore the development of a dimensionally-reduced model coupled with a reactive transport model. The modelling approach is validated by field data collected in springs and piezometers at wet and dry periods, which allows for the investigation of the spatial and temporal variability of transit times and reaction rates. The authors conclude on a hydrological control on the chemostatic behavior of the watershed, due to fairly constant transit times despite highly varying flow dynamics. The conclusions of the paper thus strongly rely on the hydrological model simulating a low variability of water transit times, from which the geochemical model logically simulate a low variability of geochemical concentrations. To my point of view, a more detailed description of the hydrological model is needed to strengthen the confidence in the results (see specific comments). I recommend moderate revisions before publication of this interesting work, mainly because information is missing in the method section and because the study would gain from a more detailed analysis of the results.

We updated this new version of the manuscript and we provide more details on the hydrological modeling (section 3.1). In particular, the way the depth-averaged hydraulic conductivity is calculated is detailed in lines 267-273. We also remind that the topic of the present work is not aimed at hydrological modeling with low-dimensional approaches, and that specific publications dealing with the features and the development of the hydrological model NIHM are quoted in lines 202-204 (i.e., Pan et al., 2015; Weill et al., 2017; Jeannot et al., 2018). We reorganized parts of the manuscript to slightly reduce the text when feasible, and we moved a few parts from the section devoted to discussion to the section on raw results for clarity (lines 712-724, section 6.2 to 4.2). A deeper interpretation of the implications associated with our main findings concerning the Strengbach catchment is also proposed in lines 641-689.

Specific comments:

Figure 1. Please add a x scale.

Done in figure 1.

l. 175 refer to Figure 1

We refer to figure 1 line 185.

METHODS

l.179 A lot of geochemical data seem available, both for springs and piezometers, but only some of them are used to validate the models. Why are you only using some of the available data? The model would be improved with a validation on all available data that span a long period of time. And if not, the authors should justify why they use only a limited part of their dataset and how they chose the data used.

The overall geochemical database for springs and piezometers is available in supplementary tables EA1-EA9, and the modeled samples are given in table 1. The caption of table 1 has been updated to clarify that the table only reports on samples used for the NIHM-KIRMAT modeling task. It is not possible to run the NIHM-KIRMAT coupled approach on the overall and vast geochemical database (see supplementary table EA1-EA9) because of computation time (hundreds of samples). In addition, trying to exploit the whole database, would probably blur the message conveyed by this study and stating that a reliable but parsimonious geochemical modeling can rely upon a lose coupling with a hydrological model via streamlines and water transit time distributions. It can also be argued that the geochemical database, started 30 years ago, as most long-term databases started monitoring things without prior knowledge on how the monitored system was working! The authors of the present contribution sorted the data according to their revisited interpretation of the catchment.

That being said, for each date, 10 flow lines are backtracked per site by the code NIHM. After this step, a significant number of KIRMAT runs are performed to generate the mean water chemistry delivered by each flow lines. This approach is possible for a reasonable number of interesting dates and samples, not for all the database including hundreds of samples. Rather than modeling all the samples, we selected a reasonable number of dates that are covering the whole range of hydrological conditions in the catchment (table 1).

l.200 "The exchange of water between the surface and subsurface flows are addressed via the hydraulic head differences between the compartments." A hydraulic conductivity value of the interface between the two compartments must be also considered. Which value was used? Was it calibrated or fixed? Please clarify.

It has been clarified in line 209-213 that water exchange between the surface (streams) and the subsurface compartments depends both on the thickness and the hydraulic conductivity of the interface layer. This interface layer has the classical physical meaning of riverbed layers that are not the aquifer on the one hand, and not the free flowing water of the river on the other hand.

The hydraulic conductivity of the interface has been calibrated. The feature has been added in the form of values reported in table 2 and mentioned in the text in line 214-226.

The model parametrization paragraph needs clarification and additional precisions.
- "several zones of heterogeneity" please refer to the Figure

Done in lines 220-221.
- Please specify all parameters instead of "other parameters" l.206 for clarity

Done in lines 223-226.
- I guess the aquifer thicknesses given Fig. 2 correspond to the values obtained after the calibration. If so, please be specific on the figure or in the legend. Also add in the legend that the grey lines show the mesh grid of the hydrological model (if it is the case).

Done in figure 2 caption.

- The hydraulic conductivity is calculated using with the Van Genutchen model (which involves parameters such as the saturated hydraulic conductivity, n and alpha), and therefore is a result of the simulations at the grid size, is it correct? I don't think it is clearly said in the paper.

It has been emphasized in lines 391-396 that the hydraulic conductivity is calculated in NIHM (thus, at the so-called grid-size, by relying upon the empirical Van Genuchten equation to define the effective conductivity compared with its saturated upper bound).

As the temporal variability of the hydraulic conductivity is an important point of this study, I think a very clear explanation of how the depth-integrated hydraulic conductivity is calculated would make it easier for the reader to follow the logic. Fig.6, do the red colour (highest hydraulic conductivity) correspond to the value of the calibrated saturated hydraulic conductivity? If yes, it might be worth saying it, as it shows that the subsurface compartment is fully saturated at high flows.

A short explanation regarding the inference of the depth-integrated hydraulic conductivity has been added in lines 267-273.

The red color in figure 6 indicates that the depth-integrated hydraulic conductivity is higher than 6.1E-5, while the saturated hydraulic conductivity of most of the catchment is 8E-05. This also means that areas at 6.1 E-5 are saturated over almost the whole local thickness of the aquifer. This has been specified in the main text when discussing Fig.6 (lines 400-403).

- How is the calibration realized? Which algorithm was used? Could the authors add some uncertainty estimates on the calibrated parameters?

More precisely, the procedure relied upon a simple Monte-Carlo approach (detailed lines 214-2328) testing various configurations of the system because automatic inversions for integrated hydrological models are not available for the moment (though some advertisements are available in the literature, but not followed by effective inversions exercises). NIHM is in the process of being associated with adjoint-state calculations to perform automatic inversions via descent-direction methods and multi-scale parameterization. A task not straightforward at all when catchments react very differently to subsurface flow, surface routing and diffuse runoff according to their geometrical, geological, topographical and meteorological settings. After multiple exploratory calculations, it was shown that that the most sensitive parameters were the depth of the substratum, the saturated hydraulic conductivity, and the porosity. This is why these are the only parameters that were not defined as uniform values over the catchment. Uncertainty estimates on model parameters are unavailable, or more precisely, would be flawed by the Monte-Carlo approach. Each solution does not converge the same way, and bounds on parameters mix models that have not been conditioned the same way. Getting estimates of parameter uncertainties via Monte Carlo approaches would require converging algorithms as for example Monte Carlo Markov Chains (MCMC). Unfortunately, these algorithms applied to a complex system would also require more than 100,000 direct simulations for being statistically meaningful, something infeasible for the moment.

- The hydrological model over the whole catchment is calibrated only on the discharge time series located at the outlet of the catchment. Do the authors have other hydraulic data they could use to strengthen their calibration, such as piezometric heads or spring discharge rates? If not, I am worried that equifinality might not be negligible, which also points for a serious estimation of parameter uncertainties and/or sensitivity analysis (see previous comment).

There are no other continuous discharge rate measurements at the catchment for the simulated period (period which goes with that of available geochemical data on outcropping springs), but there are some boreholes. Unfortunately, these boreholes have been drilled deep enough (60 m) to intercept a few fractures in the bedrock (under the substratum of the shallow subsurface aquifer, made of soil plus saprolitic rock, simulated by NIHM). This renders the water levels monitored in these open boreholes unable to reflect hydraulic pressure heads in the active shallow porous aquifer of the catchment simulated by NIHM. This has been emphasized in lines 274-289. Geochemical investigations also show that deep borehole waters are not connected with effective subsurface flows feeding the springs and streams (Chabaux et al., 2017; Pierret et al., 2018).

Equifinality is the curse of any modeler, irrespective of the mass of data to condition the model. In some cases, increasing the mass of data can also favor equifinality. In the present case, flow over the system is mainly constrained by the steep slopes (i.e., elevation gradients) of the catchment, thus rendering the water saturated thickness mainly as an indicator of storage and not of water velocity (via pressure head gradients). There obviously exist equifinalities in the model proposed, but they hardly affect water velocities which are very similar for various model configurations, provided those configurations generate transient water storage compatible with the good fitting of the stream flow rate. We are now in the process of completing the third round of review, with more than 6 different reviewers involved, and 18 months past initial submission! We did not stay arms down during that period and tried to improve the hydrological conditioning by introducing local MRS (magnetic resonance sounding) data. Those are sensitive to the vertical distribution of water contents in the subsurface. Introducing these data slightly modified model outputs mainly on the storage capacity distribution (porosity of active aquifer layers) but did not change the transit and residence time distributions and their weak variability over time. A paper has been published on the topic (Weill et al., 2019, Water 2019, 11(12), 2637; https://doi.org/10.3390/w11122637 ), but we ignore whether or not we can mention it, as it was proposed and accepted after submission of the present work. That being said, a few words have been added in the manuscript to better emphasize the feature that transit times (which are here the main hydrological output for geochemical modeling) are mainly conditioned by the steep slopes of the catchment (lines 274-289). The paper in Water is also quoted (line 761, but can be removed on request). A conditioning on MRS data (Weill et al., 2019, water content distribution) slightly modified the model but without incidence on the overall distribution of flow paths, their variability, and the associated transit time distributions.  This is more precisely emphasized in lines 274-289 but can be removed on request.

The hydrogeochemical modelling strategy presented Figure 5 deserves more details. For each flow line, how much is "several"?  What is the value of the "constant distance along the flow line" that is used?

We used a constant space step of 1m along the flow lines. This distance refers to the distance between regularly distributed inlets along a stream line where the soil solutions percolate into the aquifer. This information is given lines 333-335. 'Several' therefore refers to a variable number of simulations, as this number is specific to each flow line. For example, a flow line with a length of 100 m is discretized into 100 cells of 1 m. 100 KIRMAT simulations will be performed along the flow line as illustrated in figure 5. The integrated chemistry of waters at the sources is the arithmetic mean of all the solute concentrations given by the 100 KIRMAT runs.

Do the boxes drawn Figure 5 correspond to the grid of the NIHM model or not?

No, the boxes in Fig. 5 represent the grids of the reactive-transport code KIRMAT. But 1D simulations with KIRMAT are performed along the flow lines previously determined by the hydrological model NIHM (lines 327-329). We updated the captions of figures to make this all clear.

KIRMAT simulations were performed for different flow lines independently and then mixed at the outlet of the flow lines, if I got it correctly (Figure 5). The mixing could also occur all along the flow line, each percolated soil water mixing with the water coming from the upstream "box". Would the results be different?

Both types of mixing are employed, a mixing of water chemistry simulated at the outlet of independent flow lines (as the consequence of diverse flow paths feeding the outcropping sources) and a mixing as presented in Fig. 5. The latter states that along a single flow line, water chemistry is the mix of water that entered the subsurface system (i.e., the flow line) at various points upstream. Because each flow line is associated with a 1m space step injection locations, one can consider that the whole process corresponds to percolations along the whole flow line. The point is that the water flux entering at each location along the flow line is unknown. Calculating at the outlet of the line an arithmetic mean is equivalent to state that percolation of soil water is uniform along the line. This assumption is supported by the modeling results but also by observations of similar geochemical fluxes from stream, springs and regolith profiles (lines 662-673, Viville et al., 2012; Ackerer et al., 2016; 2018).

What is the justification for no mixing between flow lines within the subsurface compartment?
With the physics of Darcian flow in continuous media, the stream lines cannot intersect. Mixing of solutes between lines could only occur by diffusion, which is undoubtedly less efficient in an advection-dominated problem than the spreading generated by taking means of ten independent stream lines (and which could also correspond to solute spreading due to heterogeneous velocities and flow paths).

RESULTS
l. 320 Where are the results of the water velocities?

Results of the mean water velocities are presented figures 7A, 7B, 10A, 10B, 11A and 11B. We also describe these results in section 4.2.
l. 328 Please specify which characteristics of the flow lines you consider similar (geometric characteristic? length, position…), as the velocity along the flow lines for instance differ from given dates (4.2). Maybe not talking about water velocity along flow lines in this paragraph might be less confusing.

We detailed that for the sites located on linear or slightly convex slopes, all the characteristics (geometry, flow rates, transit times) of the different flow lines that feed each site are comparable for a given site and for a given date (lines 378-380).

l. 341 "or parameters" Parameters should not change under transient conditions as they have been previously calibrated, right?

Yes, we removed 'or parameters' because this expression was confusing. We mean here: hydraulic variables (lines 392).

l. 344 Please describe the driving factors of the spatial and temporal change in the simulated hydraulic conductivity. How much is this result related to the geometry of the watershed (small thickness, steep topography…)?

This point is detailed lines 416-429. Spatial variability of the saturated hydraulic conductivity is associated with zones of various thicknesses (some degree of alteration in the saprolitic aquifer) and that fact that low storage capacity, no contributive zones, and rapid draining downward are conducive to dry crests with smaller effective conductivities. Variations over time are mainly associated with the water content in the system. With NIHM calculating effective mean conductivity values over the aquifer thickness, temporal variations of the effective conductivity depend on both hydro-meteorological forcing and the geometry (mainly the thickness) of the aquifer layer.

l. 385 refer to Figure 7

Done line 452.

Figure 7. Where do the uncertainty bars come from in the KIRMAT simulated concentrations? Please, clarify what you mean by "induced" in the legend. A priori it could not be induced by the hydrological model as uncertainties are not taken into account. Is it only coming from the propagation of soil solution uncertainties?

Yes, the uncertainty bars take into account the propagation in the KIRMAT simulations of analytical uncertainties from pH and chemical concentrations measured in the soil solutions. This is now clarified in the figure captions.

Figure 8. The CS1 geochemistry was simulated at 6 different dates, but we cannot tell which of the observation blue point corresponds to each simulated orange point on the Figure. This information is needed to assess the reliability of the modelling. Maybe using different colors, please link each simulated point to the corresponding observation point.

The figure 8 is already relatively heavy and we avoided to overload it with additional color codes. Instead, we added in the caption of table 1 the information relative to the different dates used for the hydrogeochemical modeling. In the caption of figure 8, we refer now to Table 1 for the dates, chemical data and water discharges of the modeled samples. We also refer in the caption to the supplementary material EA1 for the overall geochemical data. A visual comparison between simulated and measured concentrations is also available in figures 7C, 7D, 10C, 10D, 12A, 12B and 13.

Figure 9. and l. 403-405. The sentence is quite vague. The authors chose to present the differences between elements in a figure, which is very interesting. But then their explanation for the differences remain very vague and not specific to any element. Could the authors expand on the geochemical mechanisms yielding to the different C-MTT relations?

This point is clearly detailed in the discussion section in lines 725-746.

PZ3 and PZ5 piezometers. You already specify paragraph 4.2 that, such as for CS1 and CS3 springs, single flow lines can be used, so you can shorten the first part of the section. Maybe this section could be merged with the previous one as the approach and the results are similar.

We slightly reduced the text in the section 4.2.

CS2 and CS4 springs. Same comment as above, you can probably shorten the justification of the scattered flow line distributions as it is a repetition of paragraph 4.1.

We also slightly reduced the text in this part.

DISCUSSION

Have the standard kinetic constants used here been determined on minerals all coming from the Strengbach catchment? Then would you recommend to use site-specific kinetic constants to account for local aging effects?

No, the standard kinetic constants are coming from studies dealing with minerals from various origins (see table EA12 in supplementary material and references therein, lines 299-300). All these minerals are from natural rocks and were collected in the field, but are not from the Strengbach catchment. Simulations performed with these kinetic constants are able to capture spring chemistry while respecting water transit times, this is why we conclude that relevant aging effects are potentially included in these kinetic constants (lines 628-632).

I am not sure if Figure 13 brings anything new. It might be more interesting to show the distribution of dissolution rates over the catchment.

The figure 13 simply brings a general overview of modeled concentrations along an elevation transect PZ5, PZ3 and CS1 in the watershed. It is not possible to show the distribution of dissolution rates over the catchment, precisely because dissolution rates in our parsimonious geochemical modeling approach are only determined along the backtracked flow lines reaching the sampled sites. That being said, these sites are representative of the various flow patterns over the system, and their variability over time.

The discussion on the chemostatic behaviour would gain from a more concise and straightforward argumentation. Some repetitions with the result section could be avoided. For instance, the whole paragraph describing changes in the simulated hydraulic conductivities, in water velocities and in mean transit times should not be detailed as much or details should be moved to the result section 4.2.

Yes, we reduced slightly the text in this section (lines 712-724). We reduced the discussion section dealing with the general chemostatic behavior and moved some text and examples in the results section 4.2.

Lines 693-699 aim to justify the modelling approach and results (same holds true for the discussion on the clay solid solution), which could be moved elsewhere for clarity. I would advise to refocus the whole paragraph only on the discussion on the potential origins for the chemostatic behaviour.

Here, we underline that the choice of the clay solid solution and the question of the distance to chemical equilibrium for waters are key points in discussing the origin of the chemostatic behavior (see the discussion section 6.2 in Ackerer et al., 2018). The choice of clay minerals and the way to handle clay precipitation is major regarding the acquisition of the water chemistry (see discussion in Godderis et al., 2006; Maher et al., 2009; Ackerer et al., 2018 for example). Clay minerals and clay precipitation rates play a role in controlling whether or not chemical equilibrium occurs along the flow lines, and thus, if chemostatic behavior is explained by chemical equilibrium or not. This is why highlighting the relevance of our clay mineral assemblage and clay precipitation rates is important in this section concerning the chemostatic behavior (lines 763-768). It is worth noting that, within another round of reviews, one of the reviewers asked us to be very picky regarding the two points mentioned above.

References:

[revised manuscript text omitted]

---

## Author Response (AR4)

François Chabaux                                                      2020, May 1st
Laboratoire d'Hydrologie et de Géochimie de Strasbourg
Université de Strasbourg-CNRS
rue Blessig - 67084 Strasbourg Cedex – France
fchabaux@unistra.fr

Pr. Jan Seibert
Handling Editor – Hydrology and Earth System Sciences (HESS)

Dear Editor,

The 4th revised version of the manuscript submitted for publication to HESS (Manuscript HESS-2018-609) has been uploaded with this file.

The answers to the reviewer comments along with a marked-up manuscript version showing the changes made are given below.

Yours Sincerely

François Chabaux, on behalf of the authors.

**Reply to Reviewers – 4th round of revisions**

**Reviewer 1:**

The authors addressed previous comments and answered most of my questions. In particular, the authors added an interesting focus on the implications of their work for the Strengbach watershed. I believe the manuscript has improved and can be published after few minor changes.

The modelling approach proposed by the authors is novel and very interesting, and has the great advantage of combining a low-dimension hydrological model with a reactive transport model. The description remains slightly confusing sometimes. Please consider clarifying the following points.

- Kirmat is derived from the Transition State Theory (l. 294) but are all chemical reaction (dissolution of primary minerals, oxido-reduction reactions and clay precipitation l.298) rates calculation based on the kinetic laws derived from the TST? Clays precipitated at the thermodynamic equilibrium (l. 307), therefore I would expect no kinetic laws. Please clarify if TST is only applied for mineral dissolution. Which oxido-reduction reactions are taken into account in this study?

In the hydrogeochemical model KIRMAT, the calculation of the dissolution rates of primary minerals is based on the Transition State Theory (see equation 1 in Ackerer et al., 2018, equation 1 in Ngo et al.,2014 for example). For the secondary minerals presented in Table EA14, and tested in precipitation, the precipitation rates are also derived from TST (equation 2 in Ngo et al., 2014). By contrast, clay minerals are described as a solid solution of clay end members precipitated at thermodynamic equilibrium. Therefore, precipitation of clay minerals is not described by a kinetic law. The precipitation of the clay solid solution depends on the solubility product (K) of the clay end members (Tardy and Fritz, 1981; Fritz, 1985; equations IV-25 and IV-26 in Schaffhauser, 2013): the amount of a given clay mineral precipitated at any step of the simulated reaction is calculated to maintain the chemical equilibrium from the moment it is reached in the geochemical reaction. We clarified these points lines 299-304, 310-323.

 KIRMAT includes the oxido-reduction processes of iron (Fe), sulfur (S) and other important species for the corrosion of iron (Fritz, 1981; Marty et al., 2010; Ngo et al., 2014). Oxido-reduction reactions are also involved in the formation of several secondary minerals tested in precipitation in this study (Hematite, Siderite, …). Oxido-reduction reactions are handled through Nerst equations in KIRMAT (Fritz 1981, Made et al., 1994, Gerard et al., 1998). We clarified these points lines 299-304.

- Clay precipitation rates are said to be realistic (l.768) but I could not find the rates in the paper. Maybe this could be added in the supplementary materials (in a table such as for the dissolution rates).

We provide the clay 'precipitation rate', or the amount of precipitated clays, in the text lines 780-782. Details on the calculation of clay precipitation and comparison with field observations are available in Ackerer et al., 2018 section 6.2. We refer to this study in lines 780-782.

- Simulated chemical compositions are far from a state of chemical equilibrium with respect to primary minerals (l. 756). What would be the order of magnitude of water transit time (or chemical equilibrium length) to reach equilibrium with respect to primary minerals in the watershed?

These points were explored and detailed in Ackerer et al., 2018 section 6.2. A water transit time around 8-12 years and a distance as long as 15-20 km are required to reach a chemical equilibrium between water and primary minerals. This long equilibrium length is explained by the precipitation and the dynamic behavior of clay minerals removing ions from solution and retarding chemical equilibrium with respect to primary minerals. We give additional information on these points in lines 773-777.

- Back tracking was used to constrain the origin of subsurface water exiting the system at certain points, as simulated by the hydrological model (l.243), meaning that water originate from the tipping point of the flow line. However in the geochemical model, the authors consider water percolating all along the flow line. How can the back tracking be used to differentiate between water particle entering the subsurface or just passing at a certain point?

The sentence line 243 was inaccurate. Back tracking was not used to constrain the origin of subsurface water exiting the system at certain points. Back tracking was used to identify which subsurface flow lines reach the sampled sites. This point is rephrased lines 242-243.

With a backtracking technique, the delineated streamlines are conditioned by an "exit" point at a prescribed time. In any case, all locations and times upstream the exit point along a stream line cannot distinguish between the fact that the water parcel (the particle) entered the system (here the subsurface) at a location $x$ at time $T$-$t$, or simply passed through $x$ at $T$-$t$. This renders transit time distributions calculated via a backtracking procedure slightly biased in comparison with distributions calculated via a forward tracking (Weill et al. 2019, the paper is quoted in the manuscript). For getting an equiprobable transit time T-t for particles entering the subsurface in x, or simply crossing x at T-t, it must be assumed that source terms from the surface to the subsurface are evenly spread over the watershed and continuous over time. This is the case for the Strengbach catchment, thus rendering times T-t calculated via backtracking significant of the time spent by a water parcel entered at a location x and exiting the system (or collected in the system) at T. These transit times are also those relevant to geochemical reactions between water and the solid phase in the subsurface. The caption of figure 4 has been slightly modified in order to make this clearer (lines 1071-1073).

We also corrected that the springs collected in this study are not emerging naturally on the slopes (lines 157-158). These springs are captured for drinkable water supply directly in the subsurface by small collectors (Ackerer et al., 2018). Therefore, spring waters are subsurface waters, not waters exiting the subsurface. For the geochemical modeling, soil solutions percolate into the regolith all along the flow lines, which imply a homogeneous infiltration of soil solutions into the subsurface.

[revised manuscript text omitted]

---

## Author Response (AR5)

**Editor Decision: Publish subject to technical corrections** (15 May 2020) by Jan Seibert
Comments to the Author:
Thanks for your efforts with this revision and again sorry for the delay due to a combination of unlucky circumstances. I find your manuscript now ok for publication, below just a few technical points:

Could you add a legend to fig 1? Make N-arrow larger, add endpoints to scale

We added a legend to figure 1 to specify the type of sites (spring, stream, bedrock…). N-arrow is larger and there are endpoints to scale.

Please add a scale on all maps (as in fig1)

Done. Maps have scales.

Could you please provide the flow values in table 3 also as specific values for comparison?

Table 3 is not a good place for this, as it is the table for mineral reactive surfaces. But it is possible to see flow values in figures 7, 10 and 11.

Please check the units of flow, you use both l/s and L/s, be consistent!

We use only L/s now.

Fig 9: month should be months, unit of x-axes in lower plots

Done. We use 'months' and provide the unit in lower plots.

Are you aware of this study:
van Meerveld, H. J. I., Kirchner, J. W., Vis, M. J. P., Assendelft, R. S., and Seibert, J.: Expansion and contraction of the flowing stream network alter hillslope flowpath lengths and the shape of the travel time distribution, Hydrol. Earth Syst. Sci., 23, 4825–4834, https://doi.org/10.5194/hess-23-4825-2019, 2019.

Interesting study also highlighting the seasonal variability of flow paths in headwater catchments. We refer this study in the discussion lines 735-737.

No need to cite this, actually I bring it up on purpose first now after accepting your paper, but I thought you might find Ilja's study interesting.

Best regards,

Jan Seibert

[revised manuscript text omitted]